# Nanopore direct RNA sequencing maps the complexity of Arabidopsis mRNA processing and m⁶A modification

Matthew T Parker[1†], Katarzyna Knop[1†], Anna V Sherwood[1†‡],
Nicholas J Schurch[1†§], Katarzyna Mackinnon[1], Peter D Gould[2], Anthony JW Hall[3],
Geoffrey J Barton[1*], Gordon G Simpson[1,4*]

[1]School of Life Sciences, University of Dundee, Dundee, United Kingdom; [2]Institute of Integrative Biology, University of Liverpool, Liverpool, United Kingdom; [3]Earlham Institute, Norwich Research Park, Norwich, United Kingdom; [4]James Hutton Institute, Invergowrie, United Kingdom

**Abstract** Understanding genome organization and gene regulation requires insight into RNA transcription, processing and modification. We adapted nanopore direct RNA sequencing to examine RNA from a wild-type accession of the model plant *Arabidopsis thaliana* and a mutant defective in mRNA methylation (m⁶A). Here we show that m⁶A can be mapped in full-length mRNAs transcriptome-wide and reveal the combinatorial diversity of cap-associated transcription start sites, splicing events, poly(A) site choice and poly(A) tail length. Loss of m⁶A from 3′ untranslated regions is associated with decreased relative transcript abundance and defective RNA 3′ end formation. A functional consequence of disrupted m⁶A is a lengthening of the circadian period. We conclude that nanopore direct RNA sequencing can reveal the complexity of mRNA processing and modification in full-length single molecule reads. These findings can refine Arabidopsis genome annotation. Further, applying this approach to less well-studied species could transform our understanding of what their genomes encode.

**\*For correspondence:**
g.j.barton@dundee.ac.uk (GJB);
g.g.simpson@dundee.ac.uk (GGS)

[†]These authors contributed equally to this work

**Present address:** [‡]Department of Biology, University of Copenhagen, Copenhagen, Denmark; [§]Biomathematics and Statistics Scotland, The James Hutton Institute, Aberdeen, United Kingdom

**Competing interests:** The authors declare that no competing interests exist.

## Introduction

Patterns of pre-mRNA processing and base modifications determine eukaryotic mRNA coding potential and fate. Alternative transcripts produced from the same gene can differ in the position of the start site, the site of cleavage and polyadenylation, and the combination of exons spliced into the mature mRNA. Collectively termed the epitranscriptome, RNA modifications play crucial context-specific roles in gene expression (*Meyer and Jaffrey, 2017*; *Roundtree et al., 2017*). The most abundant internal modification of mRNA is methylation of adenosine at the N6 position (m⁶A). Knowledge of RNA modifications and processing combinations is essential to understand gene expression and what genomes really encode.

RNA sequencing (RNAseq) is used to dissect transcriptome complexity: it involves copying RNA into complementary DNA (cDNA) with reverse transcriptases (RTs) and then sequencing the subsequent DNA copies. RNAseq reveals diverse features of transcriptomes, but limitations can include misidentification of 3′ ends through internal priming (*Jan et al., 2011*), spurious antisense and splicing events produced by RT template switching (*Houseley and Tollervey, 2010*; *Mourão et al., 2019*), and the inability to detect all base modifications in the copying process (*Helm and Motorin, 2017*). The fragmentation of RNA prior to short-read sequencing makes it difficult to interpret the combination of authentic RNA processing events and remains an unsolved problem (*Steijger et al., 2013*).

We investigated whether long-read direct RNA sequencing (DRS) with nanopores (*Garalde et al., 2018*) could reveal the complexity of Arabidopsis mRNA processing and modifications. In nanopore DRS, the protein pore (nanopore) sits in a membrane through which an electrical current is passed, and intact RNA is fed through the nanopore by a motor protein (*Garalde et al., 2018*). Each RNA sequence within the nanopore (five bases) can be identified by the magnitude of signal it produces. Arabidopsis is a pathfinder model in plant biology, and its genome annotation strongly influences the annotation and our understanding of what other plant genomes encode. We applied nanopore DRS and Illumina RNAseq to wild-type Arabidopsis (Col-0) and mutants defective in m$^6$A (*Růžička et al., 2017*) and exosome-mediated RNA decay (*Lange et al., 2014*). We reveal m$^6$A and combinations of RNA processing events (alternative patterns of 5′ capped transcription start sites, splicing, 3′ polyadenylation and poly(A) tail length) in full-length Arabidopsis mRNAs transcriptome-wide.

## Results

### Nanopore DRS detects long, complex mRNAs and short, structured non-coding RNAs

We purified poly (A)+ RNA from four biological replicates of 14-day-old Arabidopsis Col-0 seedlings. We incorporated synthetic External RNA Controls Consortium (ERCC) RNA spike-in mixes into all replicates (*Jiang et al., 2011*; *Reid and External RNA Controls Consortium, 2005*) and carried out nanopore DRS (*Supplementary file 1*). Illumina RNAseq was performed in parallel on similar material (*Supplementary file 2*). Using Guppy base-calling (Oxford Nanopore Technologies) and minimap2 alignment software (*Li, 2018*), we identified around 1 million reads per sample (*Supplementary file 1*). The longest read alignments were 12.7 kb for mRNA transcribed from *AT1G48090*, spanning 63 exons, (*Figure 1A*), and 12.8 kb for mRNA transcribed from *AT1G67120*, spanning 58 exons (*Figure 1—figure supplement 1A*). These represent some of the longest contiguous mRNAs sequenced from Arabidopsis. Among the shortest read alignments were those spanning genes encoding highly structured non-coding RNAs such as UsnRNAs and snoRNAs, for example *U3B* (*Figure 1B*).

### Base-calling errors in nanopore DRS are non-random

We used ERCC RNA spike-ins (*Jiang et al., 2011*) as internal controls to monitor the properties of the sequencing reads. The spike-ins were detected in a quantitative manner (*Figure 1—figure supplement 1B*), consistent with the suggestion that nanopore sequencing is quantitative (*Garalde et al., 2018*). For the portion of reads that align to the reference, sequence identity was 92% when measured against the ERCC RNA spike-ins (*Figure 1—figure supplement 1C*). The errors showed evidence of base specificity (*Figure 1—figure supplement 1D,E*). For example, guanine was under-represented and uracil over-represented in indels and substitutions relative to the reference nucleotide (nt) distribution. In some situations, this bias could impact the utility of interpreting nanopore sequence errors. We used the proovread software tool (*Hackl et al., 2014*) and parallel Illumina RNAseq data to correct base-calling errors in the nanopore reads (*Depledge et al., 2019*).

### Artefactual splitting of raw signal affects transcript interpretation

We detected artefacts caused by the MinKNOW software splitting raw signal from single molecules into two or more reads. As a result, alignments comprising apparently novel 3′ ends were mapped as adjacent to alignments with apparently novel 5′ ends (*Figure 1—figure supplement 1F*). A related phenomenon called over-splitting was recently reported in nanopore DNA sequencing (*Payne et al., 2019*). Over-splitting can be detected when two reads sequenced consecutively through the same pore are mapped to adjacent loci in the genome (*Payne et al., 2019*). Over-splitting in nanopore DRS generally occurs at low frequency (< 2% of reads). However, RNAs originating from specific gene loci, such as *RH3* (*AT5G26742*), appear to be more susceptible, with up to 20% of reads affected across multiple sequencing experiments (*Figure 1—figure supplement 1F*).

### Spurious antisense reads are rare or absent in nanopore DRS

Two out of 9,445 (0.02%) reads mapped antisense to the ERCC RNA spike-in collection (*Jiang et al., 2011*). None of 19,665 reads mapped antisense to the highly expressed gene *RUBISCO ACTIVASE*

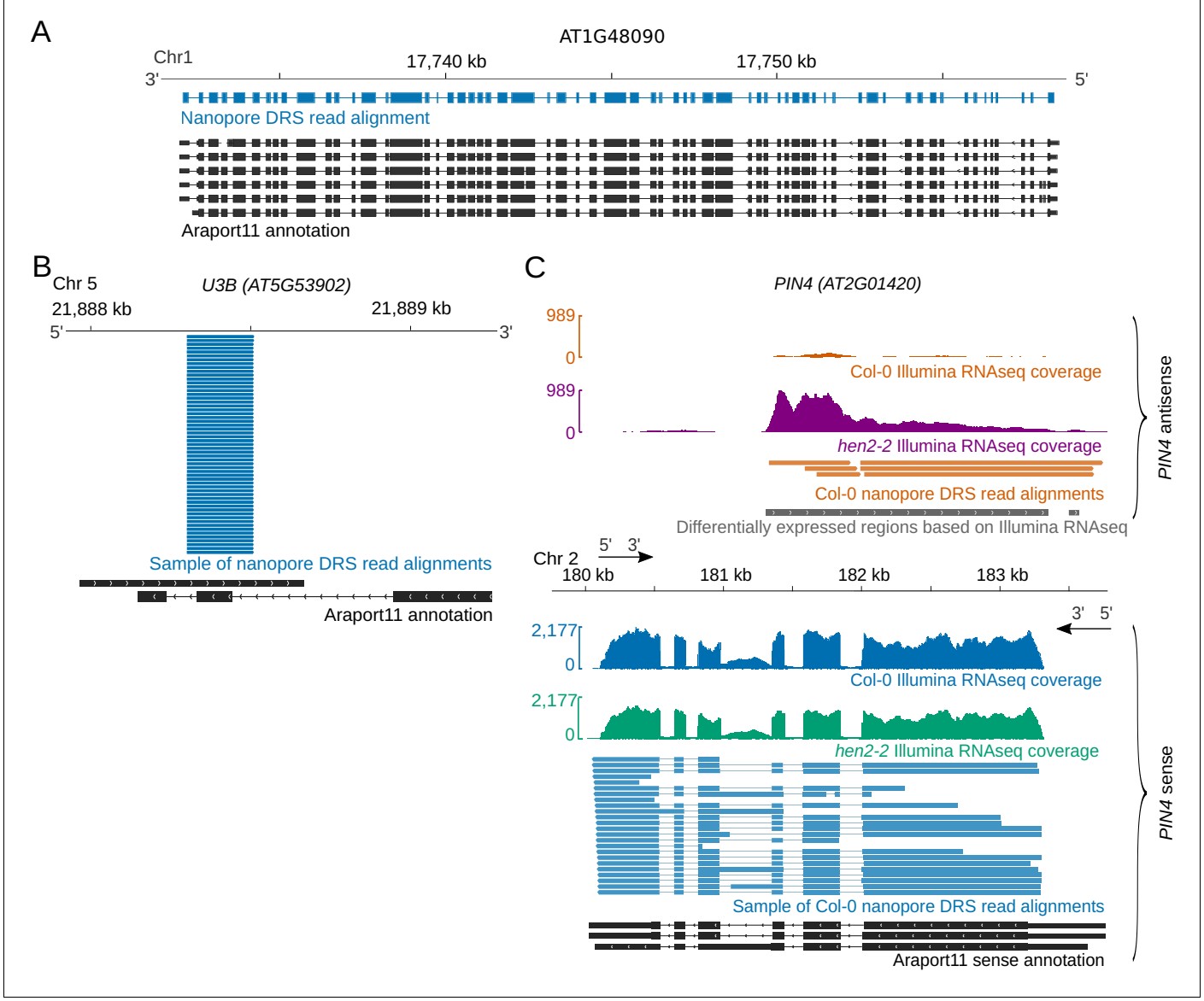

**Figure 1.** Diverse Arabidopsis RNAs are detected by nanopore DRS. (**A**) Nanopore DRS 12.7 kb read alignment at *AT1G48090*, comprising 63 exons. Blue, nanopore DRS read alignments; black, Araport11 annotation. (**B**) Nanopore DRS read alignments at the snoRNA gene *U3B (AT5G53902)*. Blue, sample of nanopore DRS read alignments; black, Araport11 annotation. (**C**) *PIN4 (AT2G01420)* antisense RNAs detected using nanopore DRS. Blue, Col-0 *PIN4* sense Illumina RNAseq coverage and sample of nanopore *PIN4* sense read alignments; orange, Col-0 *PIN4* antisense Illumina RNAseq coverage and nanopore *PIN4* antisense read alignments; green, *hen2-2* mutant *PIN4* sense Illumina RNAseq coverage; purple, *hen2-2* mutant *PIN4* antisense Illumina RNAseq coverage; black, *PIN4* sense RNA isoforms found in Araport11; grey, *PIN4* antisense differentially expressed regions detected in Illumina RNAseq with DERfinder.

The online version of this article includes the following source data and figure supplement(s) for figure 1:

**Source data 1.** *hen2-2* vs Col-0 differentially expressed regions from Ilumina RNAseq –*Figure 1C*.

**Figure supplement 1.** Properties of nanopore DRS sequencing data.

(*RCA*) (*AT2G39730*), consistent with the lack of antisense RNAs annotated at this locus. Consequently, we conclude that spurious antisense is rare or absent from nanopore DRS data (*Wyman, 2019*; *Mourão et al., 2019*). This simplifies the interpretation of authentic antisense RNAs, which is important in Arabidopsis because the distinction between RT-dependent template switching and authentic antisense RNAs produced by RNA-dependent RNA polymerases that copy mRNA is not straightforward (*Matsui et al., 2017*). For example, by nanopore DRS, we could identify

Arabidopsis long non-coding antisense RNAs, such as those at the auxin efflux carriers *PIN4* and *PIN7* (*Figure 1C*, *Figure 1—figure supplement 1G*). The existence of these previously unannotated antisense RNAs was supported by Illumina RNAseq of wild-type Col-0 and the exosome mutant *hen2–2* (*Figure 1C*, *Figure 1—figure supplement 1G*) (*Supplementary file 2A*), the latter of which had a 13-fold increase in abundance of these antisense RNAs. Consequently, the low level of steady-state accumulation of some antisense RNAs may explain why they are currently unannotated.

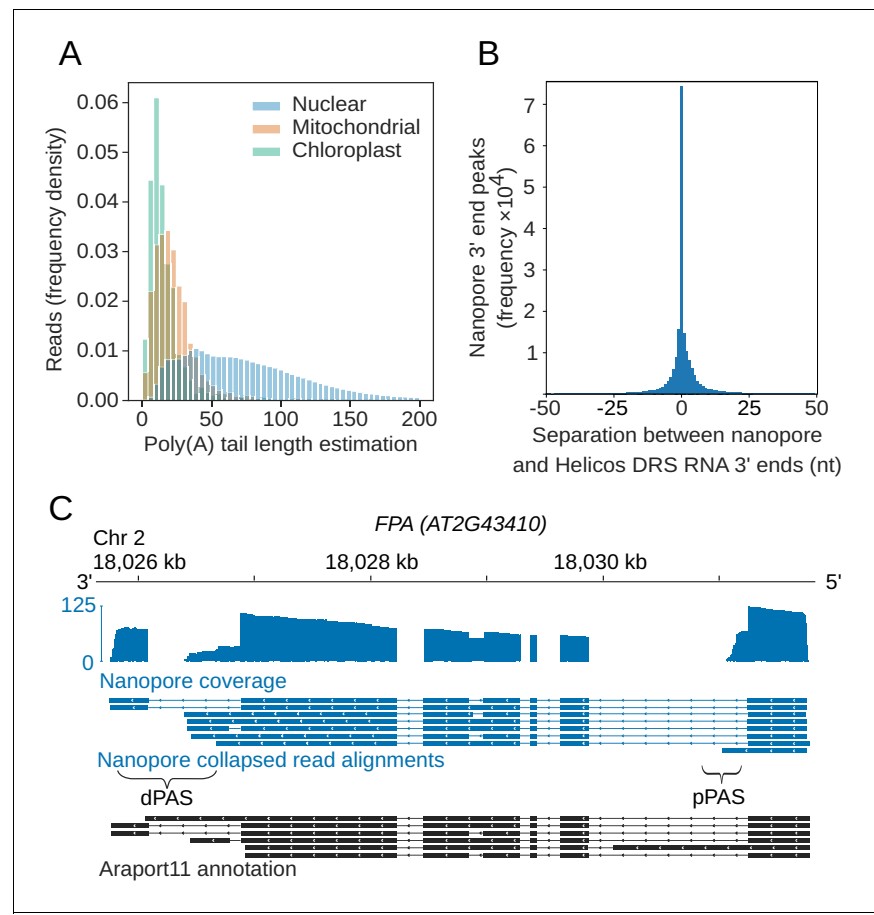

**Figure 2.** Nanopore DRS reveals poly(A) tail length and maps 3′ cleavage and polyadenylation sites. (**A**) Normalized histogram showing poly(A) tail length of RNAs encoded by different genomes in Arabidopsis. Blue, nuclear (n = 2,348,869 reads); orange, mitochondrial (n = 2,490 reads); green, chloroplast (n = 1,848 reads). (**B**) Distance between the RNA 3′ end positions in nanopore DRS read alignments and the nearest polyadenylation sites identified by Helicos data. (**C**) Nanopore DRS identified 3′ polyadenylation sites in RNAs transcribed from *FPA (AT2G43410)*. Blue track, coverage of nanopore DRS read alignments; blue, collapsed read alignments representing putative transcript annotations detected by nanopore DRS; black, Araport11 annotation; pPAS, proximal polyadenylation site; dPAS, distal polyadenylation sites.

The online version of this article includes the following source data and figure supplement(s) for figure 2:

**Source data 1.** Poly(A) tail length estimations generated from Col-0 reads – *Figure 2A*.

**Figure supplement 1.** 3′ end processing is revealed by nanopore DRS.

**Figure supplement 1—source data 1.** Poly(A) tail length estimations generated from ERCC spike-in reads – *Figure 2—figure supplement 1A*.

**Figure supplement 1—source data 2.** Per gene poly(A) tail length estimate distributions generated from Col-0 reads - *Figure 2—figure supplement 1B*.

## Nanopore DRS confirms sites of RNA 3′ end formation and estimates poly(A) tail length

Ligation of the motor protein adapter to RNA 3′ ends results in nanopores sequencing mRNA poly (A) tails first. We used the nanopolish-polyA software tool to estimate poly(A) tail lengths for individual reads (*Workman et al., 2019*). Nanopolish-polyA segments the raw signal into regions corresponding to the mRNA and poly(A) tail and uses the length of these regions to estimate poly(A) tail length. Nanopolish-polyA was developed using *in vitro* transcribed RNA appended with poly(A) tails of defined lengths ranging from 10 to 100 nt (*Workman et al., 2019*). We first examined the ability of this tool to estimate the known lengths of poly(A) tails in the ERCC spike-ins within our own datasets (*Figure 2—figure supplement 1A*). ERCC-00002 has an expected poly(A) tail length of 24 nt, and the median nanopolish polyA estimate was 23.3 nt (95% confidence intervals [CIs] [5.7, 102]; n = 1,485). ERCC-00074 also has an expected poly(A) tail length of 24 nt, and the median nanopolish-polyA estimate was 27.4 nt (95% CIs [8.9, 118]; n = 1,439). We conclude that the median nanopolish-polyA estimates of poly(A) tail length are accurate within a few bases, but individual read measurements have a wide margin of error. Application of this approach to reads mapping to the Arabidopsis genome indicated a median length estimate of 68 nt for Arabidopsis mRNA poly(A) tails, but with a wide range in estimated lengths for individual mRNAs (95% were in the 13–200 nt range; *Figure 2A*). The generally shorter poly(A) tails of chloroplast- (median length 13 nt, 95% CIs [3.1, 56]) and mitochondria-encoded (median length 20 nt, 95% CIs [4.1, 73] transcripts, which are a feature of RNA decay in these organelles, were also detectable. We find that poly(A) tail length correlates negatively (Spearman's $\rho = -0.34$, $p < 1 \times 10^{-16}$, 95% CIs [−0.36,–0.32]) with gene expression in Arabidopsis (*Figure 2—figure supplement 1B*), consistent with other species analysed by short-read TAILseq (*Lima et al., 2017*).

We previously mapped Arabidopsis mRNA 3′ ends transcriptome-wide using Helicos direct RNA sequencing (DRS) (*Sherstnev et al., 2012* ) We compared the position of 3′ ends of nanopore DRS read alignments and Helicos DRS data genome-wide. The median genomic distance between nanopore DRS and Helicos DRS 3′ ends was $0 \pm 13$ nt (one standard deviation) demonstrating close agreement between these orthogonal technologies (*Figure 2B*). Likewise, the overall distribution of the 3′ ends of aligned nanopore DRS reads resembles the pattern we previously reported with Helicos data (*Sherstnev et al., 2012*). For example, 97% of nanopore DRS 3′ ends (4,152,800 reads at 639,178 unique sites, 93% of all unique sites) mapped to either annotated 3′ untranslated regions (UTRs) or downstream of the current annotation. Mapping of 3′ ends to coding sequences or 5′UTRs was rare (2.8%, 119,524 reads at 39,610 unique sites, 5.8% of all unique sites), and mapping to introns even rarer (0.29%, 12,554 reads at 7791 unique sites, 1.1% of unique sites). Even so, examples of the latter included sites of alternative polyadenylation with well-established regulatory roles, such as in mRNA encoding the RNA-binding protein FPA, which controls flowering time (*Hornyik et al., 2010*) (*Figure 2C*), and in mRNA encoding the histone H3K9 demethylase IBM1, which controls levels of genic DNA methylation (*Rigal et al., 2012*) (*Figure 2—figure supplement 1C*). In addition, we identified novel examples of intronic alternative polyadenylation predicted to influence gene function. For example, we found frequent cleavage and polyadenylation at mRNA encoding the PTM homeodomain transcription factor (AT5G35210) (*Figure 2—figure supplement 1D*). PTM mediates retrograde signalling upon cleavage of a C-terminal transmembrane domain that sequesters it to the chloroplast (*Feng et al., 2016*). Cleavage and polyadenylation within *PTM* intron 10 (supported by our Helicos DRS data) terminates transcripts prior to sequence encoding the transmembrane domain, hence bypassing established retrograde control.

Since RT-dependent internal priming can result in the misinterpretation of authentic cleavage and polyadenylation sites (*Jan et al., 2011*), we next determined whether nanopore DRS was compromised in this way. To address this issue, we examined whether the 3′ ends of nanopore DRS reads mapped to potential internal priming substrates comprised of six consecutive adenosines within a transcribed coding sequence (according to the Arabidopsis Information Portal Col-0 genome annotation, Araport11 *Cheng et al., 2017*). Of the 10,116 such oligo(A)$_6$ sequences, only four have read alignments terminating within 13 nt in all four datasets. Of these, two were not detectable after error correction with proovread (suggesting that they resulted from alignment errors) and the other two mapped to the terminal exon of coding sequence annotation, indicating that they may be authentic 3′ ends. Hence, internal priming is rare or absent in nanopore DRS data. It is possible that a standard

nanopore DRS approach may be compromised in the quantitative detection of RNAs with very short poly(A) tails or tails incorporating uridylation for decay. However, overall, we conclude that nanopore DRS can identify multiple authentic features of RNA 3′ end processing.

## Cap-dependent 5′ RNA detection by nanopore DRS

Nanopore DRS reads are frequently truncated prior to annotated transcription start sites, resulting in a 3′ bias of genomic alignments (*Figure 3A*) (*Depledge et al., 2019*). Consequently, it is impossible to determine which, if any, aligned reads correspond to full-length transcripts. To address this issue, we used cap-dependent ligation of a biotinylated 5′ adapter RNA to purify capped mRNAs (a related approach has recently been used by others *Jiang et al., 2019*). We then re-sequenced two biological replicates of Arabidopsis Col-0 incorporating 5′ adapter ligation (*Supplementary file 1*) and filtered the reads for 5′ adapter RNA sequences using the sequence alignment tool BLASTN and specific criteria (*Supplementary file 3*). We then used high confidence examples of sequences that passed or failed these criteria to train a convolutional neural network to detect the 5′ adapter RNA in the raw signal (*Figure 3—figure supplement 1A–C*). Hence, we improved 5′ adapter-ligated RNA detection without requiring base-calling or genome alignment, and demonstrated enrichment of full-length, cap-dependent mRNA sequences (*Figure 3A,B*). This procedure reduced the median 3′ bias of nanopore read alignments per gene (as measured by quartile coefficient of variation of per base coverage) from 0.45 (95% CIs [0.43,0.47]) to 0.08 (95% CIs [0.07,0.09]; *Figure 3B*).

In order to determine whether the 5′ ends we detected reflect full-length mRNAs, we compared them against transcription start sites in datasets derived from two orthogonal approaches. First, we compared the nanopore cap-capture 5′ ends with full-length Arabidopsis cDNA clones (*Seki et al., 2002*). We found that 41% of adapter-ligated nanopore DRS reads mapped within 5 nt of transcription start sites and 60% mapped within 13 nt (*Figure 3C*). Next, we compared the nanopore cap-captured 5′ ends to Arabidopsis data obtained using an Illumina-based high throughput 5′ mapping approach called nanoPARE (parallel analysis of RNA ends) (*Schon et al., 2018*). We found that 75% of nanopore DRS cap-capture 5′ ends mapped within 5 nt of a transcription start site detected using single base resolution peak-calling from nanoPARE reads, and 82% mapped with 13 nt (*Figure 3C*). This closer agreement is presumably due to the increased depth of nanoPARE data leading to greater coverage of alternative transcription start sites. We also detected recently defined examples of alternative 5′ transcription start sites (*Ushijima et al., 2017*) at specific Arabidopsis genes, which are also supported by nanoPARE data (*Figure 3—figure supplement 1D*). We therefore conclude that this approach is effective in detecting authentic mRNA 5′ ends.

Reads with adapters had, on average, 11 nt more at their 5′ ends that could be aligned to the genome compared with the most common 5′ alignment position of reads lacking the 5′ adapter RNA (*Figure 3D*). This difference may be explained by loss of processive control by the motor protein when the end of an RNA molecule enters the pore. As a result, the 5′ end of RNA is not correctly sequenced. Consistent with these Arabidopsis transcriptome-wide nanopore DRS data, reads mapping to the synthetic ERCC RNA Spike-Ins and *in vitro* transcribed RNAs also lacked ~11 nt of authentic 5′ sequence (*Figure 3—figure supplement 1E,F*). However, the precise length of 5′ sequence missing from all of these RNAs varied, suggesting that sequence- or context-specific effects on sequence accuracy are associated with the passage of 5′ RNA through the pore (*Figure 3D*, *Figure 3—figure supplement 1E,F*).

Despite the close agreement between nanopore DRS, Illumina RNAseq and full-length cDNA data (*Seki et al., 2002*) at *RCA*, the start site annotated in Araport11 and the *Arabidopsis thaliana* Reference Transcript Dataset 2 (AtRTD2) (*Zhang et al., 2017*) is quite different (*Figure 3E*). The apparent overestimation of 5′UTR length is widespread in Araport11 annotation (*Figure 3—figure supplement 1G*), consistent with the assessment of capped Arabidopsis 5′ ends detected by nanoPARE sequencing (*Schon et al., 2018*). Consequently, with appropriate modification to the current protocol, such as we describe here, nanopore DRS data can be used to revise Arabidopsis transcription start site annotations.

We next asked whether the truncation of reads prior to predicted TSSs in conventional nanopore DRS data might inform features of biological RNA decay. We examined if there was enrichment of 5′ nanopore DRS reads terminating in proximity to miRNA cleavage sites predicted in protein coding and non-coding RNAs. We identified peaks of downstream cleavage products at mRNA encoding, for example, HAM1, which is targeted by miR170 and miR171 (*Figure 3—figure supplement 1H*),

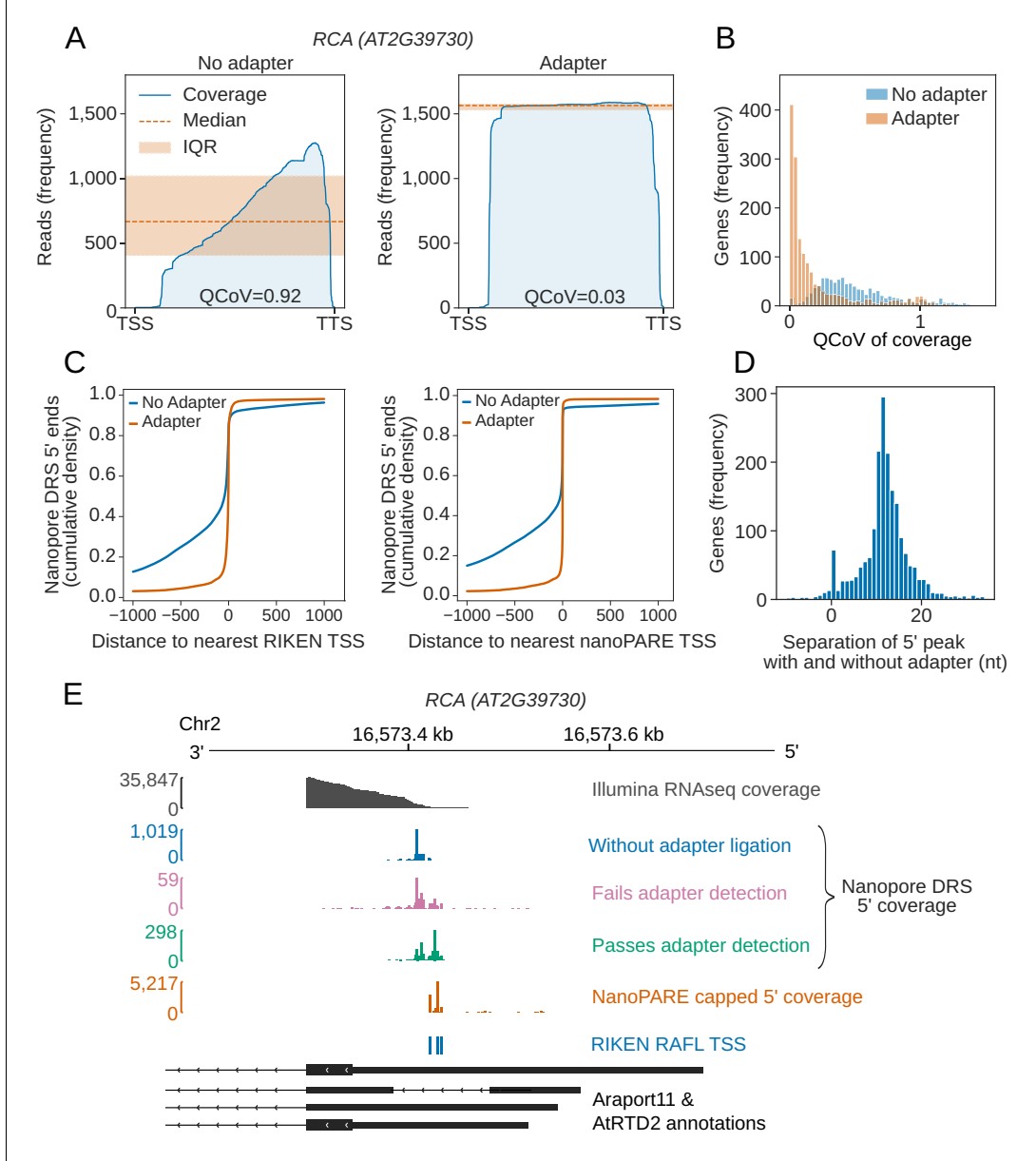

**Figure 3.** Cap-dependent ligation of an adapter enables detection of authentic RNA 5′ ends. (**A**) 5′ adapter RNA ligation reduces 3′ bias in nanopore DRS data at *RCA* (*AT2G39730*) from 0.92 to 0.03. Blue line, exonic read coverage at *RCA* for reads without (left) and with adapter (right); orange line, median coverage; orange shaded area, interquartile range (IQR). Change in 3′ bias can be measured using the IQR/median = quartile coefficient of variation (QCoV). (**B**) 5′ adapter RNA ligation reduces 3′ bias in nanopore DRS data. Histogram showing the QCoV in per base coverage for each gene in the Araport11 annotation, for reads with a 5′ adapter RNA (orange), and reads without a 5′ adapter RNA (blue). (**C**) Cap-dependent adapter ligation allows identification of authentic 5′ ends using nanopore DRS. The cumulative distribution function shows the distance to the nearest Transcription Start Site (TSS) identified from full-length transcripts cloned as part of the RIKEN RAFL project (left), or 5′ tag identified from nanoPARE data (right), for reads with a 5′ adapter RNA (orange) compared with reads without a 5′ adapter RNA (blue). (**D**) Cap-dependent adapter ligation enabled resolution of an additional 11 nt of sequence at the RNA 5′ end. Histogram showing the nucleotide shift in the largest peak of 5′ coverage for each gene in data obtained using protocols with vs without a 5′ adapter RNA ligation. (**E**) For *RCA* (*AT2G39730*), the 5′ end identified using cap-dependent 5′ adapter RNA ligation protocol was consistent with Illumina RNAseq and full-length cDNA start site data but differed from the 5′ ends in the Araport11 and AtRTD2 annotations. Grey, Illumina RNAseq coverage; blue, nanopore DRS 5′ end coverage generated without a cap-dependent ligation protocol; green/pink, nanopore DRS 5′ end coverage for read alignments generated using the cap-dependent ligation protocol with

*Figure 3 continued*

(green) and without (pink) 5′ adapter RNA; orange, 5′ coverage of nanoPARE data; blue, TSSs identified from full-length transcripts cloned as part of the RIKEN RAFL project; black, Araport11 and AtRTD2 annotations (with duplicated 5′ positions removed).

The online version of this article includes the following source data and figure supplement(s) for figure 3:

**Source data 1.** Transcriptional start site tags from RIKEN cDNA clones – *Figure 3C*.
**Source data 2.** Transcriptional start site tags from nanoPARE sequencing – *Figure 3C*.
**Figure supplement 1.** Nanopore DRS with cap-dependent ligation of 5′ adapter RNA.
**Figure supplement 1—source data 1.** microRNA cleavage site predictions supported by enrichment of nanopore 5′ ends – *Figure 3—figure supplement 1H*.

---

NAC1 (targeted by miR164), ARF proteins (targeted by miR160) and SPL domain-containing proteins (targeted by miR156/7) (*Supplementary file 4*). In addition, we could detect cleavage of the non-coding *TAS* RNAs that are targeted by miR173 (*Supplementary file 4*). Therefore, nanopore DRS can reveal regions of internal mRNA and ncRNA cleavage. Since nanopore DRS does not sequence the 5′ end of RNA accurately, precision in this approach would be improved by ligation of an adapter to uncapped RNAs (*Schon et al., 2018*).

## Nanopore DRS reveals the complexity of splicing events

In single reads, nanopore sequencing revealed some of the most complex splicing combinations so far identified in the Arabidopsis transcriptome. For example, analysis of error corrected reads revealed the splicing pattern of a 12.7 kb read alignment, comprised of 63 exons, agreeing exactly with the *AT1G48090.4* isoform annotated in Araport11 (*Figure 1A*). Mutually exclusive alternative splicing of *FLM (AT1G77080)* that mediates the thermosensitive response controlling flowering time (*Posé et al., 2013*) was also detected (*Figure 4A*). However, a combination of base-calling and alignment errors contributed to the misidentification of splicing events for uncorrected DRS data: 58% (170,702) of the unique splice junctions detected in the combined set of replicate data were absent from Araport11 and AtRTD2 annotations and were unsupported by Illumina RNAseq (*Figure 4B*, *Supplementary file 5*). We applied proovread (*Hackl et al., 2014*; *Depledge et al., 2019*) error correction with the parallel Illumina RNAseq data and then re-analysed the corrected and uncorrected nanopore DRS data. After error correction, only 13% (39,061) of unique splice junctions were unsupported by an orthogonal dataset, consistent with an improvement in alignment accuracy. The four nanopore DRS datasets for Col-0 biological replicates captured 75% (102,486) and 69% (104,686) of Araport11 and AtRTD2 splice site annotations, respectively. Most of the canonical GU/AG splicing events (100,450; 81%) detected in the error-corrected nanopore data were found in both annotations and were supported by Illumina RNAseq (*Figure 4B*, *Supplementary file 5*). A total of 3,234 unique canonical splicing events in the error-corrected nanopore DRS data were supported by Illumina RNAseq but absent from both Araport11 and AtRTD2 annotations, highlighting potential gaps in our understanding of the complexity of Arabidopsis splicing annotation (*Figure 4B*, *Supplementary file 5*). Consistent with this, we validated three of these newly identified splicing events using RT-PCR (reverse transcription polymerase chain reaction) followed by cloning and Sanger sequencing (*Figure 4—figure supplement 1A*). In order to examine the features of these unannotated splicing events, we applied previously determined splice site position weight matrices of the flanking sequences to categorize U2 or U12 class splice sites (*Sheth et al., 2006*). Of the 3,234 novel GU/AG events found in error-corrected data and supported by Illumina alignments, 74% were classified as canonical U2 or U12 splice sites, suggesting that they are authentic (*Figure 4—figure supplement 1B*).

In addition to previously unannotated splicing events, we identified unannotated combinations of previously established splice sites. For example, we identified 19 *FLM* splicing patterns that adhered to known splice junction sites (*Figure 4A*). However, 11 of these transcript isoforms were not previously annotated. In order to investigate this phenomenon transcriptome-wide, we analysed the 5′ cap-dependent nanopore DRS datasets of full-length mRNAs (*Supplementary file 1*). Unique sets of co-splicing events were extracted from error-corrected reads (so as to focus on splicing, we did not consider single exon reads or 5′ and 3′ positions). In total, 13,064 unique splicing patterns were detected that matched annotations in Araport11, AtRTD2 or both (*Figure 4C*). Another 8,659

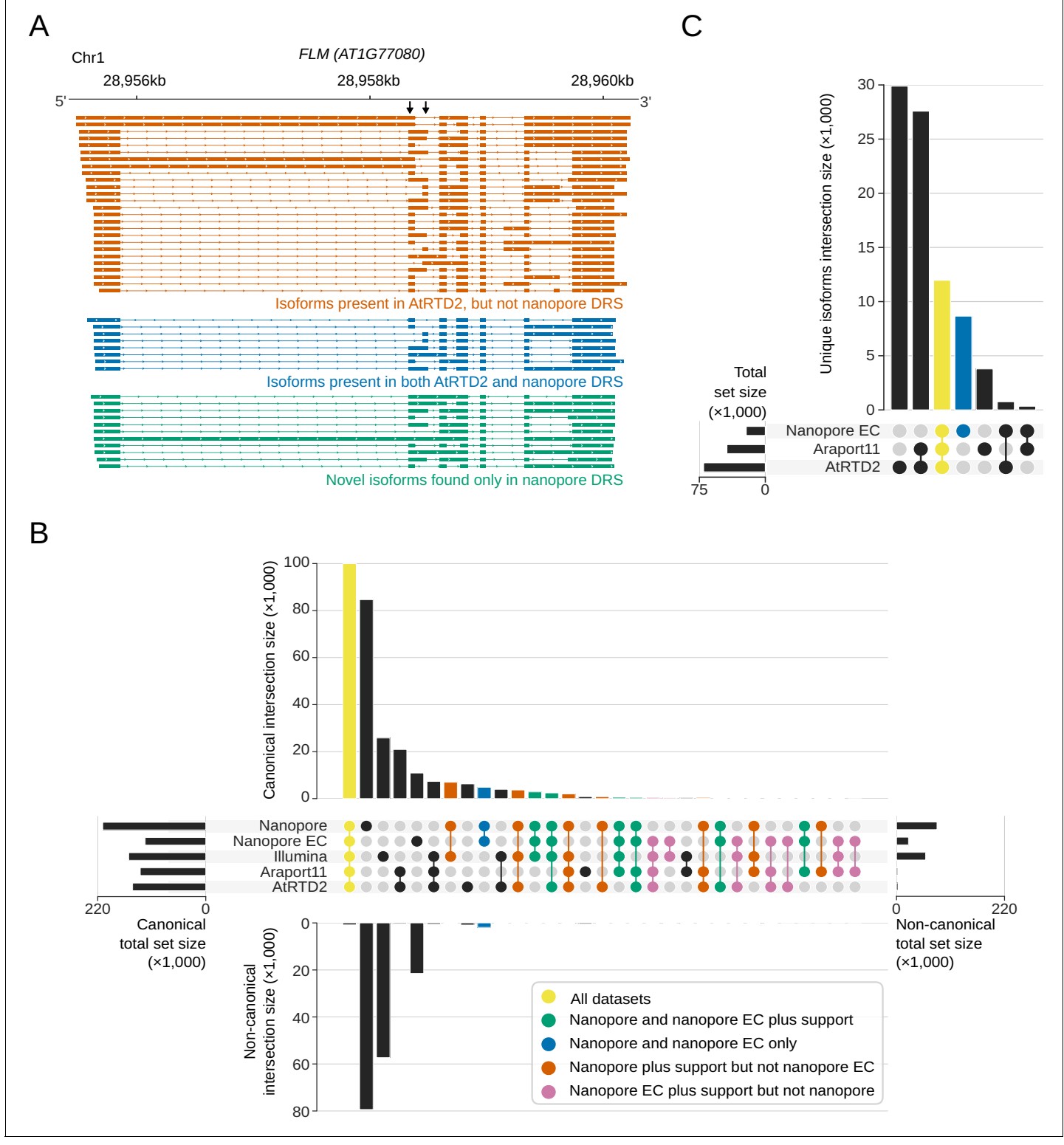

**Figure 4.** Nanopore DRS reveals the complexity of alternative splicing. (**A**) Nanopore DRS identified the mutually exclusive alternative splicing of the second and third exons of *FLOWERING LOCUS M* (*FLM, AT1G77080)* and several novel isoforms. Black arrows indicate mutually exclusive exons. Orange, isoforms present in the AtRTD2 annotation but not identified using nanopore DRS; blue, isoforms common to both AtRTD2 and nanopore DRS; green, novel isoforms identified in nanopore DRS. (**B**) Comparison of splicing events identified in error-corrected and non-error-corrected nanopore DRS, Illumina RNA sequencing, and Araport11 and AtRTD2 annotations. Bar size represents the number of unique splicing events common to the set intersection highlighted using circles (see ***Supplementary file 5*** for the exact values). GU/AG splicing events are shown on the top and non-

*Figure 4 continued on next page*

*Figure 4 continued*

GU/AG on the bottom of the plot: yellow, splicing events common to all five datasets; green, events common to both error-corrected and non-error-corrected nanopore DRS with support in orthogonal datasets; blue, events common to both nanopore DRS datasets without orthogonal support; orange, events found in uncorrected nanopore DRS (but not error corrected) with orthogonal support; pink, events found in error-corrected nanopore DRS (but not uncorrected) with orthogonal support. (C) Comparison of RNA isoforms (defined as sets of co-spliced introns) identified in error-corrected full-length nanopore DRS, Araport11 and AtRTD2 annotations. Bar size represents the number of splicing events common to a group highlighted using circles below (see *Supplementary file 5* for the exact values): yellow, unique splicing patterns found in nanopore DRS and both reference annotations; blue, novel isoforms found only in nanopore DRS.

The online version of this article includes the following source data and figure supplement(s) for figure 4:

**Figure supplement 1.** Patterns of splicing revealed using nanopore DRS.
**Figure supplement 1—source data 1.** Sanger sequencing products from novel nanopore DRS splicing events – *Figure 4—figure supplement 1A*.

unique splicing patterns were identified that were not present in either annotation (*Figure 4C*, *Supplementary file 5*). Of these, 50% (4,293) used only splice donor and acceptor pairs that were already annotated in either Araport11 or AtRTD2. Hence, this approach defines splicing patterns (including retained introns) produced from alternative combinations of known splice sites.

We next asked whether these full-length nanopore DRS reads could reveal alternative splicing events with a predicted functional impact, such as the introduction of premature termination codons (PTCs) that can target mRNAs for nonsense-mediated decay (NMD). We could identify established cases of alternative splicing that lead to NMD (*Kalyna et al., 2012*) (*Supplementary file 6*). In addition, by highlighting reads overlapping annotated ORFs, with splicing events that would shift the annotated stop codon out of frame, we identified novel events that could lead to NMD. For example, exon skipping at mRNA encoding a KH domain protein (AT5G56140) and the use of alternative acceptor sites at mRNA encoding XRCC (AT1G80420) introduce PTCs. In both cases, the splicing events detected with nanopore DRS were supported by reads from our Illumina RNAseq data.

Overall, we conclude that nanopore DRS can reveal a greater complexity of splicing in the context of full-length mRNAs compared with short-read data. However, accurate splice pattern detection benefits from error correction with, for example, high-accuracy orthogonal short-read sequencing data. However, even with error-free sequences, accurate splice detection can be confounded by the existence of equivalent alternative junctions (*Dehghannasiri et al., 2019*). Therefore, improved computational tools are required, not only for basecalling or error correction but also for splicing-aware long-read alignment.

## Differential error site analysis reveals m⁶A modifications transcriptome-wide

mRNA modifications have emerged recently as a crucial, but relatively neglected, layer of gene regulation (*Meyer and Jaffrey, 2017*; *Roundtree et al., 2017*). m$^6$A has been mapped transcriptome-wide using approaches based on antibodies that recognize this mark (*Helm and Motorin, 2017*; *Linder et al., 2015*). However, in principle, m$^6$A can be detected by nanopore DRS (*Garalde et al., 2018*). Since m$^6$A is not included in the training data for nanopore base-calling software, we asked whether its incorrect interpretation could be used to identify m$^6$A in RNA. To test this possibility, we examined whether we could detect m$^6$A in a matched set of synthetic RNAs that differ only in the presence or absence of N6 methylation at a single adenosine. We selected a 60 nt portion of the human MALAT1 lncRNA with a previously reported m$^6$A site (*Liu et al., 2013*). The m$^6$A modified and unmodified RNAs were chemically synthesised, complete with a 50 nt poly(A) tail and sequenced using nanopore DRS. Using read alignments to the known sequence of the synthetic RNA, we applied a G-test to detect differences in the basecall profile of alignments at each reference position. Using this differential error rate approach, we could correctly map the position of m$^6$A in these synthetic RNAs (*Figure 5—figure supplement 1A*).

We next asked, if this same approach could be used to map Arabidopsis m$^6$A transcriptome-wide. For this, we applied nanopore DRS to four biological replicates of an Arabidopsis mutant defective in the function of *VIRILIZER* (*vir-1*), a conserved m$^6$A writer complex component, and four biological replicates of a line expressing VIR fused to Green Fluorescent Protein (GFP) (VIR complemented; VIRc) that restores VIR activity in the *vir-1* mutant background (*Růžička et al., 2017*) (*Figure 5—figure supplement 1B*). In parallel, we sequenced six biological replicates of each genotype

with Illumina RNAseq (*Supplementary file 2B*). We used the differential error rate analysis approach to compare the mutant (defective m$^6$A) and VIR-complemented lines. We identified 17,491 sites with a more than two-fold higher error rate (compared with the TAIR10 reference base) in the VIR-complemented line with restored m$^6$A (*Figure 5A*). No VIR-dependent error sites mapped to either chloroplast or mitochondrial-encoded RNAs. In all, 99.4% of the differential error sites mapped within Araport11 annotated protein-coding genes. Motif analysis of these error sites revealed the DRAYH (D = G or U or A, R = G or A, Y = C or U, H = A or C or U) sequence (E value < 1 × 10$^{-16}$), which closely resembles the established m$^6$A target consensus (*Meyer and Jaffrey, 2017*; *Roundtree et al., 2017*) (*Figure 5B*). The most frequently detected motifs in this analysis, AAm$^6$ACU and AAm$^6$ACA (*Figure 5—figure supplement 1C*), match those most frequently detected in Arabidopsis using the orthogonal Me-RIP approach (*Luo et al., 2014*). Notably, we also detect 187 sites (4.8%) associated with the motif AGAUU (*Figure 5—figure supplement 1C*), raising the possibility that a C following m$^6$A is not an invariant feature of the Arabidopsis m$^6$A code. In addition, like the established location of m$^6$A sites in mRNAs (*Meyer and Jaffrey, 2017*; *Roundtree et al., 2017*), the error sites were preferentially found in 3'UTRs (*Figure 5C*). The relatively low resolution of Me-RIP analysis is associated with an interpretation of enrichment of m$^6$A over stop codons. However, consistent with the development of other more accurate methods of mapping m$^6$A (*Ke et al., 2015*), nanopore DRS analysis does not support this (*Figure 5D*). Since approximately five nt contribute to the observed current at a given time point in nanopore sequencing (*Garalde et al., 2018*), the presence of a methylated adenosine could affect the accuracy of base-calling for the surrounding nucleotides. Consistent with this, we identified 3,871 unique sequences (False Discovery Rate [FDR] < 0.1) matching the motif discovered at error sites (*Figure 5—figure supplement 1C*), with a median of one error site per motif (95% CIs [1, 5]). Overall, these results agree with the established and conserved properties of authentic m$^6$A sites (*Meyer and Jaffrey, 2017*; *Roundtree et al., 2017*), suggesting that differential error sites can be used to identify thousands of m$^6$A modifications in nanopore DRS datasets.

In order to examine the validity of m$^6$A sites identified by the differential error site analysis, we used an orthogonal technique to map m$^6$A. Previous maps of Arabidopsis m$^6$A are based on Me-RIP (*Shen et al., 2016a*; *Anderson et al., 2018*) and limited by a resolution of around 200 nt (*Ke et al., 2015*). Therefore, to examine Arabidopsis m$^6$A sites with a more accurate method, we used miCLIP (*Linder et al., 2015*) analysis of two biological replicates of Arabidopsis Col-0. We found that, like the differential error sites uncovered in the nanopore DRS analysis, the Arabidopsis miCLIP reads were enriched in 3' UTRs but with no enrichment over stop codons (*Figure 5D*, *Figure 5—figure supplement 1D*). In all, 66% of the called nanopore DRS differential error sites fell within 5 nt of an miCLIP peak (*Figure 5E,F*). We therefore conclude that our analysis of nanopore DRS data can detect authentic VIR-dependent m$^6$A sites transcriptome-wide.

We next examined how the power of the differential error rate analysis approach to detect m$^6$A depends upon sequencing depth. To do this we used a random sub-sampling approach at different read depths. Using the single replicate of MALAT1 synthetic RNA nanopore DRS data we were able to detect m$^6$A at the correct position in more than 95% of random sub-samples with 60 reads per condition (*Figure 5—figure supplement 1E*). Since the stoichiometry of m$^6$A varies in nature, we replaced some of the sample of methylated reads with unmethylated reads in silico. When 50% of reads in the methylation positive condition are sampled from the unmethylated condition, more than 200 reads per condition are required to reliably detect m$^6$A in > 95% bootstraps, indicating that stoichiometry is an important factor in the power to detect differentially methylated sites (*Figure 5—figure supplement 1E*). We next performed a sub-sampling analysis for a selection of m$^6$A sites using the Arabidopsis nanopore DRS datasets. We found that the number of reads required to detect m$^6$A varies between Arabidopsis genes and between methylated positions in transcripts from the same gene. Presumably, these distinctions reflect differences in stoichiometry (or possibly sequence context) (*Figure 5—figure supplement 1F*). For some positions, we can reliably detect m$^6$A with only 10 reads per replicate (in a four replicate per condition experiment).

## Defective m$^6$A perturbs gene expression patterns and lengthens the circadian period

The combination of transcript processing and modification data obtained using nanopore DRS enabled us to investigate the impact of m$^6$A on Arabidopsis gene expression. We found a global

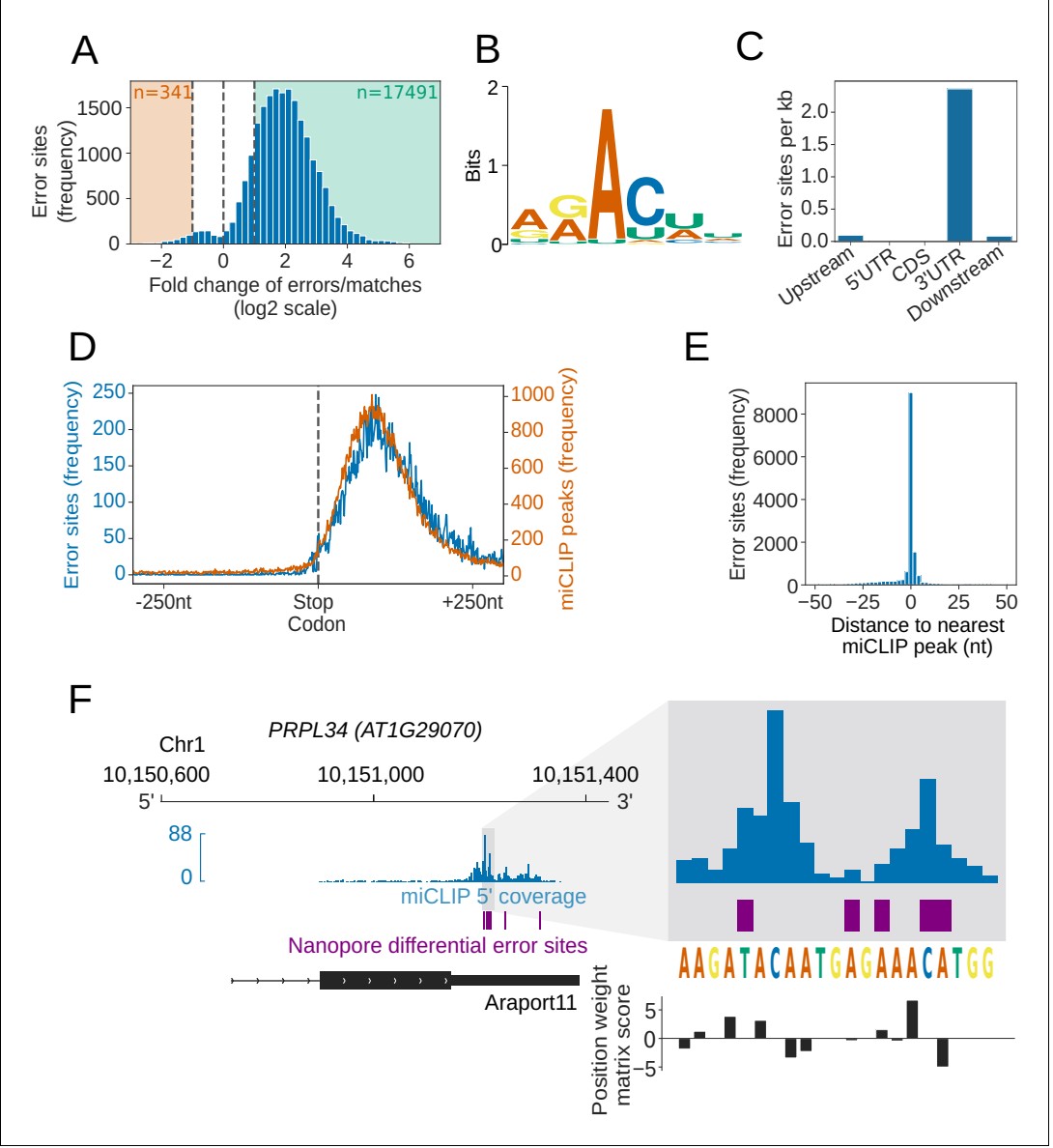

**Figure 5.** Differential error rate analysis identifies sites of VIR-dependent m⁶A modifications transcriptome-wide.
(A) Loss of VIR function is associated with reduced error rate in nanopore DRS. Histogram showing the $\log_2$ fold change in the ratio of errors to reference matches at bases with a significant change in error profile in *vir-1* mutant compared with the VIR-complemented line. Orange and green shaded regions indicate sites with increased and reduced errors in *vir-1*, respectively. (B) The motif at error rate sites matches the consensus m⁶A target sequence. The sequence logo is for the motif enriched at sites with reduced error rate in the *vir-1* mutant. (C) Differential error rate sites are primarily found in 3′ UTRs. Bar plot showing the number of differential error rate sites per kb for different genic feature types of 48,149 protein coding transcript loci in the nuclear genome of the Araport11 reference. Upstream and downstream regions were defined as 200 nt regions upstream of the annotated transcription start sites and downstream of the annotated transcription termination sites, respectively. (D) Differential error rate sites and miCLIP peaks are similarly distributed within the 3′ UTR, without accumulation at the stop codon. Metagene plot centred on stop codons from 48,149 protein coding transcript loci, showing the frequency of nanopore DRS error sites (blue) and miCLIP peaks (orange). (E) The locations of differential error rate sites are in good agreement with the locations of sites identified by miCLIP. Histogram showing the distribution of distances to the nearest miCLIP peak for each site of reduced error. Most error sites (66%) are within 5 nt of a miCLIP peak. (F) Nanopore DRS differential error site analysis and miCLIP identify m⁶A sites in the 3′ UTR of *PRPL34* RNA. Blue, miclip 5′ end coverage; purple, nanopore DRS differential error sites; black, RNA isoform from Araport11 annotation. Expanded region: blue, miCLIP coverage; purple, nanopore DRS differential error sites;

*Figure 5 continued on next page*

*Figure 5 continued*

black, motif scores using the m⁶A consensus position weight matrix (*Figure 5B*). A higher positive score denotes a higher likelihood of a match to the consensus sequence.
The online version of this article includes the following source data and figure supplement(s) for figure 5:

**Source data 1.** *vir-1* vs VIRc differential error sites identified by nanopore DRS – *Figure 5A*.
**Source data 2.** Motifs detected under *vir-1* vs VIRc differential error sites - *Figure 5B*.
**Source data 3.** m⁶A sites detected by miCLIP of Col-0 samples - *Figure 5D*.
**Figure supplement 1.** Identification of VIR-dependent m⁶A transcriptome-wide.
**Figure supplement 1—source data 1.** MALAT1 differential error rates identified by nanopore DRS – *Figure 5—figure supplement 1A*.
**Figure supplement 1—source data 2.** m⁶A:A ratios of Col-0, *vir-1,* and VIRc lines quantified using liquid chromatography – mass spectroscopy – *Figure 5—figure supplement 1B*.

reduction in protein-coding gene expression in *vir-1* (using either nanopore DRS or Illumina RNAseq data), corresponding to transcripts that were methylated in the VIR-complemented line (*Figure 6A*, *Figure 6—figure supplement 1A*). These findings are consistent with the recent discovery that m⁶A predominantly protects Arabidopsis mRNAs from endonucleolytic cleavage (*Anderson et al., 2018*). Therefore, although the m⁶A writer complex comprises conserved components that target a conserved consensus sequence and distribution of m⁶A is enriched in the last exon, it appears that this modification predominantly promotes mRNA decay in human cells (*Wang et al., 2014*), but mRNA stability in Arabidopsis (*Anderson et al., 2018*).

The changes in gene expression in *vir-1* were wide ranging (*Figure 6—figure supplement 1B,C, D*). For example, we found that the abundance of mRNAs encoding components of the interlocking transcriptional feedback loops that comprise the Arabidopsis circadian oscillator (*McClung, 2011*), such as *CCA1*, were altered in *vir-1* (*Figure 6—figure supplement 1B,C*). This distinction was associated with a biological consequence in that the *vir-1* mutant had a lengthened clock period (*Figure 6B,C*). Notably, m⁶A is also required for normal circadian rhythms in mammalian cells (*Fustin et al., 2013*). We detected m⁶A at mRNAs encoding the clock components *PRR7, GI* and *LNK1/2* in both the nanopore DRS and miCLIP data (*Figure 6—figure supplement 1B*). We also detected shifts in the poly(A) tail length distributions of mRNAs transcribed from genes previously shown to be under circadian control (Supplementary dataset 17). At *CAB1* mRNAs, for example, we detected poly(A) tail length estimates that peaked at approximately 20, 40 and 60 nt (*Figure 6D*). *vir-1* mutants had reduced abundance of *CAB1* mRNAs with 20 and 40 nt poly(A) tails, and an increased abundance of poly(A) tails of 60 nt in length (*Figure 6D*). Therefore, nanopore DRS may uncover the circadian control of poly(A) tail length, as previously reported for specific Arabidopsis genes (*Lidder et al., 2005*). An output of the Arabidopsis circadian clock is the control of flowering time, and we found that not only were photoperiod pathway components differentially expressed but so too were other flowering time genes (*Supplementary file 7*). Notably, *FLOWERING LOCUS C* (*FLC*) expression was reduced by more than 40-fold compared with the wild type (*Figure 6—figure supplement 1D*). Consequently, the proper control of circadian rhythms, flowering time and the regulatory module at *FLC* ultimately requires the m⁶A writer complex component, VIR.

## Defective m⁶A is associated with disrupted RNA 3′ end formation

In addition to measuring RNA expression, we examined the impact of m⁶A loss on pre-mRNA processing. Detectable disruptions to splicing in *vir-1* were modest. For example, using the DEX-Seq software tool (*Anders et al., 2012*) for analysis of annotated splice sites, we found only weak effect-size changes to cassette exons, retained introns or alternative donor/acceptor sites compared with the VIR-complemented line in our Illumina RNAseq data (*Figure 7—figure supplement 1A*). In contrast, a clear defect in RNA 3′ end formation in *vir-1* was apparent. Using a Kolmogorov–Smirnov test, we identified 3,579 genes (FDR < 0.05, absolute change in position of > 13 nt) with an altered nanopore DRS 3′ position profile in the *vir-1* mutant compared with the VIR-complemented line (*Figure 7A*). Of these, 3,008 displayed a shift to usage of more proximal poly(A) sites in *vir-1*: 59% of these genes also contain m⁶A sites detectable by nanopore DRS (compared with 32% of all expressed genes, $p < 1 \times 10^{-16}$; 61% were detectable by miCLIP compared with 35% of all expressed genes, $p < 1 \times 10^{-16}$) and correspond to locations of increased cleavage downstream of

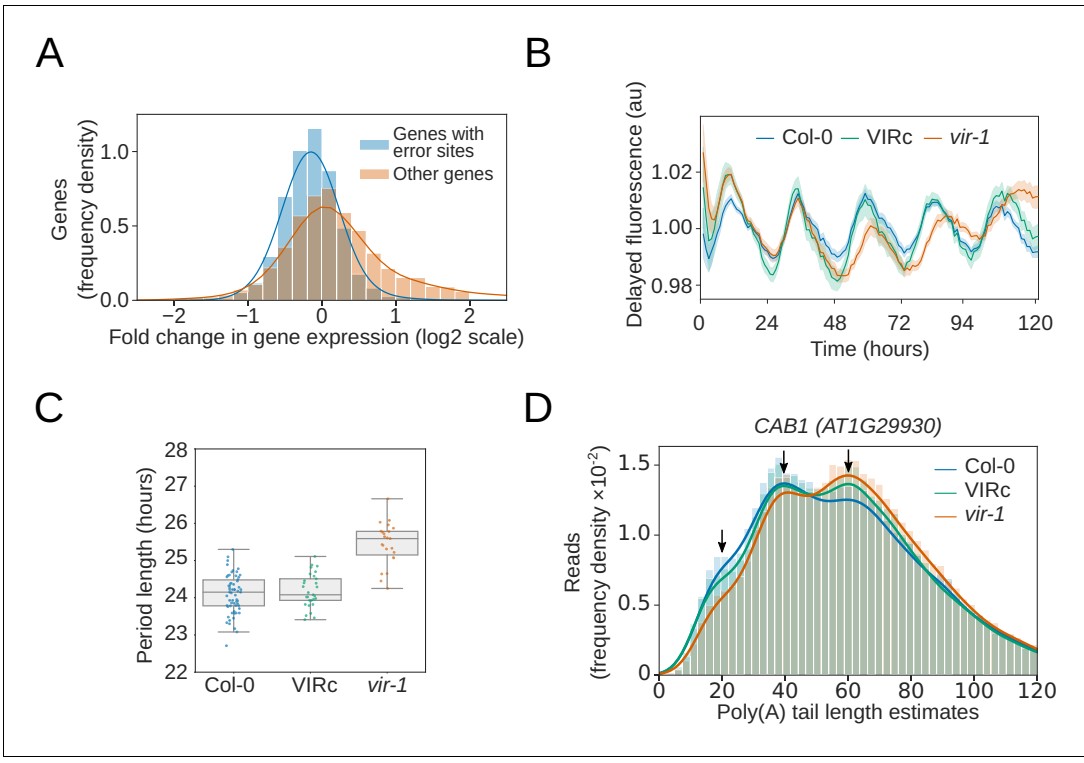

**Figure 6.** Reduction in m$^6$A RNA modification leads to disruption of the circadian clock. (**A**) Genes with differential error rate sites have lower detectable RNA levels. Histogram showing the log$_2$ fold change in protein coding gene expression based on counts from nanopore DRS reads in the *vir-1* mutant compared to the VIR-complemented line. Blue, genes with differential error rate sites (n = 5083 genes); orange, genes without differential error rate sites (n = 14,675 genes). (**B**) The circadian period is lengthened in the *vir-1* mutant. Mean delayed fluorescence measurements in arbitrary units: blue, Col-0 (n = 61 technical reps); green, the VIR-complemented line (VIRc; n = 29 technical reps); orange, *vir-1* (n = 24 technical reps). Shaded areas show bootstrapped 95% confidence intervals for the mean. (**C**) Boxplot showing the period lengths calculated from delayed fluorescence measurements shown in (**b**): blue, Col-0; green, VIRc; orange, *vir-1* (orange). (**D**) Poly(A) tail length is altered in the *vir-1* mutant. Histogram showing the poly(A) tail length distribution of *CAB1 (AT1G29930)*: blue, Col-0 (n = 40,841 reads); green, VIRc (n = 65,810 reads); orange, *vir-1* (n = 68,068 reads). Arrows indicate phased peaks of poly(A) length at approximately 20, 40 and 60 nt. *vir-1* distribution is significantly different from both Col-0 and VIRc, according to the Kolmogorov–Smirnov test (p < 1 × 10$^{-16}$, p < 1 × 10$^{-16}$ respectively).

The online version of this article includes the following source data and figure supplement(s) for figure 6:

**Source data 1.** Differential gene expression results for *vir-1* vs VIRc, using counts derived from nanopore DRS – *Figure 6A*.

**Source data 2.** Delayed fluorescence and circadian period length for Col-0, *vir-1* and VIRc – *Figure 6B*.

**Source data 3.** Differential poly(A) tail length estimate results for *vir-1* vs VIRc – *Figure 6D*.

**Source data 4.** Poly(A) tail length estimates for *CAB1* for reads from Col-0, *vir-1* and VIRc – *Figure 6D*.

**Figure supplement 1.** Changes in the gene expression of circadian clock components in the *vir-1* mutant.

**Figure supplement 1—source data 1.** Differential gene expression results for *vir-1* vs VIRc, using counts derived from Illumina RNAseq – *Figure 6—figure supplement 1A*.

**Figure supplement 1—source data 2.** qPCR expression of *CCA1* and *FLC* in Col-0, *vir-1* and VIRc – *Figure 6—figure supplement 1C*.

m$^6$A sites in the *vir-1* mutant (*Figure 7B*). A total of 571 genes showed increased transcriptional readthrough beyond the 3′ end in *vir-1* (*Figure 7A*): 72% of these loci also contained nanopore-mapped m$^6$A sites (p < 1 × 10$^{-16}$; 71% were detectable by miCLIP, p < 1 × 10$^{-16}$).

In order to examine the impact of m$^6$A on 3′ end formation in an orthogonal manner, we analysed our parallel Illumina RNAseq datasets using the DaPars algorithm which estimates shifts in patterns of alternative polyadenylation (*Xia et al., 2014*). DaPars-mediated analysis also revealed a clear

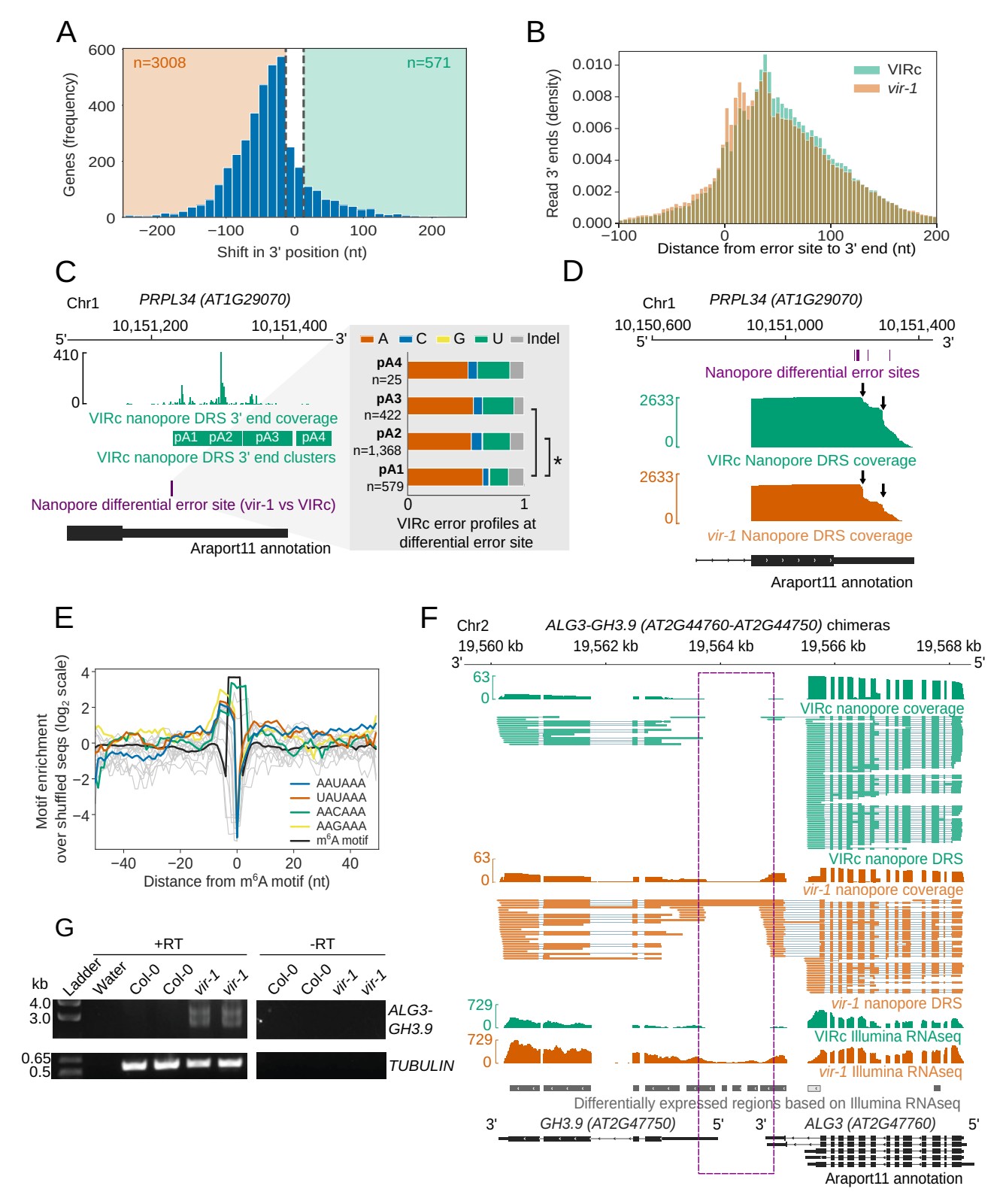

**Figure 7.** RNA methylation is closely linked to poly(A) site selection. (**A**) Global shift to proximal 3′ end usage observed in the *vir-1* mutant compared with the VIRc line. Histogram showing the distance in nucleotides between the most reduced and most increased 3′ end positions for genes in which the 3′ end profile is altered in *vir-1* (detected with the Kolmogorov–Smirnov test, FDR < 0.05). A threshold of 13 nt was used to detect changes in the 3′ end position. (**B**) A shift to the use of more proximal poly(A) sites is observed in m6A containing transcripts in the *vir-1* mutant. Histogram showing

*Figure 7 continued on next page*

*Figure 7 continued*

distance from the error site (n = 17,491 error sites) to upstream and downstream 3′ ends: orange, *vir-1*; green, VIRc. (C) Poly(A) site choice is phased with m⁶A at *PRPL34 (AT1G29070)*. *PRPL34* has four distinct poly(A) site clusters identified from VIRc reads. Analysis of the error profiles of these clusters at the closest upstream m⁶A motif indicates that proximally polyadenylated isoforms are less likely to be methylated. Green, VIRc 3′ coverage and poly (A) clusters; purple, nanopore differential error site tested; black, Araport11 annotation. Zoomed view shows stacked barplots of VIRc error profiles for the tested nanopore differential error site. Pairs of clusters with significantly different profiles (G-test, FDR < 0.05) are indicated with asterisks. (D) *PRPL34*, which is methylated in at least two positions in the 3′UTR (*Figure 5F*), also displays an increase in alternative polyadenylation at a proximal poly(A) site in the *vir-1* mutant. The proximal poly(A) site is approximately 17 nt downstream of the closest methylated position. Purple, nanopore differential error sites; green, VIRc coverage; orange, *vir-1* coverage; black, Araport11 annotation. (E) RNA methylation sites overlap with canonical and degenerate poly(A) site motifs. Line plot showing the enrichment of poly(A) site motifs (with an edit distance of one) around m⁶A motifs detected de novo under nanopore differential error sites: blue, the canonical PAS motif AAUAAA; grey, other similar motifs (excluding AAAAAA); orange, the UAUAAA motif; green, the AACAAA motif; yellow, the AAGAAA motif; black, the m⁶A motif (includes all sequences shown in *Figure 5—figure supplement 1E*). (F) Readthough events and chimeric RNAs are detected in *vir-1*. Green, nanopore DRS and Illumina RNAseq data for the VIRc line; orange, nanopore DRS and Illumina RNAseq data for the *vir-1* mutant; grey and white, differentially expressed regions between *vir-1* and VIRc detected using Illumina RNAseq data with DERfinder that are upregulated and downregulated, respectively; black, Araport11 annotation. Intergenic readthrough regions are highlighted by the purple dashed rectangle. (G) *ALG3-GH3.9* chimeric RNAs transcribed in the *vir-1* mutant. Chimeric readthrough RNAs were validated using RT-PCR. RT-PCR products were separated on agarose gel and stained with SYBR Safe. RT+ and RT-, cDNA produced in the presence and absence of reverse transcriptase, respectively; ladder, DNA size ladder; Water, no template control. *TUBULIN* was used as a control.

The online version of this article includes the following source data and figure supplement(s) for figure 7:

**Source data 1.** Differential poly(A) site choice results for *vir-1* vs VIRc, derived from nanopore 3′ ends – *Figure 7A*.
**Source data 2.** *vir-1* vs VIRc differentially expressed regions from Illumina RNAseq - *Figure 7F*.
**Source data 3.** Differential chimeric read results for *vir-1* vs VIRc - *Figure 7F*.
**Figure supplement 1.** Changes in RNA 3′ end formation in the *vir-1* mutant.
**Figure supplement 1—source data 1.** Differential exon usage for *vir-1* vs VIRc, derived from Illumina RNAseq - *Figure 7—figure supplement 1A*.
**Figure supplement 1—source data 2.** Differential poly(A) site choice results for *vir-1* vs VIRc, derived from Illumina RNAseq data – *Figure 7—figure supplement 1B*.

directional shift to proximal poly(A) site selection at thousands of loci in the *vir-1* background, with 3408 genes displaying a decrease in distal poly(A) site usage, compared to only 166 genes displaying an increase in distal poly(A) site usage (*Figure 7—figure supplement 1B*). Consequently, DaPars analysis is consistent with our nanopore DRS study, indicating that the predominant disruption in 3′ end formation in the *vir-1* mutant is a shift to proximal poly(A) site selection.

To address the relationship between m⁶A and 3′ end formation further, we asked if differential error sites were phased (associated) with specific poly(A) sites. We clustered reads from VIRc samples by their 3′ position using kernel density estimation. We then compared the error profiles of reads from different 3′ clusters using a G-test to identify whether 3′ end positions were associated with m⁶A levels (only positions which had differential error in the *vir-1* vs VIRc comparison were tested). We identified 327 error sites which were phased with 3′ end position, overlapping 207 annotated genes. 149 (72%) of these also display a statistically detectable change in 3′ end position in the *vir-1* mutant compared to VIRc. For example, at *PRPL34 (AT1G29070)*, selection of the distal poly(A) sites pA2 and pA3 is associated with increased mismatches to the reference base (presumably resulting from higher levels of m⁶A) at an upstream error site, compared to the proximal poly (A) site pA1 (*Figure 7C*). Consistent with this, there is a quantitative shift to selection of the proximal poly(A) site pA1 in the *vir-1* mutant (*Figure 7D*). These findings indicate that changes in 3′ end position in the *vir-1* mutant may result directly from the loss of m⁶A.

In the *vir-1* mutant, there is an increase in the relative abundance of 3′ end positions in the region approximately 0-20 nt downstream of differential error sites (*Figure 7B*). Notably, the distribution of the canonical poly(A) signal AAUAAA peaks 19 nt upstream of the cleavage site in Arabidopsis mRNA (related hexamers, differing at only a single position, show similar distribution patterns) (*Sherstnev et al., 2012*). We therefore compared the distribution of the poly(A) signal AAUAAA (and related hexamers differing at only a single position, except AAAAAA) with the motifs underlying differential error sites. This analysis reveals a close overlap of the peak of AAUAAA-like poly(A) signals with the m⁶A motif (*Figure 7E*). This finding may reflect the relative similarity of the most common Arabidopsis m⁶A motifs (such as AAACU and AAACA) to the poly(A) signal, and indicates that adenosine residues directly adjacent to, or within, the poly(A) signal itself may be methylated. Since loss of m⁶A activates the selection of upstream poly(A) signals, these findings are consistent

with m$^6$A at (or in the vicinity of) AAUAAA inhibiting recognition of the poly(A) signal for subsequent cleavage and polyadenylation.

We next investigated those events where downstream readthrough was detected in *vir-1*. The impacts of altered 3′ processing can be complex and have the potential to change the relative abundance of transcripts processed from the same gene but with different coding potential. For example, we detected increased readthrough of an intronic cleavage site in the Symplekin-like gene *TANG1* (*AT1G27595*; *Figure 7—figure supplement 1C*) and increased readthrough and cryptic splicing at *ALG3* (*AT2G47760*) that also results in chimeric RNA formation with the downstream gene *GH3.9* (*AT2G4G7750*; *Figure 7F*). The existence of the *ALG3-GH3.9* chimeric RNAs was supported by Illumina RNAseq (*Figure 7F*) and confirmed by RT-PCR, cloning and sequencing (*Figure 7G*). We detected 523 loci with increased levels of chimeric RNAs in *vir-1* resulting from unterminated transcription proceeding into downstream genes on the same strand (Supplementary dataset 24). Chimeric RNAs were recently detected in mutants affecting other components of the Arabidopsis m$^6$A writer complex, MTA and FIP37 (*Pontier et al., 2019*). However, only 33% of upstream genes forming the chimeric RNAs had detectable m$^6$A sites in the VIR-complemented line with restored VIR activity. Consequently, these findings might be explained either by an m$^6$A-independent role for VIR (or the writer complex) in 3′ end formation or an indirect effect on factors required for 3′ processing. m$^6$A independent roles for the human m$^6$A methyltransferases METTL3 (*Lin et al., 2016*) and METTL16 (*Pendleton et al., 2017*) have been found previously, and a role for the writer complex in controlling Arabidopsis RNA processing independent of m$^6$A cannot be overlooked (*Pendleton et al., 2017*).

Overall, we conclude that loss of *VIRILIZER*-dependent m$^6$A disrupts Arabidopsis mRNA 3′ formation. Although this includes readthrough at poly(A) sites with the formation of chimeric RNAs, the predominant impact is loss of suppression of proximal poly(A) sites.

## Discussion

We used nanopore DRS to reveal the Arabidopsis transcriptome. Using orthogonal datasets as validation in each case, we could demonstrate the utility of nanopore DRS in revealing full-length mRNAs, mapping 5′ cap position, the position and estimated length of the poly(A) tail, patterns of alternative splicing, sites of internal RNA cleavage, and m$^6$A modification of RNA. As a result, we could not only confirm previously established complexity in Arabidopsis RNA processing, but uncover novel events too. We found that spurious antisense signal was rare or absent, as were internal priming artefacts that complicate interpretation of authentic antisense RNAs and poly(A) sites respectively. Improvements in the utility of nanopore DRS likely depend upon more accurate basecalling, refined splitting of signal, increased read-depth and the development of software for splice-aware alignment of long sequencing reads.

The key technological question we addressed in this study, was whether nanopore DRS could be used to map m$^6$A transcriptome-wide. We show that a simple and transparent statistical test of basecalling error rate can identify m$^6$A. Available basecalling software depend upon recurrent neural networks trained with signal produced from sequencing RNA where modified bases are unlabelled and present in relatively low abundance. As a result, basecallers cannot accurately assign m$^6$A. Since 5 RNA bases occupy a nanopore at any one time, m$^6$A can affect the accuracy of basecalling of surrounding nucleotides. Formally then, an error-rate based approach has an approximate resolution of 9 nt. However, the methylated base can be interpreted using a position weight matrix of the consensus motif associated with error sites. Nanopore DRS technology and basecalling software continues to develop. In our experience, so long as the basecallers are not trained to directly identify m$^6$A, and sequencing and analysis is performed on a common iteration of the nanopore DRS platform (the same generation of sequencing kit, pore and basecalling software), then an error rate approach is still able to map m$^6$A. Since this is a statistical test, it depends upon an aggregation of reads and does not map m$^6$A directly in single molecules. Consequently, the power to detect m$^6$A will depend upon read depth.

Until recently, high throughput m$^6$A mapping has depended on antibodies that recognise this modification. However, antibody-based approaches such as MeRIP/m$^6$A-seq and miCLIP are limited in some respects. For example, they cannot readily define stoichiometry, do not necessarily detect m$^6$A only, have unclear sensitivity, and the fragmentation of RNA required for library preparation

may result in the underrepresentation of sites close to RNA 3' ends. As a result, antibody-independent approaches to mapping m⁶A have been sought (*Garcia-Campos et al., 2019*; *Liu et al., 2019*; *Meyer, 2019*). The enzyme MaZF cleaves RNA at ACA in a methylation-dependent manner and is used in MAZTER-Seq to give single base resolution and insight into m⁶A stoichiometry (*Garcia-Campos et al., 2019*). Since MAZTER-Seq is limited to the detection of m⁶As in an ACA context (and spatial constraints) the majority of m⁶A sites in human or yeast cells cannot be detected (*Garcia-Campos et al., 2019*). Another recently developed nanopore DRS-based approach to map m⁶A, called EpiNano, also has context-specific limitations (*Liu et al., 2019*); nanopore DRS of *in vitro* transcribed RNAs was used to train a support vector machine model to detect m⁶A (*Liu et al., 2019*). However, EpiNano cannot currently call m⁶A in a *k*-mer with more than one adenosine, and hence 92% of m⁶A sites we identify in Arabidopsis are undetectable with this approach (*Figure 5—figure supplement 1E*) (*Liu et al., 2019*). Since nanopore DRS is a relatively recent development in RNA sequencing, we anticipate that alternative approaches will refine our ability to identify RNA modifications in single molecule reads (*Lorenz et al., 2019*).

The key biological question we set out to address, was whether an unbiased transcriptome-wide sequencing approach (that mapped m⁶A in the context of full-length mRNAs) could reveal the functional impacts of m⁶A on Arabidopsis gene expression. We found that disrupting m⁶A was associated with altered patterns of mRNA cleavage and polyadenylation. Although we detected readthrough at some poly(A) sites, the predominant effect was the activation of proximal poly(A) sites in the absence of m⁶A, indicating that m⁶A can suppress proximal poly(A) site selection. We found that the distribution of m⁶A sites and the canonical poly(A) signal AAUAAA (and related hexamers with one mismatch position) closely overlapped, indicating that adenosines within the AAUAAA signal itself could be methylated. In mammals, recognition of AAUAAA involves direct binding by the CPSF30 and WDR33 proteins of the cleavage and polyadenylation machinery (*Chan et al., 2014*; *Schönemann et al., 2014*). The Arabidopsis homologs (CPSF30 and FY respectively) show conservation of amino acid residues involved in recognizing AAUAAA (*Clerici et al., 2018*; *Sun et al., 2018*). In Arabidopsis, the conserved, constitutive role for CPSF30 and FY in recognising the poly(A) signal is associated with the evolution of features for specific gene regulation. For example, FY has C- terminal proline-containing motifs that mediate an interaction with the plant-specific RNA binding protein FCA, conferring upon FY a specialised function in controlling the timing of Arabidopsis flower development (*Simpson et al., 2003*). Meanwhile, Arabidopsis (like other plants and the apicomplexa) expresses a CPSF30 isoform (regulated by alternative polyadenylation in Arabidopsis) that encodes a YT521-B homology (YTH) domain with the potential to bind and read m⁶A (*Stevens et al., 2018*). We speculate that m⁶A at AAUAAA either directly inhibits binding of CPSF30 and FY to the poly(A) signal, or that binding of m⁶A by YTH domain containing proteins (possibly CPSF30-YTH itself) blocks canonical recognition by CPSF30 and FY, and hence assembly of an active cleavage and polyadenylation complex. Future experiments will dissect the mechanisms involved.

The impact of individual m⁶A sites on Arabidopsis biology is likely to be context-specific. However, it is clear that m⁶A is almost exclusively found in mRNA 3'UTRs (in steady-state analyses). Consequently, the effect of m⁶A is likely to be mediated mostly through mRNA alternative polyadenylation, stability, translatability and the formation of specific protein complexes (*Mayr, 2019*). We (and others) find that transcripts modified with m⁶A are predominantly stabilised in Arabidopsis (*Anderson et al., 2018*). An explanation for this phenomenon is that m⁶A protects against upstream endonucleolytic mRNA cleavage (*Anderson et al., 2018*). We report here that m⁶A predominantly inhibits the selection of downstream proximal cleavage and polyadenylation sites. These distinct phenotypes may reflect temporal differences in how m⁶A affects mRNA fate. During co-transcriptional processing, m⁶A may influence the site of cleavage and polyadenylation, but subsequently affect mRNA phase separation and fate in the cytoplasm (*Ries et al., 2019*). Arabidopsis has 13 YTH domain containing proteins with the potential to bind and read m⁶A (*Reichel et al., 2019*). Understanding the complexity of the roles that they play will help explain the context-specific functions of m⁶A. The combination of m⁶A mapping and RNA complexity revealed by nanopore DRS, provides an additional approach to address these questions.

## Concluding remarks

We have shown that nanopore DRS has the potential to refine multiple features of Arabidopsis genome annotation and to generate the clearest map to date of m⁶A locations, despite the genome

sequence being available since 2000 (*Kaul and Arabidopsis Genome Initiative, 2000*). Modern agriculture is dominated by a handful of intensely researched crops (*United Nations, Department of Economic and Social Affairs, Population Division, 2017*), but to diversify global food supply, enhance agricultural productivity and tackle malnutrition there is a need to focus on crops utilized in rural societies that have received little attention for crop improvement (*Chang, 2018*). Based on our experience with Arabidopsis, we anticipate that the combination of nanopore DRS and other sequencing approaches will improve genome annotation. Consistent with this, we recently applied nanopore DRS to refine the annotation of water yam (*Dioscorea alata*), an African orphan crop. Indeed, we are moving into an era where thousands of genome sequences are available and programmes such as the Earth BioGenome Project aim to sequence all eukaryotic life on Earth (*Lewin et al., 2018*). However, genome sequences provide only part of the puzzle: annotating what they encode will be essential for us to fully utilize this information.

## Materials and methods

### Plants
Plant material and growth conditions
Wild-type *A. thaliana* accession Col-0 was obtained from Nottingham Arabidopsis Stock Centre. The *vir-1* and VIR-complemented (*VIR::GFP-VIR*) lines (*Růžička et al., 2017*) were provided by K. Růžička, Brno. The *hen2–2* (Gabi_774HO7) mutant (*Lange et al., 2014*) was provided by D. Gagliardi, Strasbourg. Seeds were sown on MS10 medium plates, stratified at 4°C for 2 days, germinated in a controlled environment at 22°C under 16 hr light/8 hr dark conditions and harvested 14 days after transfer to 22°C.

### Clock phenotype analysis
Clock phenotype experiments were performed as previously described by measuring delayed fluorescence as a circadian output (*Gould et al., 2009*). Briefly, plants were grown in 12-h light/12-h dark cycles at 22°C and 80 µmol m$^{-2}$ sec$^{-1}$ light for 9 days. Next, delayed fluorescence measurements were recorded every hour for 6 days at constant temperature (22°C) and constant light (20 µmol m$^{-2}$ sec$^{-1}$ red light and 20 µmol m$^{-2}$ sec$^{-1}$ blue light mix). Fast Fourier Transform (FFT) non-linear least-squares fitting to estimate period length was conducted using Biodare (*Moore et al., 2014*).

### RNA
RNA isolation
Total RNA was isolated using RNeasy Plant Mini kit (Qiagen) and treated with TURBO DNase (Thermo Fisher Scientific). The total RNA concentration was measured using a Qubit 1.0 Fluorometer and Qubit RNA BR Assay Kit (Thermo Fisher Scientific), and RNA quality and integrity were assessed using a NanoDrop 2000 spectrophotometer (Thermo Fisher Scientific) and Agilent 2200 TapeStation System (Agilent).

mGFP *in vitro* transcription
The mGFP coding sequence was amplified using CloneAmp HiFi PCR Premix (Clontech) and a forward primer containing the T7 promoter sequence (Merck; *Supplementary file 8*). The PCR product was purified using GeneJET Gel Extraction (Thermo Fisher Scientific) and DNA Cleanup Micro (Thermo Fisher Scientific) kits, according to the manufacturer's instructions. mGFP was *in vitro* transcribed using a mMESSAGE mMACHINE T7 ULTRA Transcription Kit (Thermo Fisher Scientific) with and without the anti-reverse cap analogue, according to the manufacturer's instructions. mGFP transcripts were treated with TURBO DNase, polyA-tailed using *Escherichia coli* poly(A) Polymerase (E-PAP) and ATP (Thermo Fisher Scientific) and recovered using a MEGAclear kit (Thermo Fisher Scientific) according to the manufacturer's instructions. The quantity of mGFP mRNAs was measured using a Qubit 1.0 Fluorometer (as described above), and the quality and integrity was checked using the NanoDrop 2000 spectrophotometer and agarose-gel electrophoresis. *In vitro* capped and non-capped mGFP mRNAs were used to prepare the libraries for DRS using nanopores.

## MALAT1 synthetic RNA

A matched set of synthetic RNAs differing only in a single m$^6$A site were chemically synthesized by Dharmacon (Horizon Discovery). The sequence corresponds to a 60-nt portion of a human lncRNA MALAT1 5'-AAGUAAUUCAAGAUCAAGAGUAAUUACCAACUUAAUGUUUUUGCAUUGG**A**CUUU-GAGUUA(A$_{50}$)−3' (**Supplementary file 8**) with a previously mapped m$^6$A (highlighted in bold) (**Liu et al., 2013**). The 2'-ACE protected synthetic RNAs were deprotected using 100 mM acetic acid adjusted to a pH of 3.4–3.8 using TEMED, according to the manufacturer's instructions. The quantity of deprotected synthetic RNAs was checked using the NanoDrop 2000 spectrophotometer. 1 μg of the MALAT1 m$^6$A and MALAT1 control synthetic RNAs were used to prepare the libraries for DRS using nanopores.

## RT-PCR and RT-qPCR

Total RNA was reverse transcribed using SuperScript III polymerase or SuperScript IV VILO Master Mix (Thermo Fisher Scientific) according to the manufacturer's protocol. For RT-PCR, reactions were performed using the Advantage 2 Polymerase Mix (Clontech) using the primers (Merck) listed in **Supplementary file 8**. PCR products were gel purified using GeneJET Gel Extraction and DNA Cleanup Micro kits (Thermo Fisher Scientific), cloned into the pGEM T-Easy vector (Promega; according to the manufacturer's instruction) and sequenced. For RT-qPCR, reactions were carried out using the SYBR Green I (Qiagen) mix with the primers (Merck) listed in **Supplementary file 8**, following manufacturer's instructions.

## Illumina RNA sequencing

### Preparation of libraries for Illumina RNA sequencing

Illumina RNA sequencing libraries from purified mRNA were prepared and sequenced by the Centre for Genomic Research at University of Liverpool using the NEBNext Ultra Directional RNA Library Prep Kit for Illumina (New England Biolabs). Paired-end sequencing with a read length of 150 bp was carried out on an Illumina HiSeq 4000. Illumina RNA libraries from ribosome-depleted RNA were prepared using the TruSeq Stranded Total RNA with Ribo-Zero Plant kit (Illumina). Paired-end sequencing with a read length of 100 bp was carried out on an Illumina HiSeq 2000 at the Genomic Sequencing Unit of the University of Dundee. ERCC RNA Spike-In mixes (Thermo Fisher Scientific) (**Jiang et al., 2011**; **Lee et al., 2016**) were included in each of the libraries using concentrations advised by the manufacturer.

### Mapping of Illumina RNA sequencing data

Reads were mapped to the TAIR10 reference genome with Araport11 reference annotation using STAR version 2.6.1 (**Dobin et al., 2013**), a maximum multimapping rate of 5, a minimum splice junction overhang of 8 nt (three nt for junctions in the Araport11 reference), a maximum of five mismatches per read and intron length boundaries of 60–10,000 nt.

### Differential gene expression analysis using Illumina RNA sequencing data

Transcript-level counts for Illumina RNA sequencing reads were estimated by pseudoalignment with Salmon version 0.11.2 (**Patro et al., 2017**). Counts were aggregated to gene level using tximport (**Soneson et al., 2016**) and differential gene expression analyses for *vir-1* mutant vs wild type and *vir-1* mutant vs the VIR-complemented line were conducted in R version 3.5 using edgeR version 3.24.3 (**Robinson et al., 2010**).

### Differentially expressed region analysis using Illumina RNA sequencing data

Mapped read pairs originating from the forward and reverse strands were separated and coverage tracks were generated using samtools version 1.9 (**Li et al., 2009**). Coverage tracks were then used as input for DERfinder version 1.16.1 (**Collado-Torres et al., 2017**). Expressed regions were identified using a minimum coverage of 10 reads, and differential expression between the *vir-1* and VIR-complemented lines was assessed using the analyseChr method with 50 permutations.

## Differential exon usage analysis using Illumina RNA sequencing data

Annotated gene models from Araport11 were divided into transcript chunks (i.e. contiguous regions within which each base is present in the same set of transcript models). Read counts for each chunk were generated using bedtools version 2.27.1 (*Quinlan and Hall, 2010*) intersect in count mode. Chunk counts were then processed using DEXseq version 1.28.3 (*Anders et al., 2012*) to identify differentially expressed chunks between *vir-1* and VIR-complemented lines, using an absolute log-fold-change threshold of 1 and an FDR threshold of 0.05. Chunks were annotated as 5′ variation if they included a start site of any transcript and as 3′ variation if they contained a termination site. Chunks representing overhangs from alternative donor or acceptor sites were also classified separately. Internal exons were subclassified as a cassette exon if they could be wholly contained within any intron.

## Differential polyadenylation analysis using DaPars and illumina RNA sequencing data

We used the DaPars (*Xia et al., 2014*) algorithm to estimate shifts in patterns of alternative polyadenylation with Illumina RNAseq data. Annotated gene models from Araport11 were flattened to give one model per gene. Exonic annotation was given priority over intronic or intergenic annotation and CDS annotation was given priority over UTR annotation. Unstranded bigwig coverage files were prepared for each Illumina RNAseq sample using bedtools genomecov 2.27.1 (*Quinlan and Hall, 2010*), as DaPars does not support stranded RNAseq data. DaPars version 0.9.1 (*Xia et al., 2014*) was then run using the flattened Araport11 annotation as a reference. Transcripts from genes which were differentially polyadenylated in the *vir-1* condition were identified using an FDR threshold of 0.05.

## Nanopore DRS

### Preparation of libraries for direct RNA sequencing using nanopores

mRNA was isolated from approximately 75 µg of total RNA using the Dynabeads mRNA purification kit (Thermo Fisher Scientific) following the manufacturer's instructions. The quality and quantity of mRNA was assessed using the NanoDrop 2000 spectrophotometer (Thermo Fisher Scientific). Nanopore libraries were prepared from 1 µg poly(A)+ RNA combined with 1 µl undiluted ERCC RNA Spike-In mix (Thermo Fisher Scientific) using the nanopore DRS Kit (SQK-RNA001, Oxford Nanopore Technologies) according to manufacturer's instructions. The poly(T) adapter was ligated to the mRNA using T4 DNA ligase (New England Biolabs) in the Quick Ligase reaction buffer (New England Biolabs) for 15 min at room temperature. First-strand cDNA was synthesized by SuperScript III Reverse Transcriptase (Thermo Fisher Scientific) using the oligo(dT) adapter. The RNA–cDNA hybrid was then purified using Agencourt RNAClean XP magnetic beads (Beckman Coulter). The sequencing adapter was ligated to the mRNA using T4 DNA ligase (New England Biolabs) in the Quick Ligase reaction buffer (New England Biolabs) for 15 min at room temperature followed by a second purification step using Agencourt beads (as described above). Libraries were loaded onto R9.4 SpotON Flow Cells (Oxford Nanopore Technologies) and sequenced using a 48 hr run time.

To incorporate cap-dependent ligation of a biotinylated 5′ adapter RNA, the following modifications were introduced into the library preparation protocol. First, 4 µg mRNA was de-phosphorylated by Calf Intestinal Alkaline Phosphatase (Thermo Fisher Scientific) and the 5′ cap was removed by Cap-Clip Acid Pyrophosphatase (Cambio) according to the manufacturer's instructions. Next, the 5′ adapter RNA (5′-CGACUGGAGCACGAGGACACUGACAUGGACUGAAGGAGUAGAAA-3′) biotinylated at the 5′ end (Integrated DNA Technologies; *Supplementary file 8*) was ligated to dephosphorylated, de-capped mRNA using T4 RNA ligase I (New England Biolabs) and mRNA was purified using Dynabeads MyOne Streptavidin C1 beads (Thermo Fisher Scientific) according to the manufacturer's instructions. mRNA was assessed for quality and quantity using the NanoDrop 2000 spectrophotometer and used for nanopore DRS library preparation (as described above).

### Processing of nanopore DRS data

Reads were basecalled with Guppy version 2.3.1 (Oxford Nanopore Technologies) using default RNA parameters and converted from RNA to DNA fastq using seqkit version 0.10.0 (*Shen et al., 2016b*). Reads were aligned to the TAIR10 *A. thaliana* genome (*Kaul and Arabidopsis Genome Initiative, 2000*) and ERCC RNA Spike-In sequences (*Jiang et al., 2011*; *Lee et al., 2016*) using

minimap2 version 2.8 (*Li, 2018*) in spliced mapping mode using a kmer size of 14 and a maximum intron size of 10,000 nt. Sequence Alignment/Map (SAM) and BAM file manipulations were performed using samtools version 1.9 (*Li et al., 2009*). Variation in mapping rates is likely caused by differences in the amount of RNA loaded onto the flowcell. For example, the cap-capture procedure, resulted in less input RNA for library preparation. This has two consequences: first, it results in lower pore occupancy during sequencing, resulting in many apparently low quality 'reads' which are actually noise signal from empty pores (*Mojarro et al., 2018*; *Pontefract et al., 2018*); second, an increased number of reads map to Enolase II, the internal standard control RNA (Oxford Nanopore Technologies control strand) added to each library. For MALAT1 synthetic RNAs, which were only 110 nt in length including a 50 nt poly(A) tail, minimap2 parameters were relaxed to allow lower mapping scores: kmer size (-k) was reduced to 10; the minimum number of minimisers per chain (-n), to 2; the minimum chaining score (-m), to 12; and the minimum dynamic programming alignment score (-s), to 8.

Proovread version 2.14.1 (*Hackl et al., 2014*) was used to correct errors in the nanopore DRS reads (*Depledge et al., 2019*). Each nanopore DRS replicate was split into 200 chunks for parallel processing. Each chunk was corrected using four samples of Illumina poly(A) RNAseq data selected randomly from the 36 Illumina files (six biological replicates sequenced across six lanes). Illumina reads 1 and 2 were merged into fragments using FLASh version 1.2.11 (*Magoč and Salzberg, 2011*). Unjoined pairs were discarded. Error correction with proovread was conducted in sampling-free mode using a minimum nanopore read length of 50 nt. Since both the Illumina RNAseq and nanopore DRS datasets were strand specific, proovread was modified to prevent opposite strand mapping between the datasets. Corrected reads were then mapped to the Araport11 reference genome using minimap2 (as described above). All figures showing gene tracks with nanopore DRS reads use error-corrected reads, visualized using the Integrative Genomics Viewer (*Robinson et al., 2011*).

## Error rate analysis using nanopore DRS data

Error rate analysis of aligned reads was conducted on ERCC RNA Spike-In mix controls using pysam version 0.15.2 (*Heger et al., 2014*) for BAM file parsing. Matches, mismatches, insertions and deletions relative to the reference were extracted from the cs tag (a more informative version of CIGAR string, output by minimap2) and normalised by the aligned length of the read. Reference bases and mismatch bases per position were also recorded and used to assess the frequency of each substitution and indel type by reference base.

## Over-splitting analysis of nanopore DRS data

To identify read pairs resulting from over-splitting of the signal originating from a single RNA molecule, the sequencing summary files produced by Guppy were parsed for sequencing time and channel identifier and then used to identify pairs of consecutively sequenced reads. Genomic locations of reads were parsed from minimap2 mappings, and consecutively sequenced reads with adjacent alignment within a genomic distance of −10 nt to 1000 nt were identified. Samples sequenced before or during May 2018 had very low levels of over-splitting (between 0.01% and 0.05% of reads) compared with those sequenced in September 2018 onwards (between 0.8% and 1.5% of reads).

## Analysis of the potential for internal priming in nanopore DRS data

To determine whether internal priming caused by the RT step can occur in nanopore data, the location of oligo(A) hexamers within Arabidopsis coding sequence (CDS) regions (Araport11) was determined and reads that terminated within a 20 nt window of each hexamer were counted. Of the 10,116 CDS oligo(A) runs, 160 (1.58%) had at least one supporting read in one Col-0 nanopore dataset. Of these, 137 were supported by only one replicate, and only four were supported by all four biological replicates. In total, 66 (41%) occurred in Araport11-annotated terminal exons, suggesting that they may be genuine sites of 3' end formation.

## Poly (A) length estimation using nanopore DRS data

Poly (A) tail length estimations were produced using nanopolish version 0.11.0 (*Workman et al., 2019*) and added as tags to bam files using pysam version 0.15.2 (*Heger et al., 2014*). Per gene

length distributions were then produced by identifying reads overlapping each gene in the Araport11 annotation by > 20% of their aligned length, and genes with significant changes in length distribution in the *vir-1* mutant compared with the VIR-complemented line were identified using a Kolmogorov–Smirnov test. *p*-values were adjusted for multiple testing using Benjamini-Hochberg correction.

## 3′ end analyses of nanopore and Helicos reads

Helicos data were prepared as previously described (*Duc et al., 2013*; *Schurch et al., 2014*). Positions with three or more supporting reads were considered to be peaks of nanopore or Helicos 3′ ends. The distance between each nanopore peak and the nearest Helicos peak was then determined. In all, 37% of nanopore peaks occurred at the same position as a Helicos peak, and the standard deviation in distance was 12.5 nt. To determine the percentage of nanopore DRS 3′ ends mapping within annotated genic features, transcripts were first flattened into a single record per gene. Exonic annotation was given priority over intronic or intergenic annotation and CDS annotation was given priority over UTR annotation. Reads were assigned to genes if they overlapped them by > 20% of their aligned length, and the annotated feature type of the 3′ end position was determined. Counts were generated both for all reads and for unique positions per gene.

## Isoform collapsing of nanopore DRS data

Error-corrected full-length alignments were collapsed into clusters of reads with identical sets of introns. These clusters were then subdivided by 3′ end location by using a Gaussian kernel with sigma of 100 to find local minima between read ends, which were used as cut points to separate clusters. The read with the longest aligned length in each cluster was used as the representative in the figure.

## Splicing analysis of nanopore DRS and Illumina RNAseq data

Splice junction locations, their flanking sequences and the read counts supporting them were extracted from Illumina RNA sequencing, nanopore DRS and nanopore error-corrected DRS reads using pysam version 0.14 (*Heger et al., 2014*), as well as from Araport11 (*Cheng et al., 2017*) and AtRTD2 (*Zhang et al., 2017*) reference annotations. Splice junctions at the same position but on the opposite strand were counted independently. Junctions were classified by their most likely snRNP machinery using Biopython version 1.71 (*Cock et al., 2009*), with position weight matrices as previously calculated (*Sheth et al., 2006*). Position weight matrices were scored against the sequence – three nt to +10 nt of the donor site and –14 nt to +3 nt of the acceptor site. Position weight matrix scores greater than zero indicate a match to the motif, while scores of around zero, or negative scores, indicate background frequencies or deviation from the motif. Positive scores were normalized into the range 50–100 as done previously (*Sheth et al., 2006*). Junctions with U12 donor scores of > 75 and U12 acceptor scores of > 65 were classified as U12 junctions, while junctions with U2 donor and acceptor scores of > 60 were classified as U2, as done previously (*Zhang et al., 2017*). Junctions were further categorized as canonical or non-canonical based on the presence or absence, respectively, of GT/AG intron border sequences. For isoform analysis, linked splices from the same read were extracted from full-length nanopore error-corrected reads and counted to create unique sets of splice junctions. Intronless reads were not counted. UpSet plots were generated in Python 3.6 using the package upsetplot (*Nothman, 2018*).

## Validation of novel splice sites

To validate novel splice junctions detected in nanopore DRS, five splice sites out of the 20 most highly expressed splice sites were selected for further validation; three of the five selected splice sites were successfully amplified in RT-PCR followed by Sanger sequencing (described above).

## Identification of potential novel NMD targets

Error corrected reads from Col-0 samples were filtered to retain only those that overlapped both annotated starts and termination codons from the Araport11 reference annotation (*Cheng et al., 2017*). These reads were then assessed to identify those where the length of the open reading frame was not divisible by three, indicating a splicing event causing a frameshift compared to the

reference. These reads were then further filtered to remove examples without splicing support from Col-0 poly(A) Illumina RNAseq datasets. Novel examples were selected by filtering out splice patterns already present in the Araport11 or AtRTD2 reference annotations.

## 5′ adapter detection analyses using nanopore DRS data

To produce positive and negative examples of 5′ adapter-containing sequences, 5′ soft-clipped regions were extracted from aligned reads for the Col-0 replicate one datasets (with and without adapter ligation) using pysam (*Heger et al., 2014*). These soft-clipped sequences were then searched for the presence of the adapter sequence using BLASTN version 2.7.1 (*Camacho et al., 2009*). Two rules were initially applied to filter BLASTN results: a match of 10 nt or more to the 44 nt adapter, and an E value of < 100. Reads from the adapter-containing dataset that failed one or both criteria were used as negative training examples. A final rule requiring the match to the adapter sequence to occur directly adjacent to the aligned read was also applied. Reads from the adapter-containing dataset that passed all three rules were used as the positive training set. When comparing the ratio of positive to negative examples between datasets containing the adapter and those generated from the same tissue but without the adapter, we found that these three rules gave a signal-to-noise ratio of > 5000 (*Supplementary file 3*).

In all, 72,083 positive and 123,739 negative training examples from Col-0 tissue replicate one were collected to train the neural network. We then estimated the amount of raw signal from the 5′ end of the squiggle that was required on average to capture the 5′ adapter. To do this, we used nanopolish eventalign version 0.11.0 (*Loman et al., 2015*) to identify the interval in the raw read corresponding to the mRNA alignment to the reference in the positive examples of 5′ adapter-containing sequences. Since the adapter can be identified immediately adjacent to the alignment in sequence space for these reads, the signal after the event alignment should correspond to the signal originating from the adapter. The median length of these signals was 1441 points, and 96% of the signals were < 3000 points. Therefore, we used a window size of 3000 to make predictions.

The model was trained in Python 3.6 using Keras version 2.2.4 with the Tensorflow version 1.10.0 backend (*Chollet, 2018*; *Abadi, 2016*). A ResNet-style architecture was used (*Kaiming He et al., 2015*), composed of eight residual blocks containing two convolutional layers of kernel size five and a shortcut convolution with kernel size 1. Down-sampling using maximum pooling layers with a stride of 2 was used between each residual block. A penultimate densely connected layer of size 16 was used, with training dropout of 0.5. Input signals were standardized by median absolute deviation scaling across the whole read before the final 3000 points were taken, and the negative samples were augmented by addition of random internal signals from reads and pure Gaussian, multi-Gaussian and perlin noise signals (*Wick et al., 2018*). The whole dataset was also augmented on the fly during training by the addition of Gaussian noise with a standard deviation of 0.1. Models were trained for a maximum of 100 epochs (batch size of 128, 100 batches per epoch, positive and negative examples sampled in a 1:1 ratio) using the RMSprop optimizer with an initial learning rate of 0.001, which was reduced by a factor of 10 after three epochs with no reduction in validation loss. Early stopping was used after five epochs with no reduction in validation loss. We found that a number of negative training examples from the ends of reads, but not from internal positions, were likely to be incorrectly labelled by the BLASTN method, because the model predicted them to contain adapters. We therefore filtered these to clean the training data, before repeating the training process. Model performance was evaluated using five-fold cross validation and by testing on independently generated datasets from Col-0 replicate 2, produced with and without the adapter ligation protocol (*Figure 3—figure supplement 1B,C*) (*Chollet, 2018*; *Abadi, 2016*).

To evaluate the reduction in 3′ bias of adapter-ligated datasets, we used Araport11 exon annotations to produce per base coverage for each gene in the Col-0 replicate two dataset. Coverage was generated separately for reads predicted to contain adapters and for those predicted not to contain adapters. Leading zeros at the extreme 5′ and 3′ ends of genes were assumed to be caused by mis-annotation of UTRs and so were trimmed. The quartile coefficient of variation (interquartile range/median) was then used as a robust measure of variation in coverage across each gene.

Two orthogonal approaches were used to validate the 5′ ends of adapter-ligated reads. First, full-length cDNA clone sequences were downloaded from RIKEN RAFL (Arabidopsis full-length cDNA clones) (*Seki et al., 2002*). These were mapped with minimap2 (*Li, 2018*) in spliced mode. The

distance from each nanopore alignment 5′ end to the nearest RIKEN RAFL alignment (*Seki et al., 2002*) 5′ end was calculated using bedtools (*Quinlan and Hall, 2010*). The amount of 5′ end sequence that is rescued when 5′ adapters are used was estimated by identifying the largest peak in 5′ end locations per gene in the absence of adapter, and then measuring how this peak shifted using reads predicted to contain adapters. Second, Illumina RNAseq reads derived from 5′ tags using nanoPARE were downloaded from GSE112869 (*Schon et al., 2018*). nanoPARE reads and matched control RNAseq reads were mapped to TAIR10 with Araport11 reference annotation (*Cheng et al., 2017*) using STAR version 2.6.1 (*Dobin et al., 2013*), a maximum multimapping rate of 5, a minimum splice junction overhang of 8 nt (three nt for junctions in the Araport11 reference), a maximum of five mismatches per read, and intron length boundaries of 60–10,000 nt. Mapped reads were filtered into those representing 'capped' and 'uncapped' 5′ ends by the presence or absence of soft-clipped untemplated Gs at the 5′ end, which are added by reverse transcriptase in a cap dependent manner (*Schon et al., 2018*). 5′ coverage and matched RNAseq 5′ coverage tracks were generated using bedtools version 2.27.1 (*Quinlan and Hall, 2010*). nanoPARE capped read peaks were called at single nucleotide resolution with Piranha version 1.2.1 (*Uren et al., 2012*) and relaxed *p*-value thresholds of 0.5. Reproducible peaks across pairwise combinations of the three replicates were identified by irreproducible discovery rate (IDR) analysis using Python package idr version 2.0.3 with an IDR threshold of 0.05 (*Li et al., 2011*). The final set of peaks was identified by pooling the three replicates, re-analysing using Piranha, ranking the peaks by FDR and selecting the top N peaks, where N is the smallest number of reproducible peaks discovered by pairwise comparisons of the three replicates. This yielded 47,986 unique single nucleotide resolution nanoPARE peaks. The distance from each nanopore alignment 5′ end to the nearest nanoPARE peak was calculated using bedtools (*Quinlan and Hall, 2010*).

## Identification of miRNA cleavage products in nanopore DRS data

Nanopore miRNA cleavage product identification was adapted from the EndCut method used previously (*Schon et al., 2018*). Known miRNA and siRNA sequences from *Arabidopsis thaliana* were downloaded from https://github.com/Gregor-Mendel-Institute/nanoPARE. miRNA target sites were predicted using GSTAr (https://github.com/MikeAxtell/GSTAr). miRNAs were also shuffled 1000 times and targets were predicted for shuffled sequences to model random chance predictions. *P* values for Allen scores of miRNA-target interactions were produced by permutation test, using the Allen scores of targets predicted from randomly shuffled versions of the miRNA as a null distribution. *P* values for the enrichment of nanopore DRS 5′ ends at the target cleavage sites were also produced by permutation test, comparing the log ratio of coverage in a 20 nt window upstream and downstream of each predicted cleavage site for an miRNA, to the coverage upstream and downstream of predicted cleavage sites of the shuffled miRNA. A 20 nt window was used instead of measuring 5′ ends precisely at the cleavage site, because as we demonstrate in this study, approximately 11 nt are lost from the 5′ end of RNA sequencing reads in standard nanopore DRS. Predictions which fell within 5′UTRs or across splice junctions were filtered out to reduce false positives. The *P* values derived from nanopore coverage and Allen scores were combined using the fisher method and corrected for multiple testing using the Benjamini-Hochberg method.

## Differential error site analysis using nanopore DRS data

To detect sites of *VIRILIZER*-dependent m$^6$A RNA modifications, we developed scripts to test changes in per base error profiles of aligned reads. Pileup columns for each position with coverage of > 10 reads were generated using pysam (*Heger et al., 2014*) and reads in each column were categorized as either A, C, G, U or indel. The relative proportions of each category were counted. Counts from replicates of the same experimental condition were aggregated and a $2 \times 5$ contingency table was produced for each base comparing *vir-1* and VIR-complemented lines. A G-test was performed to identify bases with significantly altered error profiles. For bases with a *p*-value of < 0.05, G-tests for homogeneity between replicates of the same condition were then performed. Bases where the sum of the G statistic for homogeneity tests was greater than the G statistic for the *vir-1* and VIR-complemented line comparison were filtered. Multiple testing correction was carried out using the Benjamini-Hochberg method, and an FDR threshold of 0.05 was used. The log$_2$ fold change in mismatch to match ratio (compared with the reference base) between the *vir-1* and VIR-

complemented lines was calculated using Haldane correction for zero counts. Bases that had a log fold change of > 1 were considered to have a reduced error rate in the *vir-1* mutant.

To identify motifs enriched at sites with a reduced error rate, reduced error rate sites were increased in size by five nt in each direction and overlapping sites were merged using bedtools version 2.27.1 (*Quinlan and Hall, 2010*). Sequences corresponding to these sites were extracted from the TAIR10 reference and over-represented motifs were detected in the sequences using MEME version 5.0.2 (*Bailey et al., 2009*), run in zero or one occurrence mode with a motif size range of 5–7 and a minimum requirement of 100 sequence matches. The presence of these motifs at error sites was then detected using FIMO version 5.0.2 (*Grant et al., 2011*). A relaxed FDR threshold of 0.1 was used with FIMO to capture more degenerate motifs matching the $m^6A$ consensus.

To determine the effect of sequencing depth on the ability to call nanopore differential error sites, we performed a bootstrapped sub-sampling analysis. We conducted this analysis using MALAT1 synthetic RNA sequencing reads, and four differential error sites derived from three genes in the Arabidopsis data. 500 bootstrapped subsamples, at depths between 10 and 250 reads per condition, were generated for each gene and each base in the gene was then tested using differential error rate analysis as described above. Detection was considered successful if any of the differential error sites at the $m^6A$ motif which could be detected with all reads, was significant in the subsampled reads (FDR threshold 0.05). For the MALAT1 dataset, we also performed analyses on the effect of $m^6A$ stoichiometry on the ability to detect sites using differential error. For these analyses, reads from the $m^6A$ positive sample were replaced with reads from the $m^6A$ negative sample at rates of 0%, 25%, 50% and 75%, to simulate methylation levels of 100%, 75%, 50% and 25% respectively.

## Analysis of phasing of nanopore DRS error sites with poly(A) site choice

To identify error sites which were phased with 3′ ends, we used error sites identified in the *vir-1* vs VIRc comparison. We identified reads overlapping each error site in the VIRc samples using pysam version 0.15.2 (*Heger et al., 2014*), and clustered them by 3′ end position using local minima detected after gaussian smoothing with a sigma of 13 nt. Clusters containing fewer than 10 reads in total were discarded. Error sites where there was only one cluster of reads after filtering were not assessed further. Basecall profiles for each 3′ cluster at the overlapping error site were then produced and tabulated into an N × 5 contingency table, where N is the number of 3′ clusters, and five is the basecall type, that is A, C, G, U, or indel. Error sites with differences in basecall profile in different 3′ clusters were detected using a G-test. *P* values were corrected for multiple testing using the Benjamini-Hochberg method. For *PRPL34 (AT1G29070)*, which is featured as an example in *Figure 7D*, pairwise comparisons of poly(A) cluster error profiles were performed with G-tests and corrected for multiple testing using the Benjamini-Hochberg method.

## Differential gene expression analysis using nanopore DRS data

Gene level counts were produced for each nanopore DRS replicate using featureCounts version 1.6.3 (*Liao et al., 2013*) in long-read mode with strand-specific counting. Differential expression analysis between the *vir-1* and VIR-complemented lines was then performed in R version 3.5 using edgeR version 3.24.3 (*Robinson et al., 2010*).

## Identification of alternative 3′ end positions and chimeric RNA using nanopore DRS data

Genes with differential 3′ end usage were identified by producing 3′ profiles of reads which overlapped with each annotated gene locus by > 20%. These profiles were then compared between the *vir-1* and VIR-complemented lines using a Kolmogorov–Smirnov test to identify changes. Multiple testing correction was performed using the Benjamini-Hochberg method. To approximately identify the direction and distance of the change, the normalized single base level histograms of the VIR-complemented line profile was subtracted from that of the mutant profile, and the minimum and maximum points in the difference profile were identified. These represent the sites of most reduced and increased relative usage, respectively. Results were filtered for an FDR of < 0.05 and absolute change of site > 13 nt (the measured error range of nanopore DRS 3′ end alignment). The presence of $m^6A$ modifications at genes with differential 3′ end usage was assessed using bedtools intersect

(*Quinlan and Hall, 2010*), and significant enrichment of m⁶A at these genes was tested using a hypergeometric test (using all expressed genes as the background population).

To identify genes with a significant increase in chimeras in the *vir-1* mutant, we used Araport11 annotation (*Cheng et al., 2017*) to identify reads that overlapped with multiple adjacent gene loci by more than 20% of their aligned length (i.e. chimeric reads) and those originating from a single locus (i.e. non-chimeric reads). To reduce false positives caused by reads mapping across tandem duplicated loci, reads mapping to genes annotated in PTGBase (*Yu et al., 2015*) were filtered out. Chimeric reads were considered to originate from the most upstream gene with which they overlapped. We pooled reads from replicates for each experimental condition and used 50 bootstrapped samples (75% of the total data without replacement) to estimate the ratio of chimeric to non-chimeric reads at each gene in each condition. Haldane correction for zero counts was applied. The distributions of chimeric to non-chimeric ratios in the *vir-1* and VIR-complemented lines were tested using a Kolmogorov–Smirnov test to detect loci with altered chimera production. All possible pairwise combinations of VIR-complemented and *vir-1* bootstraps were then compared to produce a distribution of estimates of change in the chimeric to non-chimeric ratio in the *vir-1* mutant. Loci that had more than one chimeric read in *vir-1*, demonstrated at least a two-fold increase in the chimeric read ratio in > 50% of bootstrap comparisons and were significantly changed at a FDR of < 0.05 were considered to be sites of increased chimeric RNA formation in the *vir-1* mutant.

## Poly(A) site motif enrichment at nanopore DRS differential error motifs

To assess the relationship between m⁶A and the poly(A) signal motif, we counted the occurrence of the sequence AAUAAA and related sequences (with an edit distance of one, excluding AAAAAA) in a 100 nt window centred around m⁶A motifs detected de novo underneath nanopore error sites. To assess enrichment at these sites, we also shuffled the sequences derived from these windows 1000 times, maintaining the dinucleotide content using ushuffle version 1.1.0 (*Jiang et al., 2008*), and counted the occurrences of the same motifs each time. The log enrichment score was then derived from the ratio of the observed count over the median expected count from shuffled sequences.

## miCLIP

### Preparation of miCLIP libraries

Total RNA for miCLIP was isolated from 7.5 mg of 14 day old Arabidopsis Col-0 seedlings as previously described (*Quesada et al., 2003*). mRNA was isolated from ~1 mg total RNA using oligo(dT) and streptavidin paramagnetic beads (PolyATtract mRNA Isolation Systems, Promega) according to the manufacturer's instructions. miCLIP was carried out as previously described (*Grozhik et al., 2017*) in three biological replicates, using 15 µg mRNA per each replicate, and an antibody against N6-methyladenosine (#202 003 Synaptic Systems), with minor modifications. No-antibody controls were processed throughout the experiment. RNA-antibody complexes were separated by 4–12% Bis-Tris gel electrophoresis at 180 V for 50 min and transferred to nitrocellulose membranes (Protran BA85 0.45 µm, GE Healthcare) at 30 V for 60 min. Membranes were then exposed to Medical X-Ray Film Blue (Agfa) at −80˚C overnight. Reverse transcription was carried out using barcoded RT primers: RT41, RT48, RT49 and RT50 (Integrated DNA Technologies; *Supplementary file 8*). After reverse transcription, cDNA fraction corresponding to 70–200 nt was gel purified after 6% TBE-urea gel electrophoresis (Thermo Fisher Scientific). After the final PCR step, all libraries were pooled together, purified using Agencourt Ampure XP magnetic beads (Beckman Coulter) and eluted in nuclease-free water. Paired-end sequencing with a read length of 100 bp was carried out on an Illumina MiSeq v2 at Edinburgh Genomics, University of Edinburgh. Input sample libraries were prepared using NEBNext Ultra Directional RNA Library Prep Kit for Illumina (New England Biolabs) and sequenced on an Illumina HiSeq2000 at the Tayside Centre for Genomics Analysis, University of Dundee, with a pair-end read length of 75 bp.

### Processing of miCLIP sequencing data

miCLIP data were assessed for quality using FastQC version 0.11.8 (*Andrews, 2010*) and MultiQC version 1.7 (*Ewels et al., 2016*). Only the forward read was used for analysis because the miCLIP site is located at the 5′ position of the forward read. 3′ adapter and poly(A) sequences were trimmed using cutadapt version 1.18 (*Martin, 2011*) and unique molecular identifiers were extracted from the

5′ end of the reads using UMI-tools version 0.5.5 (*Smith et al., 2017*). Immunoprecipitation and no-antibody controls were demultiplexed and multiplexing barcodes were trimmed using seqkit version 0.10.0 (*Shen et al., 2016b*). Reads were mapped to the TAIR10 reference genome with Araport11 reference annotation (*Cheng et al., 2017*) using STAR version 2.6.1 (*Dobin et al., 2013*), a maximum multimapping rate of 5, a minimum splice junction overhang of 8 nt (three nt for junctions in the Araport11 reference), a maximum of five mismatches per read, and intron length boundaries of 60–10,000 nt. SAM and BAM file manipulations were performed using samtools version 1.9 (*Li et al., 2009*). Removal of PCR duplicates was then performed using UMI-tools in directional mode (*Smith et al., 2017*). miCLIP 5′ coverage and matched input 5′ coverage tracks were generated using bedtools version 2.27.1 (*Quinlan and Hall, 2010*) and these were used to call miCLIP peaks at single nucleotide resolution with Piranha version 1.2.1 (*Uren et al., 2012*) and relaxed $p$-value thresholds of 0.5. Reproducible peaks across pairwise combinations of the three replicates were identified by irreproducible discovery rate (IDR) analysis using Python package idr version 2.0.3 with an IDR threshold of 0.05 (*Li et al., 2011*). miCLIP replicate three was found to correlate less well, with fewer reproducible sites (Spearman's rho 0.54 for replicates 1 vs. 2, 93,046 reproducible sites, Spearman's rho 0.42 for replicates 1 vs. 3, 12,777 reproducible sites, Spearman's rho 0.47 for replicates 2 vs. 3, 19,948 reproducible sites). We therefore chose to use the reproducible sites detected in the comparison of replicates 1 and two as the final set of miCLIP sites for downstream analysis.

## m⁶A liquid chromatography–mass spectroscopy analysis

m⁶A content analysis using liquid chromatography–mass spectroscopy (LC-MS) was performed as previously described (*Huang et al., 2018*). Chromatography was carried out by the FingerPrints Proteomics facility, University of Dundee.

## Code availability

All scripts, pipelines and notebooks used for this study are available on GitHub at https://github.com/bartongroup/Simpson_Barton_Nanopore_1 (*Parker and Schurch, 2019*; copy archived at https://github.com/elifesciences-publications/Simpson_Barton_Nanopore_1).

## Acknowledgements

This work was funded by BBSRC grants BB/J00247X/1, BB/M004155/1, BB/M010066/1, the University of Dundee Global Challenges Research Fund and the European Union's Horizon 2020 research and innovation programme under Marie Skłodowska-Curie grant agreement No. 799300. The m⁶A LC-MS analysis was carried out by Abdel Atrih of the FingerPrints Proteomics Facility, University of Dundee, which is supported by a Wellcome Trust Technology Platform award (097945/B/11/Z). We thank Kasper Rasmussen and Csaba Hornyik for their comments on the manuscript.

## Additional information

### Funding

| Funder | Grant reference number | Author |
|---|---|---|
| Biotechnology and Biological Sciences Research Council | BB/J00247X/1 | Geoffrey J Barton<br>Gordon G Simpson |
| Biotechnology and Biological Sciences Research Council | BB/M004155/1 | Geoffrey J Barton<br>Gordon G Simpson |
| Biotechnology and Biological Sciences Research Council | BB/M010066/1 | Geoffrey J Barton<br>Gordon G Simpson |
| H2020 Marie Skłodowska-Curie Actions | 799300 | Katarzyna Knop |
| University of Dundee | GCRF Challenge Fund | Geoffrey J Barton<br>Gordon G Simpson |
| Wellcome | 097945/B/11/Z | Geoffrey J Barton<br>Gordon G Simpson |

The funders had no role in study design, data collection and interpretation, or the decision to submit the work for publication.

### Author contributions
Matthew T Parker, Conceptualization, Data curation, Software, Formal analysis, Validation, Investigation, Visualization, Methodology, Writing—original draft, Writing—review and editing; Katarzyna Knop, Conceptualization, Funding acquisition, Validation, Investigation, Visualization, Methodology, Writing—original draft, Project administration, Writing—review and editing; Anna V Sherwood, Conceptualization, Validation, Investigation, Visualization, Methodology, Writing—original draft, Project administration, Writing—review and editing; Nicholas J Schurch, Conceptualization, Data curation, Software, Formal analysis, Supervision, Validation, Visualization, Methodology, Writing—original draft, Project administration, Writing—review and editing; Katarzyna Mackinnon, Validation, Investigation, Visualization, Writing—review and editing; Peter D Gould, Formal analysis, Investigation, Writing—review and editing; Anthony JW Hall, Resources, Supervision, Writing—review and editing; Geoffrey J Barton, Conceptualization, Resources, Supervision, Funding acquisition, Project administration, Writing—review and editing; Gordon G Simpson, Conceptualization, Resources, Supervision, Funding acquisition, Writing—original draft, Project administration, Writing—review and editing

### Author ORCIDs
Matthew T Parker ⬤ https://orcid.org/0000-0002-0891-8495
Katarzyna Knop ⬤ https://orcid.org/0000-0002-2636-9450
Nicholas J Schurch ⬤ https://orcid.org/0000-0001-9068-9654
Geoffrey J Barton ⬤ https://orcid.org/0000-0002-9014-5355
Gordon G Simpson ⬤ https://orcid.org/0000-0001-6744-5889

### Decision letter and Author response
Decision letter https://doi.org/10.7554/eLife.49658.sa1
Author response https://doi.org/10.7554/eLife.49658.sa2

## Additional files

### Supplementary files
• Supplementary file 1. Properties of the nanopore DRS sequencing data. Dataset statistics for all nanopore DRS sequencing runs conducted. Datasets are sorted by the date of the sequencing run. All data was collected using a MinION with R9.4 flow cell and SQK-RNA001 library kit. Increases in mapping and over-splitting rate that occur in samples collected after September 2018 are therefore likely to have resulted from changes in the MinKNOW software. Other variations in mapping rates are likely caused by differences in the amount of RNA loaded onto the flowcell. Depletion of degraded RNA during 5' adapter RNA ligation, for example, results in smaller libraries for nanopore sequencing. This causes lower pore occupancy during sequencing and results in many reads with low median quality scores, which are actually noise signal from empty pores (*Mojarro et al., 2018*; *Pontefract et al., 2018*). Related to *Figures 1–7*.

• Supplementary file 2. Alignment statistics for Illumina RNAseq dataset. Related to *Figures 1–7*.

• Supplementary file 3. Adapter detection using BLASTN rules approach. The number of reads with adapters were detected in two biological replicates (Tables A and B respectively) of Col-0 sequenced with and without adapter ligation protocol. Rules are applied cumulatively (i.e. row one shows reads that pass the first rule, row two shows read alignments that pass the first and second rules, etc.). The signal-to-noise ratio shows the number of positive examples detected using rules in the adapter-ligated dataset divided by the number of false positives from the dataset collected without adapters. Related to *Figure 3*.

• Supplementary file 4. Cleavage products are detectable at predicted miRNA target sites in nanopore DRS data. Table showing the genes where we were able to detect degradation intermediates from miRNA cleavage events using nanopore DRS. Predictions in 5'UTRs and across splice junctions

have been filtered to reduce false positives. Only predictions with support from nanoPARE data are shown. Related to *Figure 3*.

• Supplementary file 5. Splice junctions supported by nanopore DRS and Illumina RNAseq. Numbers are shown for (**A**) the unique splice junction set intersections upset plot (*Figure 4B*) and (**B**) unique linked splicing events upset plot (*Figure 4C*). Shaded cells denote sets included in the intersection for that row, while unshaded cells denote sets excluded from the intersection. Rows are sorted by the size of the intersection for canonical splice junctions. Related to *Figure 4*.

• Supplementary file 6. Known *upf1* sensitive alternative splicing events which are supported by nanopore DRS read alignments. Table showing examples of NMD sensitive alternative splicing events taken from Table 1 of *Kalyna et al. (2012)*. which are supported by nanopore DRS reads. Duplicate genes and unsupported examples have been removed from the table. [Linked to *Figure 4*].

• Supplementary file 7. Flowering time gene expression. Change in gene expression of curated genes involved in flowering time in Arabidopsis, as detected using Illumina RNAseq for *vir-1* compared with the VIR-complemented line. In all, 12.2% of flowering time genes show a change in mRNA level expression in the *vir-1* mutant. Source of flowering time genes: George Coupland, Cologne: https://www.mpipz.mpg.de/14637/Arabidopsis_flowering_genes. Related to *Figure 6*.

• Supplementary file 8. Primers used in this study.

• Supplementary file 9. Key resources table.

• Transparent reporting form

## Data availability

Illumina FASTQ sequencing data and ONT FAST5 sequencing data have been deposited in ENA under accession code PRJEB32782. All source code is available on GitHub at https://github.com/bartongroup/Simpson_Barton_Nanopore_1 (copy archived at https://github.com/elifesciences-publications/Simpson_Barton_Nanopore_1). Raw source data for Delayed fluorescence, sanger sequencing, LC-MS, and qPCR is available in supporting files. Derived data used to generate figures from Nanopore and Illumina datasets is also available as supporting files.

The following dataset was generated:

| Author(s) | Year | Dataset title | Dataset URL | Database and Identifier |
|---|---|---|---|---|
| Parker M, Knop K, Sherwood A, Schurch N, Mckinnon K, Gould P, Hall A, Barton GJ, Simpson GG | 2019 | Nanopore Direct RNA Sequencing Maps the Arabidopsis m6A Epitranscriptome | https://www.ebi.ac.uk/ena/data/view/PRJEB32782 | European Nucleotide Archive, PRJEB32782 |

The following previously published datasets were used:

| Author(s) | Year | Dataset title | Dataset URL | Database and Identifier |
|---|---|---|---|---|
| Sherstnev A, Duc C, Cole C, Zacharaki V, Hornyik C, Ozsolak F, Milos PM, Barton GJ, Simpson GG | 2012 | Study of genome-wide alternative polyadenylation in A. thaliana | http://www.ebi.ac.uk/ena/data/view/ERP001018 | European Nucleotide Archive, ERP001018 |
| Seki M, Narusaka M, Kamiya A, Ishida J, Satou M, Sakurai T, Nakajima M, Enju A, Akiyama K, Oono Y, Muramatsu M, Hayashizaki Y, Kawai J, Carninci P, Itoh M, Ishii Y, Arakawa T, | 2002 | Functional annotation of a full-length Arabidopsis cDNA collection. | http://rarge.psc.riken.jp/archives/rafl/ | RIKEN, RAFL |

| Shibata K, Shinagawa A, Shinozaki K | | | | |
|---|---|---|---|---|
| Schon MA, Kellner MJ, Plotnikova A, Hofmann F, Nadine MD | 2018 | NanoPARE: parallel analysis of RNA 5' ends from low input RNA | https://www.ncbi.nlm.nih.gov/geo/query/acc.cgi?acc=GSE112869 | NCBI Gene Expression Omnibus, GSE112869 |

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
