## [Decision Letter]

**Acceptance summary:**

Studying the full transcriptome complexity and processing is key to understanding the biology of any organism. While the advent of RNA sequencing using short-read sequencing has greatly enriched our understanding of the transcriptome, challenges due to short-read sequencing include difficulties in mapping uniquely to individual isoforms, and assembling full transcripts, especially in repeat regions. In addition, the need to convert RNA into cDNA libraries for sequencing can result in artifacts in transcript annotation due to template switching of reverse transcriptase enzymes, and cDNA sequencing limited in its ability to detect RNA modifications. Using nanopore direct RNA sequencing, the authors updated the annotation of the Arabidopsis transcriptome by identifying new antisense transcripts, new splicing patterns, 3' end usage, polyA tail lengths and full length transcripts using 5'cap capture. In addition, using a mutant of the m^6^A writer in Arabidopsis (*vir-1* mutant), and *vir-1* mutant reconstituted with *vir-1*, the authors used increased error rates in direct RNA sequencing to detect m^6^A modifications transcriptome-wide. The authors showed that m^6^A modifications impact RNA stability and circadian cycles are associated with 3' end formation in arabidopsis. The paper is very informative in the utility and limitations of using direct RNA sequencing to interrogate transcriptomes and the strategies described in the manuscript is widely applicable to not only Arabidopsis but also to any biologist interested in their transcriptome of choice.

**Decision letter after peer review:**

Thank you for submitting your article "Nanopore direct RNA sequencing maps an Arabidopsis N6 methyladenosine epitranscriptome" for consideration by *eLife*. Your article has been reviewed by three peer reviewers, and the evaluation has been overseen by a Reviewing Editor and Christian Hardtke as the Senior Editor. The reviewers have opted to remain anonymous.

The reviewers have discussed the reviews with one another and the Reviewing Editor has drafted this decision to help you prepare a revised submission.

Summary:

This is a timely manuscript to demonstrate the utility of direct RNA sequencing in discovering transcriptomic features in the model organism *Arabidopsis thaliana*. In addition to studying the primary sequences of the transcriptome, including 5' ends, splicing and polyA+ tail length, the authors also utilized direct RNA sequencing to identify m^6^A modifications in *vir-1* mutants and in *vir-1* mutants with restored VIR activity. They found that m^6^A modifications is associated with 3' end processing in *Arabidopsis thaliana*. Overall the manuscript is well written and interesting.

We have suggestions to improve the manuscript to increase the impact of the novelty and utility of using direct RNA sequencing, as well as the technical strength of the manuscript by adding controls and performing validations to determine the accuracy of the new signals that they find.

Essential revisions:

1) For polyA tail length determination- could the authors show accuracy of their polyA tail determination on a set of standards with known poly(A) tail lengths?

2) 75% of the author's adapter ligated library failed to align to the transcriptome. The authors should look into the failed reads to explain why this is so.

3) For transcripts with new 5' ends identified by 5'end capture, the authors should validate of the new 5'ends using 5'end RACE.

4) A set of RNAs with known m^6^A sites should be generated to determine the accuracy and sensitivity of the author's method in detecting m^6^A to other available softwares, such as Tombo from Nanopore, as well as other published methods including (https://www.biorxiv.org/content/10.1101/525741v1). The authors should also comment on the robustness of their method with regards to new software changes.

5) The authors should show data on the correlation between sequencing depth and the ability to detect m^6^A modifications. They should also do deeper analysis to look at the nucleotide composition of the called m^6^A modifications that they capture, as well as whether there is potential bias towards capturing modifications on specific transcripts.

6) Direct RNA sequencing benefits from the aspect that it is long read, single molecule sequencing. As such, there is the possibility of "phasing" modifications that cannot be performed using short-read sequencing. The authors should show some examples of RNAs with more or less modifications than expected and attempt to associate that with RNA processing to demonstrate the utility of direct RNA sequencing.

Reviewer #1:

The possibility to perform RNA sequencing in its native form, without the requirement of cDNA synthesis, represents a major significant advance in RNA biology. Within this context, in this manuscript Matthew Parker and colleagues applied the Oxford Nanopore direct RNA sequencing to characterize the profiling of RNA transcripts in the model plant *Arabidopsis thaliana*. In the first part of the manuscript, the aim of this research is to: 1) prove that the Nanopore direct RNA sequencing has the ability to detect long, complex mRNAs under an acceptable error rate and to resolve a great complexity of splicing events; 2) prove that the over-splitting and spurious antisense reads generally occurs at low frequency in Nanopore direct RNA sequencing; 3) prove that Nanopore direct RNA sequencing can be used in estimation of poly-A tail length; 4) prove that Nanopore direct RNA sequencing using Cap-dependent capturing is effective in detecting authentic mRNA 5′ ends by using a convolutional neural network classifier of raw signals. In the second part, the authors present a direct RNA sequencing-based approach to detection of m^6^A methylation in data produced from both the *vir-1* mutants defective in m^6^A and the *vir-1* mutants restoring VIR activity. Furthermore, taking advantage of long read sequencing, the authors found that VIR activity is associated with the maintenance of 3' end formation of mRNAs. Overall, this manuscript describes a scientifically sound investigation of an important scientific area, that of epitranscriptomics. This manuscript will no doubt be a valuable resource to the growing field of direct RNA sequencing. I recommend this manuscript for publication in *eLife*.

I have some specific points detailed below.

1) In the supplementary Table 1, only 25% and 27% of sequencings reads in the 5' adapter ligation library could be mapped to the TAIR10 genome. The authors should check the sequencing reads to explain in detail the reason for the failed alignment in a large majority of sequencing reads.

2) Subsection “Differential error site analysis reveals the m^6^A epitranscriptome”, 17,491 sites with a more than two-fold higher error rate in the VIR-complemented line with restored m^6^A. Are these error sites exclusively located in adenine positions? A nucleotide composition summary is required to improve clarity.

3) Subsection “Spurious antisense reads are rare or absent in nanopore DRS”, the absence of reads mapping to antisense to RCA suggests that spurious antisense is rare or absent from Nanopore direct RNA sequencing data. Is it a generally accepted that the genomic loci of RCA do not generate antisense transcripts? If so, this information should be included.

4) The webpage (https://github.com/bartongroup/Simpson_Barton_Nanopore_1) which is used to deposit scripts and pipelines on GitHub is not accessible. The code availability will represent a valuable addition to the Nanopore RNA sequencing community and can be used a guide for direct RNA methylation analysis.

5) This is a technical paper to assess the performance of Nanopore direct RNA sequencing in a pioneer manner. Since m^6^A RNA modification detection and full-length RNA transcript sequencing using Cap-dependent capturing constitutes the major contribution of this paper, the recent advances relating to these topics should be included in the Introduction section.

Liu H, et al. Accurate detection of m^6^A RNA modifications in native RNA sequences. bioRxiv 2019:525741.

Jiang F, et al. Long-read direct RNA sequencing by 5'-Cap capturing reveals the impact of Piwi on the widespread exonization of transposable elements in locusts. RNA Biol 2019:1-10.

Reviewer #2:

In this manuscript, Parker et al. demonstrated several interesting applications of nanopore direct RNA sequencing (DRS) to advance the current understanding of Arabidopsis transcriptome. They have assessed the primary performance factors of DRS that are related to the basecalling error profile, poly(A) length distribution, full-length splicing profile, and the reliability of antisense reads. Also, they have developed two new techniques for DRS to find the 5′-end positions of capped RNAs and the m^6^A -modified positions in mRNAs. Finally, they applied an error-based m^6^A -detection method to find the association between the m^6^A modifications of 3′ UTR and RNA 3′-end processing in *Arabidopsis thaliana*.

This manuscript lists many different types of RNA processing and modification that can be detected by DRS. This is a nice demonstration of the potential applications and strengths of DRS. But most parts do not connect to each other, and the conceptual novelty is limited. I have some concerns that will need to be addressed before publication.

1) This technique is conceptually similar to the work by Liu et al. (https://www.biorxiv.org/content/10.1101/525741v1). Please clarify the similarities/differences between Liu et al. and the method described in this manuscript.

2) Base-calling error profiles can vary depending on the basecaller version, the model in the basecaller, the processing/filtering parameters used for running the basecaller, the pore protein version, and the motor protein version. Recently, there was a huge upgrade for the basecalling model (high-accuracy model, a.k.a. flip-flop basecaller) introduced in Guppy 3.2. Moreover, according to the Oxford Nanopore Technologies (ONT), a faster motor protein and the new R10 pore will be introduced to their DRS kits very soon. The authors need to clarify and inform the readers that the error profile presented here may change depending on the basecaller software, a model, a pore, and a motor protein.

The method described in this manuscript can work in the specific combinations that the authors used, but it does not guarantee its performance in the other settings. Considering the change in the flagship basecaller, Guppy, with higher accuracy after the version used in the manuscript, the authors need to check if this method works with comparable accuracy with the newer Guppy.

3) In addition, Tombo from the ONT has a mode called "level_sample_compare" that enables the modified base detection by comparing control and experiment groups. Can you discuss a bit about the benefits and drawbacks of your method in comparison with Tombo?

4) Subsection “Nanopore DRS confirms sites of RNA 3′ end formation and estimates poly(A) tail length”: The standard nanopore DRS library preparation uses the double-stranded RNA-DNA ligation assisted by an oligo(dT) splint. It inevitably introduces a substantial underrepresentation of short poly(A) tails. Even 10 nt oligo(dT) splint often results in a strong bias for > 20 nt poly(A) tails against the shorter tails. Moreover, short poly(A) tails often carry additional U tails which interfere with the ligation to the adapter. In addition, the means of the single-read "poly(A) length measurements" may not be a suitable summarization to estimate the means of "poly(A) lengths." Nanopolish gives the best approximations of poly(A) lengths at a single-read level. However, as the poly(A) dwell time roughly follows a gamma distribution with a long tail, the mean of the best approximations at a single-read level systematically overestimates the mean of poly(A) length.

5) Subsection “Cap-dependent 5′ RNA detection by nanopore DRS”: The authors used RNA ligase 1 to mark the 5′ ends of RNAs. Due to the substrate specificity of this enzyme, circularized mRNAs and mRNA-mRNA concatamers might have been produced. Supplementary Table 1 shows that the libraries using the standard protocol yielded ~90% of mappable reads while the libraries with cap-dependent ligation yielded only ~25%. Please describe what the other 75% are. Also, it would be helpful if the sequence information of the 5′ adapter were presented in the manuscript.

6) Subsection “Differential error site analysis reveals the m^6^A epitranscriptome”: How many of the miCLIP peaks could be detected with nanopore DRS? The authors only show the analyses using the population-level detection of m^6^A -modified sites. A real benefit of nanopore DRS lies in the associative analyses at a single-read level. Is it possible to use the DRS method to call m^6^A -modified sites within single molecules and analyze to see if a modified RNA has different polyadenylation status or 3′ end position from those in an unmodified RNA?

I feel that the current discoveries related to m^6^A modification in this manuscript can be better done using miCLIP than nanopore DRS.

Reviewer #3:

In this manuscript, "Nanopore direct RNA sequencing maps an Arabidopsis N6 methyladenosine epitranscriptome" the authors use Nanopore direct-read RNA sequencing (DRS) to characterize the Arabidopsis transcriptome, including features that can't easily be determined by short read sequencing, such as cap-associated transcription start sites, splicing events, poly(A) site choice and poly(A) tail length. The authors also use this approach to map sites of m^6^A modification and use a *vir-1* mutant strain to validate m^6^A sites. The identification of modified transcripts allowed the authors to identify a role of m^6^A in regulating length of circadian period. Overall the paper is well written and is easy to follow.

The authors use cap-dependent ligation to enrich for capped mRNAs, reducing the 3' end bias observed in DRS. It would be of interest to discuss if the transcripts with early 5' ends result from technical artifacts (for example, degradation during RNA preparation) or represent transcripts present in the cell that lack a cap structure that is compatible with the ligation protocol. For example, the authors demonstrate that nanopore DRS can identify rare anti-sense transcripts. Could transcripts with early 5' ends, which are selected against in the cap dependent libraries, represent rare transcripts or degradation intermediary products?

Additionally, the authors leverage DRS to expand the set of annotated transcripts and splicing isoforms. Can the authors use this data to describe new endogenous targets of NMD, poison exons, or find examples of proteins with new functional domains?

One feature of the transcriptome the authors focus on is the localization and effect of m^6^A modification on RNA metabolism. It would be informative if in addition to the ERCC controls, a set of RNAs with known m^6^A sites were also included.

One aspect of m^6^A biology that can't easily be studied with current methodologies is the stoichiometry of the modification at each position in each transcript. Can the authors comment at all on stoichiometry of m^6^A from DRS? Furthermore, can the authors comment on how many reads per transcript are necessary to detect a modification? Is there a bias towards more abundant transcripts? In mammalian cells it has been shown that modified RNAs tend to have shorter 3' UTRs (Molinie et al., 2016)(PMID: 27376769). Can this observation be tested in DRS, and if so, does the same phenomena occur in Arabidopsis? Information on stoichiometry can be validated for ACA sites with transcript specific assays using the MazF nuclease.

Lastly, can the authors comment on the ability of detecting modifications other than m^6^A through DRS?

---

## [Author Response]

Essential revisions:1) For polyA tail length determination- could the authors show accuracy of their polyA tail determination on a set of standards with known poly(A) tail lengths?

We used the software tool nanopolish (polyA), developed by Jared Simpson’s lab, to estimate poly(A) tail length. Details on the development of the tool are incorporated within the Workman et al., pre-print (https://www.biorxiv.org/content/10.1101/459529v1). Importantly, in this context, the tool was tested on a set of RNA substrates with poly(A) tails of known lengths. Workman et al. state:

“These datasets consisted of ionic current traces for synthetic *S. cerevisiaeS. cerevisiae* enolase transcripts appended with 3′ poly(A) tails of 10, 15, 30, 60, 80 or 100 nucleotides”

The analysis of this set of standards with known poly(A) tail lengths in the Workman et al. study reports:

“Median estimates fell within 4 nucleotides of the expected tail length for the 10-to-80 poly(A) datasets; for the 100 nt dataset, the median estimate was 109 nt.”

To examine this question independently, using our own nanopore DRS data, we asked how well nanopolish (polyA) performed in identifying the poly(A) tail of the ERCC spike-ins included in our experiments. According to the manufacturers, most of the spike-ins have a terminal poly(A) stretch of 24 nt, with all spike-in poly(A) tails in the range 20-26 nt. For example, ERCC- 00002 has an expected poly(A) tail length of 24 nt, and the median nanopolish polyA estimate was 23.3 nt (95% confidence intervals [CIs] [5.7, 102], n = 1,485). ERCC-00074 also has an expected poly(A) tail length of 24 nt, and the median nanopore polyA estimate was 27.4 nt (95% CIs [8.9, 118], n = 1,439). We conclude that the median estimated poly(A) tail length by nanopolish (polyA) is similarly accurate to that reported by Workman et al., but with a broad margin of error.

Text relating to these findings is now included in Results section and these data are now included as Figure 2—figure supplement 1A.

2) 75% of the author's adapter ligated library failed to align to the transcriptome. The authors should look into the failed reads to explain why this is so.

This finding is explained by the relatively low amount of input RNA used in the corresponding sequencing runs. In our standard nanopore DRS experiments we used 1,000ng of poly(A+) RNA for input libraries loaded onto the MinION flowcell. However, the recovery of RNA after cap capture was less than 250ng. This reduction in sample RNA has two consequences:

First, relatively low amounts of input RNA reduces the ratio of the sample RNA to ONT’s internal standard RNA (yeast Enolase II [ENOII]). As a result, an order of magnitude more reads (11-20%) map to Enolase II in the adapter ligated samples compared to our standard nanopore DRS samples (Author response image 1).

Second, low input RNA results in low pore occupancy and “noise reads” consisting of signal from empty pores^1,2^. Consistent with this, the remaining unmapped reads are very short (75% of unmapped reads are 100 nt or less in length) and of low quality (98% of unmapped reads which are 100 nt or less in length have a median per-base quality score of 7, compared to only 1% of mapped reads) as illustrated in Author response image 2. Consequently, these unmappable “reads” do not equate to signals from RNA molecules but from empty pore “noise”^1,2^.

**Author response image 1. respfig1:** Low amounts of input RNA increase the number of Enolase II reads and short noise reads. Read length distribution of mapped (blue) and unmapped (orange) reads originating from a standard DRS experiment (left panel) and an adapter ligation experiment (right panel).

**Author response image 2. respfig2:** Noise reads have low quality scores. Boxplot showing distribution of median per-read quality scores for mapped and unmapped reads from an adapter ligation DRS experiment.

Text has been inserted into the computational analysis methods section and the legend to Table 1, of the revised manuscript to explain this phenomenon.

3) For transcripts with new 5' ends identified by 5'end capture, the authors should validate of the new 5'ends using 5'end RACE.

Our 5’ cap capture method is essentially a cap-dependent 5’ RACE experiment: We ligate a biotinylated synthetic RNA oligonucleotide matching the oligo sequence used in the RLM GeneRacer kit designed for cap-dependent 5’ RACE (Thermo Fisher Scientific). Consequently, stronger validation of the new 5’ ends we identify could derive from a more orthogonal approach such as the Illumina-based high throughput 5’ end capture method called nanoPARE published at the end of 2018 by Michael Nodine’s group using Arabidopsis RNA3.

NanoPARE is able to capture 5’ ends using reverse transcription of RNA with a template switching oligonucleotide (TSO), followed by tagmentation and PCR using TSO and transposase specific primers. PCR products are then sequenced using the IIlumina platform. Template switching using the TSO occurs most often at the 5’ end of the mRNA regardless of whether there is an m7G cap or not. Reads originating from the 5’ end of capped mRNAs can be distinguished from reads from uncapped 5’ ends and internal template switching by the addition of untemplated guanosine at the 5’ end of the read.

To address whether novel 5’ ends identified by nanopore can be validated with another approach, we downloaded and reanalysed nanoPARE data, focussing on those reads with untemplated guanosine that originate from capped mRNAs. The 5’ ends identified by nanopore DRS at *RCA (AT2G39730*), which agree with both our Illumina RNAseq coverage and the RIKEN RAFL clones, also agree well with the coverage from nanoPARE data (Figure 3E in the revised version of manuscript). Furthermore, the alternative transcriptional start sites identified at *AT1G17050* and *AT5G18650* using nanopore DRS are also clearly visible in the nanoPARE data (Figure 3—figure supplement 1D in the revised version of manuscript).

We reported in our original submission that 60% of nanopore 5’ ends sequenced using the cap capture protocol were within 13 nt of a transcription start site (TSS) identified using RIKEN RAFL cDNA clones (Figure 3C, left panel, in the revised version of manuscript). We can now also report that 75% of nanopore DRS 5' ends sequenced using cap capture are within 5 nt of a TSS detected using single base resolution peak-calling from nanoPARE reads, and 82% of nanopore DRS 5’ ends are within 13 nt of a TSS (Figure 3C, right panel, in the revised version of manuscript). This closer agreement is presumably due to the higher depth of the nanoPARE compared to RIKEN RAFL dataset, leading to the greater coverage of alternate TSSs.

This information is now reported in the Results section of the manuscript and the data analysis has been incorporated into the Materials and mmethods section.

4) A set of RNAs with known m^6^A sites should be generated to determine the accuracy and sensitivity of the author's method in detecting m^6^A to other available softwares, such as Tombo from Nanopore, as well as other published methods including (https://www.biorxiv.org/content/10.1101/525741v1). The authors should also comment on the robustness of their method with regards to new software changes.

To address the first of the points raised by the reviewers, we examined whether we could detect m^6^A in a matched set of synthetic RNAs that differ only in the presence/absence of N6 methylation at a single adenosine. We selected a 60 nt portion of a human lncRNA (MALAT1) that has been reported to be modified by m^6^A, and for which base-specific validation of the modification had been obtained using the orthogonal technique, SCARLET4. The m^6^A -modified and unmodified RNAs were chemically synthesized by Dharmacon (Horizon Discovery), complete with a poly(A) tail 50 nt in length. We prepared and sequenced a library of each RNA using the nanopore DRS procedure.

Following the application of our differential error rate analysis we could successfully identify the methylated adenosine.

Text describing this approach has been added to the Results section, with an accompanying Figure 5—figure supplement 1A. In addition, details of the associated methodology involved have been added to the Materials and methods section.

The relative evaluation of our nanopore differential error analysis against other approaches detecting m^6^A is not straightforward. No comparable assessment of the accuracy or sensitivity of any of the established m^6^A mapping techniques Me-RIP/ m^6^A -Seq5, mi-CLIP6, MAZTER-Seq7 or EpiNano8 exists. This is partly because no absolute “ground truth” for m^6^A sites exists, aside from individual synthesised RNAs. However, we have addressed this question in the following ways:

Since our manuscript was reviewed, the EpiNano bioRxiv preprint has been published in *Nature Communications*^8^. A distinction from the preprint is that the subsequent publication includes an assessment of EpiNano on biological data i.e. nanopore DRS of *Saccharomyces cerevisiae* strain SK1 defective in the *S. cerevisiae* m^6^A methyltransferase, IME4. *S. cerevisiaeS. cerevisiae* mRNAs are methylated during meiosis. Hence, synchronisation of cells is required to have appropriate starting material to detect m^6^A. Data corresponding to 3 biological replicates of the “wild-type” strain (Say841 comprising a deletion of NDT80 to facilitate meiosis synchronization) and 3 biological replicates of the *ime4* mutant strain (Say966 in which both NDT80 and IME4 were deleted) were produced.

Liu et al.^8^ compared the relative performance of EpiNano against TOMBO using m^6^A sites identified by Me-RIP/ m^6^A -seq5 as the “ground truth”. However, while our manuscript was under review, another study was published in *Cell* that used an antibody-independent method called MAZTER-seq7 to map m^6^A. MAZTER-seq uses the enzyme MazF, which cleaves RNA in an ACA sequence context, unless the proximal A is methylated7. The MAZTER-seq study reported a minimal estimate of false positives and false negatives (validated in some cases using SCARLET) in *S. Cerevisiae* Me-RIP/ m^6^A -seq data^5,9^. Consequently, Me-RIP/ m^6^A -seq data does not define an absolute “ground truth”. However, MAZTER-seq does not define an absolute “ground truth” either because only a fraction of *S. Cerevisiae* m^6^A sites are in an ACA context

The nanopore DRS dataset for *S. cerevisiae*^8^ is smaller than that which we report here for Arabidopsis (3 biological replicates, 2,215,099 read alignments compared to our 4 Arabidopsis biological replicates with 10,364,150 read alignments). Consequently, this smaller dataset likely results in reduced power to detect m^6^A. Furthermore, only basecalled data appears to be publicly available and it is not certain whether data is basecalled using MinKNOW (on which basecaller versions can vary due to auto-updates) or independently using a fixed version of the basecaller. Possibly for these reasons, we detect a large number of false positives in the differential error site approach when using an FDR threshold of 0.05 and log fold change threshold of 1 during analysis of these data. We therefore filtered the data with a more stringent FDR threshold of 10-6, resulting in the detection of 2,294 differential error sites (Author response image 3). A de novode novo motif search using MEME at these sites returned the motif RGACWWT (Author response image 3), which resembles very closely the motif identified by Schwartz et al. using MeRIP/ m^6^A -seq5. Application of the identified motif to the sequences underlying error sites using FIMO (FDR < 0.1) identified 555 unique m^6^A motif sites. The most common motifs detected with nanopore DRS are shown in Author response image 3 and included GGACA (34.1%) and AGACA (22.3%). Notably, these two motifs were also the most frequently detected with the MeRIP/ m^6^A -seq approach (GGACA [29.4%], AGACA [17.3%])5.

**Author response image 3. respfig3:** Differential error rate analysis of *S. cerevisiae* SK1 WT vs ime4. (**a**) Histogram showing the log_2_ fold change in the ratio of errors to reference matches at bases with a significant change in error profile in ime4 mutant compared with wild type *S. cerevisiae* SK1. Orange and green shading indicates sites with increased and reduced errors in ime4, respectively. (**b**) Sequence logo showing the motif enriched at sites with reduced error rate in ime4. (**c**) Frequency of m^6^A motifs detected at ime4 reduced error sites, as detected by FIMO using the motif detected de novode novo by MEME and an FDR threshold of 0.1.

We conclude that the output of analysis of these *S. cerevisiae* nanopore DRS datasets with a differential error rate approach is consistent with well-established features of m^6^A in this organism.

We assessed the performance of our nanopore DRS differential error rate approach by using the nanopore DRS data published in the Liu et al. EpiNano study^8^ and Me-RIP/ m^6^A -seq sites5 and MAZTER-seq sites7 as “ground truths”. However, it is important to bear in mind the limitations of such “ground truths” and hence the interpretation of “false positives” or “false negatives”. As a result, we refer here to “ground truth" and “predicted” sites that were “supported” or “unsupported” by either the Me-RIP/ m^6^A -seq or MAZTER-seq data.

We calculate three metrics for each set of predictions compared to the ground truth: (i) precision, (ii) recall and (iii) F1 score.

*Precision* describes the fraction or percentage of predicted sites which have support in the ground truth dataset.

*Recall* describes the fraction or percentage of ground truth sites which are detected in the predicted dataset.

*The F1-score* an overall measure of accuracy and is defined as the harmonic mean of the precision and recall.

We first used Me-RIP/ m^6^A -seq data as an m^6^A “ground truth” dataset (Author response table 1). Schwartz et al ^5^ identified 1,308 Me-RIP/ m^6^A -seq peaks, of which 1,100 have coverage in the nanopore DRS data. Me-RIP/ m^6^A -seq peak intervals were increased in length to 50 nt centred around the peak summits downloaded from GEO accession GSE51583. Using the motifs predicted de-novo underneath differential error sites, 11.3% of Me-RIP/ m^6^A -seq sites were recovered with a precision of 22.4% for an F1-score of 0.15. EpiNano recovered 21.2% of Me-RIP/ m^6^A -seq sites with a precision of 2.9%, for an F1-score of 0.05. MAZTER-seq recovered 11.5% of Me-RIP/ m^6^A -seq peaks with a precision of 10%, with an F1-score of 0.11. Analysed in this way, our nanopore DRS differential error site approach performs similarly to MAZTER-seq. Although EpiNano recovered more sites, it also called many more sites (32,309) for which there is no supporting Me-RIP/ m^6^A -seq data. Indeed, using EpiNano sites with a threshold of zero (*p* >= 0.0, Author response table 1) gave an F1-score of 0.03, suggesting that much of the predictive power of EpiNano can be achieved by simply using a DRACH motif search in regions with nanopore DRS read coverage.

**Author response table 1. resptable1:** Performance of MAZTER-seq, EpiNano and Nanopore differential error rate methods using *S. cerevisiae* Me-RIP/m6A-seq peaks as the “ground truth”.

	MAZTER seq	EpiNano *p* >= 0.0	EpiNano *p* >= 0.5	EpiNano *p* >= 0.9	Nanopore error sites	Nanopore error site motifs
Total ground truth sites	1,100	1,100	1,100	1,100	1,100	1,100
Total prediction sites	1,257	32,920	8,012	169	2,294	555
Ground truth sites supported	127	489	233	14	138	124
Unsupported prediction sites	1,126	32,309	7,767	155	2,087	430
Unsupported ground truth sites	973	611	867	1,086	962	976
Precision	0.101	0.015	0.029	0.083	0.062	0.224
Recall	0.115	0.445	0.212	0.013	0.125	0.113
F1 score	0.108	0.029	0.051	0.022	0.083	0.150
Fisher *p* value	1.7E-124	6.9E-205	1.5E-112	4.9E-13	3.9E-98	3.9E-161

We next used MAZTER-seq datasets as a "ground truth” dataset (Author response table 2). Garcia-Campos et al. identified 1,341 MAZTER-seq m^6^A sites in ACA contexts^7^, of which 1,257 have coverage in the nanopore data. Because MAZTER-seq only maps m^6^A in an ACA context^7^, and EpiNano is unable to map m^6^A in 5mers with more than one A, there was no overlap between the datasets, meaning EpiNano scored a precision, recall, and F1 of zero. Me-RIP/m6A-seq peaks data recovered 10.4% of sites with a precision of 11.9%, for an F1-score of 0.11. When we performed de novo motif detection under MeRIP/m6A-seq peaks (motif identified was GGACWWT) and filtered out m^6^A sites which were not in ACA contexts, we recovered only 7.5% of MAZTER-seq sites, but with an improved precision of 26.2%, for an F1 score of 0.12. Motifs detected de novo under nanopore DRS differential error sites recovered 9.2% of MAZTER-seq sites with a precision of 33.0%, for an F1-score of 0.14, when filtered to ACA contexts. Consequently, when analysed in this way, nanopore DRS differential error site analysis performed marginally better than Me-RIP/m6A-seq (with and without motif detection) and outperforms EpiNano.

**Author response table 2. resptable2:** Performance of Me-RIP/m6A seq, EpiNano and Nanopore differential error rate methods using *S.cerevisiae* MAZTER-seq sites as the “ground truth”.

	Me-RIP seq	Me-RIP seq motifs	Me-RIP seq motifs (ACA)	EpiNano *p* >= 0.5	Nanopore error sites	Nanopore error site motifs	Nanopore error site motifs (ACA)
Total ground truth sites	1,257	1,257	1,257	1,257	1,257	1,257	1,257
Total prediction sites	1,100	669	361	8,012	2,294	555	348
Ground truth sites supported	131	95	95	0	123	116	116
Unsupported prediction sites	973	575	267	8,012	2,130	442	235
Unsupported ground truth sites	1,126	1,162	1,162	1,257	1,134	1,141	1,141
Precision	0.119	0.142	0.262	0	0.055	0.208	0.330
Recall	0.104	0.076	0.076	0	0.098	0.092	0.092
F1 score	0.111	0.099	0.117	0	0.070	0.128	0.144
Fisher *p* value	1.7E-124	2.6E-155	1.9E-183	1	8.5E-143	6.93E-211	4.26E-238

Overall, we conclude that nanopore DRS differential error analysis can perform similarly to Me- RIP/m6A-seq and MAZTER-seq, but outperforms EpiNano.

The strengths of MAZTER-seq are that single nucleotide resolution and stoichiometric data are obtained^7^. However, since MAZTER-seq only detects m^6^A in an ACA context (and with spatial resolution constraints from other ACA sites), the majority of mRNA m^6^A sites are undetectable with this approach^7^.

The limitations of EpiNano may derive from the training data used: RNAs were transcribed *in vitro* in either the presence or absence of m^6^A. As a result, not only are RNAs with methylation profiles that have not been reported in nature used in the training set, but also the most frequently occurring sequence contexts found in nature e.g. AAm^6^ACA are missing from the training data. In order to address these limitations, EpiNano does not incorporate sites with more than one A in the *k-mer*, and so like MAZTER-seq, the majority of m^6^A sites found in mRNA are undetectable using EpiNano.

When we were characterising the features of the differential error sites we identified in *S. cerevisiae* nanopore DRS data, we asked if these sites were enriched toward the 3’ end of mRNAs as has been previously reported^5^. We found that this was the case (Author response image 4). However, when we plotted the “ground truth” MeRIP/m6A-seq and MAZTER-seq sites in the same way, we found the profiles to be quite different (Author response image 4). Strikingly, MeRIP/m6A-seq data was relatively depleted in sites that map closest to RNA 3’ ends. These distinctions may be explained either by different degrees of synchronisation in meiosis in the different studies, analysis-based definition of peaks in Me- RIP/m6A-seq data or from the well-established limitations of Illumina-based fragmentation procedures providing sequence coverage at the 5’ and 3’ extremities of RNA molecules. Whatever the explanation, it reveals the limitations in the field to defining m^6^A ground truths in biological data and hence caveats to the analysis that we could perform.

**Author response image 4. respfig4:** Smoothed distribution of m^6^A sites detected by different approaches around stop codons in *S*.*cerevisiae* SK1.

In order to address the final point regarding software changes, we have found that comparing samples that have been sequenced with different generations of ONT sequencing kits or pores, or have been basecalled/mapped with different software can result in false positives. Consequently, it is necessary to restrict analysis to samples that were sequenced with comparable kits and used the same pipeline, tools, and parameters for basecalling and mapping of all samples.

We have included a new Discussion section where we comment on the impact of software (and hardware/wetware) changes to nanopore DRS.

5) The authors should show data on the correlation between sequencing depth and the ability to detect m^6^A modifications. They should also do deeper analysis to look at the nucleotide composition of the called m^6^A modifications that they capture, as well as whether there is potential bias towards capturing modifications on specific transcripts.

In order to examine the power of our differential error rate analysis approach at detecting m^6^A sites, we performed a bootstrapped subsampling approach using the nanopore DRS data derived from the synthetic MALAT1 RNAs (described in response to point 4), and example methylated positions in the Arabidopsis transcriptome. This analysis reveals that the ability to detect m^6^A from errors is dependent upon sequencing depth, as well as the fraction of reads which are methylated at the position in the methylation positive example.

Using the single replicate MALAT1 data, we can reliably detect m^6^A at the correct position in more than 95% of bootstraps using only 60 reads per condition. However, this represents an idealised scenario, as the synthetic reads are 100% unmethylated in one condition and 100% methylated in the other. We therefore also performed an *in silico* "dilution" experiment, where we diluted the bootstrapped sample of methylated reads with unmethylated reads and determined whether m^6^A could still be detected. Here we found that when 50% of reads in the methylation positive condition are sampled from the unmethylated condition, > 200 reads per condition are required to reliably detect m^6^A in > 95% of bootstraps, suggesting that the percentage of reads which are methylated at a position is an important factor in the power to detect differentially methylated sites.

When performing subsampling experiments using example methylated positions from the Arabidopsis nanopore DRS dataset, we find that the number of reads required to detect m^6^A varies between genes, and between methylated positions in the same gene. This is likely due to differences in the percentage of transcripts that are methylated at each position. We also cannot rule out that sequence context-specific error rates affect the sequencing depth required to detect m^6^A. However, we find that for some positions m^6^A can be reliably detected with as little as 10 reads per replicate (in a 4 replicate per condition experiment) suggesting that it is possible to detect m^6^A robustly at the threshold of a median coverage of 10 reads per replicate used in our original analysis.

These data have now been incorporated into the Results section of the revised manuscript and included as Figure 5—figure supplement 1E, F.

In order to address the second point of “a deeper analysis at the nucleotide composition of the called m^6^A modifications”, we examined the reference bases at differential error sites. We find that the error sites are not exclusively located at adenosine positions. This is explained by the fact that 5 RNA bases contribute to the nanopore signal at a particular moment in time. Consequently, the presence of a methylated adenosine in the mRNA can cause basecaller errors anywhere within approximately 5 nt of the methylated adenosine. Hence, detection of m^6^A using differential error rates has a resolution of around 9 nt, and multiple bases surrounding a single modified adenosine may be detected as having a differential error rate. Adenosines account for most error sites (61.3%). However, since m^6^A tends to occur in A rich sub-sequence (the most commonly detected methylated motifs in our dataset were AAm^6^ACA and AAm^6^ACU) it is also possible that this represents a mixture of methylated positions and adenosines surrounding methylated positions.

We previously characterised the motif that underpinned the differential error sites (Figure 5B in our original submitted manuscript) and detailed the frequency of different sequences matching the motif (Supplementary Figure 5B of our originally submitted manuscript). Notably, the two most frequently occurring motifs that we reported (AAACA and AAACU) match the two most enriched motifs detected in Arabidopsis Me-RIP peaks analysis reported by Chuan He and collaborators^10^ and also by Wan et al.^11^. In other words, the most frequently occurring m^6^A motifs we identify with nanopore DRS match those identified for Arabidopsis using orthogonal Me-RIP/m6A seq data produced and analysed by other labs.

We have included text on this further support to the validity of the approach we used into the revised manuscript.

6) Direct RNA sequencing benefits from the aspect that it is long read, single molecule sequencing. As such, there is the possibility of "phasing" modifications that cannot be performed using short-read sequencing. The authors should show some examples of RNAs with more or less modifications than expected and attempt to associate that with RNA processing to demonstrate the utility of direct RNA sequencing.

Indeed, the ultimate goal would be to distinguish modifications on a single read originating from an individual transcript molecule. We continue to work on this, with the aim of identifying relationships between m^6^A modification and other RNA processing events. However, at this stage, our method for detecting m^6^A is aggregated across all reads, meaning that true phasing is not possible.

To address the question posed by the reviewers, we developed an approach to test the relationship between mRNA 3' end position and the error profile of the nanopore DRS reads at detected differential error sites using only the methylated (VIRc) samples. Reads from VIRc samples, which overlapped error sites with identifiable m^6^A motifs detected in the *vir-1* vs VIRc comparison, were clustered by their 3’ position using a kernel density estimate with a σ of 13. Clusters with fewer than 10 reads were discarded. We then compared the error profiles of reads from different clusters at the error site, using a G-test approach, to identify whether different 3’ end positions had different levels of methylation. We identified 327 error sites which were phased with 3’ end position, overlapping 207 annotated genes. 149 (72%) of these genes also display a statistically detectable change in 3’ end position in the *vir-1* mutant compared to VIRc, indicating that the changes in 3’ end position in the *vir-1* mutant may result directly from the loss of m^6^A methylation. See *PRPL34 (AT1G29070*) as an example.

We have included these new analyses in to the Results section and as Figure 7C of the revised manuscript.

Reviewer #1:[…] I have some specific points detailed below.1) In the supplementary Table 1, only 25% and 27% of sequencings reads in the 5' adapter ligation library could be mapped to the TAIR10 genome. The authors should check the sequencing reads to explain in detail the reason for the failed alignment in a large majority of sequencing reads.

Our response to this point is covered in the Essential Revisions section, Point 2.

2) Subsection “Differential error site analysis reveals the m^6^A epitranscriptome”, 17,491 sites with a more than two-fold higher error rate in the VIR-complemented line with restored m^6^A. Are these error sites exclusively located in adenine positions? A nucleotide composition summary is required to improve clarity.

Our response to this point is covered in the Essential Revisions section, Point 5.

3) Subsection “Spurious antisense reads are rare or absent in nanopore DRS”, the absence of reads mapping to antisense to RCA suggests that spurious antisense is rare or absent from Nanopore direct RNA sequencing data. Is it a generally accepted that the genomic loci of RCA do not generate antisense transcripts? If so, this information should be included.

Antisense transcripts are not annotated at *RCA* in either the Araport 11 or TAIR10 annotations, so it is not generally accepted that antisense RNAs are found at this locus.

Nanopore DRS indicates antisense RNAs are rare or absent at *RCA*. To investigate this question with an orthogonal dataset, we consulted our Illumina RNA-seq dataset, in which we sequenced Col-0 and a *hen2-2* exosome mutant, prepared using ribosomal RNA depletion. Illumina RNAseq datasets have been shown to suffer from spurious antisense reads produced during library construction which complicate the detection of genuine antisense RNAs. Spurious antisense signals can be identified in count data using ERCC spike-in controls, which should not have antisense RNAs. The ratio of sense to antisense reads mapping to the ERCC spike-in controls can be used as an estimate of the level of spurious antisense signal. In our datasets we find that the ratios of sense to antisense reads at *RCA* are similar to those found in the ERCC spike-ins, and do not change in the *hen2-2* mutant (ind t-test *p* = 0.88), suggesting that the antisense reads detected at *RCA* in Illumina RNA-Seq data may be spurious and not from genuine antisense RNAs (Author response image 5).

**Author response image 5. respfig5:** Antisense Illumina RNA-seq reads at RCA are likely to be spurious. (**a**) Scatter plot showing correlation of sense and antisense read abundances derived from Illumina RNAseq data mapping to ERCC spike-in controls, shown in blue. Each point represents a single spike in transcript detected in a single RNAseq replicate (both Col-0 and *hen2-2*). Since ERCC spike-ins should not have antisense RNAs, this indicates the level of spurious antisense signal in the dataset. RCA, shown in the orange, does not display antisense RNA levels greater than the ERCC spike-ins, indicating that there are not likely to be genuine antisense RNAs at this locus. *FLC,* which is known to have antisense RNAs, is shown in green as a positive control. (**b**) Strip plot showing the sense/antisense read ratio at *RCA* does not change in the *hen2-2* mutant (T test *p*=0.88). Many genuine antisense RNAs, e.g. those at *FLC*, are unstable and are therefore upregulated in the *hen2-2* mutant relative to sense transcripts.

4) The webpage (https://github.com/bartongroup/Simpson_Barton_Nanopore_1) which is used to deposit scripts and pipelines on GitHub is not accessible. The code availability will represent a valuable addition to the Nanopore RNA sequencing community and can be used a guide for direct RNA methylation analysis.

We have prepared a GitHub site and ENA sites for sequence data (detailed in the submitted manuscript). Access to these resources will be made available upon publication. Indeed, our rationale for submitting this work to *eLife*, was so that all source data and code would be freely available, open access and directly linked to the figures in which the analysis is presented.

5) This is a technical paper to assess the performance of Nanopore direct RNA sequencing in a pioneer manner. Since m^6^A RNA modification detection and full-length RNA transcript sequencing using Cap-dependent capturing constitutes the major contribution of this paper, the recent advances relating to these topics should be included in the Introduction section.Liu H, et al. Accurate detection of m^6^A RNA modifications in native RNA sequences. bioRxiv 2019:525741.Jiang F, et al. Long-read direct RNA sequencing by 5'-Cap capturing reveals the impact of Piwi on the widespread exonization of transposable elements in locusts. RNA Biol 2019:1-10.

A manuscript corresponding to this pre-print has now been published in *Nature Communications* and is now cited in our revised manuscript.

Reviewer #2:[…] 1) This technique is conceptually similar to the work by Liu et al. (https://www.biorxiv.org/content/10.1101/525741v1). Please clarify the similarities/differences between Liu et al. and the method described in this manuscript.

The general principle of using nanopore Direct RNA Sequencing to identify RNA modifications is shared with that used in the pre-print Liu et al. (https://www.biorxiv.org/content/10.1101/525741v1), which is now published in *Nature Communications*^8^. Liu et al. used aggregated measurements of insertion, deletion, and mismatch frequencies at DRACH (where D=A or G or U, R=A or G, H=A or C or U) *k-mer*s, from RNAs transcribed *in vitro* using either ATP or m^6^ATP, to train a support vector machine model to classify methylated and unmethylated positions. There are several important similarities and distinctions between this methodology and our own:

The models produced by Liu et al.^8^, once trained on positive and negative examples, should be able to predict m^6^A in a normal sample without controls. In contrast, our method relies upon comparison between samples with methylation and reduced methylation controls.Like our method, the model produced by Liu et al.^8^ makes predictions for reference positions and cannot identify the presence or absence of m^6^A in individual reads.The models produced by Liu et al.^8^ should detect methylated positions in the analysed transcriptome independently of the writer complex or enzyme which catalyses the methylation. By contrast, our method, when using a mutant with reduced methylation such as *vir-1*, will only detect methylation sites which are lost in the mutant, e.g. VIR-dependent m^6^A sites.The models produced by Liu et al.^8^ rely on the prior knowledge of the DRACH motif to detect m^6^A, meaning that any modifications in other contexts will not be detected. In contrast, our method takes an unbiased statistical comparison approach to detecting m^6^A and is therefore sequence agnostic. We find that 4.8% of detected Arabidopsis m^6^A motifs are in the context AGAUU (Supplementary Figure 5B in our originally submitted manuscript), and 9.55% of detected yeast m^6^A motifs are in the context DGACG (see Author response image 3) indicating that the prior assumption of the DRACH motif will lead to false negatives.The data used to train the model from Liu et al.^8^ is produced entirely from *in vitro* transcribed RNAs. This imposes limitations to this approach:The *in vitro* transcribed RNAs will not contain other known RNA modifications (see https://mods.rna.albany.edu for a comprehensive list). These modifications are also likely to cause basecalling errors, which the machine learning model has not been exposed to. This means that when the model is applied to biological samples and encounters errors caused by alternative modifications, it may behave unexpectedly, e.g. by calling them as m^6^A. Our method is unlikely to incorrectly flag other RNA modifications as m^6^A, unless the mutant used as a negative control alters the pattern of these modifications in RNA.

Because the *in vitro* transcribed RNAs contain 100% methylated or 100% unmethylated adenosines only, biologically important *k-mer*s with mixtures of A and m^6^A, such as AAm^6^ACA or AAm^6^ACU, are not sampled. Liu et al.^8^ acknowledge this and therefore do not make predictions in 5mer contexts with more than one A. However, by our estimates this would lead to the failure to detect 92% of m^6^A sites we detected in Arabidopsis, and 76.6% of m^6^A sites we detected in *S. cerevisiae* SK1. The two most frequently occurring motifs that we report in Arabidopsis (AAm^6^ACA and AAm^6^ACU) match the two most enriched motifs detected in Arabidopsis Me-RIP peaks analysis reported by Chuan He and collaborators^10^ and also by Wan et al.^11^, providing orthogonal support for these estimates.

2) Base-calling error profiles can vary depending on the basecaller version, the model in the basecaller, the processing/filtering parameters used for running the basecaller, the pore protein version, and the motor protein version. Recently, there was a huge upgrade for the basecalling model (high-accuracy model, a.k.a. flip-flop basecaller) introduced in Guppy 3.2. Moreover, according to the Oxford Nanopore Technologies (ONT), a faster motor protein and the new R10 pore will be introduced to their DRS kits very soon. The authors need to clarify and inform the readers that the error profile presented here may change depending on the basecaller software, a model, a pore, and a motor protein.The method described in this manuscript can work in the specific combinations that the authors used, but it does not guarantee its performance in the other settings. Considering the change in the flagship basecaller, Guppy, with higher accuracy after the version used in the manuscript, the authors need to check if this method works with comparable accuracy with the newer Guppy.

The reviewer raises an important point, namely that there are many variables controlling the error profiles of nanopore DRS reads. Indeed, we have found that comparing samples that have been sequenced with different kits or pores, or have been basecalled/mapped with different software can result in large numbers of false positives. We have therefore restricted analysis to samples that were sequenced with comparable kits, pores and used the same pipeline, tools, and parameters for basecalling and mapping of all samples.

In order to address the reviewers comment that the newest versions of ONT's basecaller, Guppy, may not be able to be used in conjunction with error rate analysis to detect m^6^A, we re-basecalled the data derived from the sequencing of the synthetic MALAT1 RNA using Guppy version 3.2.4. This version uses the so-called "flip-flop" approach to achieve higher accuracy. Using this basecaller, we found that there were still statistically detectable changes in error profile at the synthetically modified m^6^A site (Author response image 6).

ONT sequencing and data analysis continues to evolve. Indeed, in the time since the reviewer made this comment ONT have announced another improvement to the basecaller using run length encoding, which may replace the flip-flop models implemented in Guppy in the near future.

However, our experience with machine learning models leads us to believe that so long as ONT do not explicitly train their models to identify m^6^A, it is likely to remain a source of errors in base- calling. This is because machine learning models typically do not generalise well outside the bounds of the data they are trained on. Even if the training datasets do include unlabelled examples of m^6^A, due to the extreme imbalance of m^6^A and A, they are unlikely to be well modelled by the basecaller.

We have included text to reflect this in a new Discussion section of the manuscript.

**Author response image 6. respfig6:** Comparison of differential error rate analysis in MALAT1 data using older (3.1.2) and newer (3.2.4) Guppy releases. Gene track showing the change in the per- reference base test statistic attained when comparing methylated and unmethylated synthetic MALAT1 fragments basecalled using Guppy 3.1.2, which uses recurrent neural network basecalling, and Guppy 3.2.4, which uses recurrent neural network basecalling with a flip-flop architecture.

3) In addition, Tombo from the ONT has a mode called "level_sample_compare" that enables the modified base detection by comparing control and experiment groups. Can you discuss a bit about the benefits and drawbacks of your method in comparison with Tombo?

The performance of Tombo has been evaluated by Liu et al.^7^; using yeast Me-RIP/m6A-seq data as a ground truth. The authors suggested that EpiNano had an increased precision over Tombo, but recovered fewer ground truth sites^7^. We understand that ONT is no longer supporting the further development of Tombo and are instead focussing on the development of a new tool called Megalodon for modification detection. However, this tool does not yet have support for RNA.

4) Subsection “Nanopore DRS confirms sites of RNA 3′ end formation and estimates poly(A) tail length”: The standard nanopore DRS library preparation uses the double-stranded RNA-DNA ligation assisted by an oligo(dT) splint. It inevitably introduces a substantial underrepresentation of short poly(A) tails. Even 10 nt oligo(dT) splint often results in a strong bias for > 20 nt poly(A) tails against the shorter tails. Moreover, short poly(A) tails often carry additional U tails which interfere with the ligation to the adapter. In addition, the means of the single-read "poly(A) length measurements" may not be a suitable summarization to estimate the means of "poly(A) lengths." Nanopolish gives the best approximations of poly(A) lengths at a single-read level. However, as the poly(A) dwell time roughly follows a gamma distribution with a long tail, the mean of the best approximations at a single-read level systematically overestimates the mean of poly(A) length.

The reviewer makes very good points and we thank them for their informed insight. We have revised our wording in the manuscript text to refer to the poly(A) length of the sequenced reads rather than Arabidopsis poly(A) tail length. In addition, we have changed our data analysis and edited our text to use the median rather than the mean in these contexts. In addition we have included mention of the impact of short poly(A) tails and uridylated tails might have on quantitative gene expression analysis using the current standard nanopore DRS procedure.

5) Subsection “Cap-dependent 5′ RNA detection by nanopore DRS”: The authors used RNA ligase 1 to mark the 5′ ends of RNAs. Due to the substrate specificity of this enzyme, circularized mRNAs and mRNA-mRNA concatamers might have been produced.

We first investigated whether mRNA-mRNA concatemers were produced in the cap-capture datasets. In our submitted manuscript, we indicated the presence of “over-splitting” of read signal in our DRS datasets, caused by the misinterpretation of signal originating from a single RNA molecule as two molecules. Given that over-splitting can occur, we suggest that “under-splitting” of signal that originates from two unique molecules sequenced consecutively through the same pore may also occur. These will generally be less problematic than over-splitting events, as the chances of under- splitting occurring with two transcripts that have been transcribed from adjacent genomic loci is very low, meaning that they will seldom lead to misidentification of chimeric RNAs. Once base- called, however, these under-split reads would be indistinguishable from physically concatemerized mRNAs catalysed by RNA ligase 1. To identify whether under-splitting occurs in all of our datasets, and whether RNA ligase 1-catalysed concatemers occur in our adapter ligated datasets, we extracted soft-clipped regions from the 5’ and 3’ ends of read alignments to the TAIR10 reference, and remapped them to determine what percentage were alignable to the genome. We found that the percentages of alignable softclipped regions were consistently low in both adapter ligation negative and positive datasets (between 0.04 and 0.27% overall for all experiments), suggesting that the low levels of spurious concatemers are likely to be *in silico* under-splitting errors (Author response image 7).

**Author response image 7. respfig7:** Prevalence of concatemerised reads in nanopore DRS data. Barplot showing the percentage of read alignments in nanopore DRS data where softclipped regions (i.e. unmapped sequence at 5’ and 3’ ends of alignments) which is mappable elsewhere in the TAIR10 genome, possibly indicating undersplitting of read signal by the MinKNOW software. Adapter ligated datasets do not show increased levels, suggesting that RNA ligase I does not cause a detectable increase in the number of concatemers/chimeric reads.

Supplementary Table 1 shows that the libraries using the standard protocol yielded ~90% of mappable reads while the libraries with cap-dependent ligation yielded only ~25%. Please describe what the other 75% are.

Our response to the reviewer’s comment on the percentage of mappable reads in the adapter ligated datasets is given in the Essential Revisions section Point 2.

Also, it would be helpful if the sequence information of the 5′ adapter is presented in the manuscript.

The sequence of the 5’ adapter RNA has now been added to the Materials and methods section.

6) Subsection “Differential error site analysis reveals the m^6^A epitranscriptome”: How many of the miCLIP peaks could be detected with nanopore DRS? The authors only show the analyses using the population-level detection of m^6^A -modified sites. A real benefit of nanopore DRS lies in the associative analyses at a single-read level. Is it possible to use the DRS method to call m^6^A -modified sites within single molecules and analyze to see if a modified RNA has different polyadenylation status or 3′ end position from those in an unmodified RNA?

Whilst approaching this question we discovered an error in our miCLIP analysis pipeline that meant that the results from the irreproducible rate analysis were not thresholded correctly. Upon correcting this mistake we found that because replicate 3 of our miCLIP experiment was less well correlated with replicates 1 and 2 (spearman's rho 0.54 for replicates 1 vs. 2, 93,046 reproducible sites, spearman's rho 0.42 for replicates 1 vs. 3, 12,777 reproducible sites, spearman's rho 0.47 for replicates 2 vs. 3, 19,948 reproducible sites), it was reducing the quality of the results, and decided to exclude it from the downstream analysis. This has altered our analysis and some results slightly. The Materials and methods and Results sections of the manuscript have been updated accordingly.

We now find that 65.7% of nanopore error sites are within 5 nt of an miCLIP site detected in two replicates at IDR < 0.05 (Figure 5E, 5F in the revised version of manuscript). Conversely, 31.2% of miCLIP peaks are within 5 nt of an error site detected with nanopore DRS. This is likely because miCLIP is a more sensitive but less precise technique (i.e. has antibody-dependent limitations) than nanopore DRS for mapping m^6^A: we detected 93 thousand significant miCLIP sites compared to only 17 thousand nanopore differential error sites. Furthermore, miCLIP can detect a wider range of methylated adenosines in the transcriptome. Here we report a differential error rate method capable of detecting m^6^A sites dependent on VIR, in poly(A+) RNA. m^6^A sites in poly- or oligo- adenylated RNAs, which are detectable by miCLIP, but which are methylated by other enzymes or writer complexes are not detected by our experimental approach.

We address the issues regarding single molecule detection and phasing in the Essential Revisions section Point 6.

I feel that the current discoveries related to m^6^A modification in this manuscript can be better done using miCLIP than nanopore DRS.

To clarify, we carried out 3 biological replicates of miCLIP in this study using wild type *Arabidopsis thaliana* Col-0 poly(A)+ RNA. Consequently, in addition to our nanopore DRS datasets, we provide here the first miCLIP-based map of m^6^A in any plant species. Compared to the read depth of our current nanopore DRS datasets, our miCLIP data does indeed detect more significant sites using the statistical approach described.

However, antibody-based approaches to mapping m^6^A, such as miCLIP, are limited in some respects because (i) they cannot readily define stoichiometry (ii) they do not necessarily detect m^6^A only (iii) they have unclear sensitivity. These limitations account, in part, for the recent high-profile publication of the MAZTER-Seq procedure (published while our manuscript was in review) as an antibody-independent approach to mapping m^6^A that addresses these issues (albeit for a sub-set of sequence contexts)^7^.

The overlap of miCLIP sites mapped in different studies is low. Examination of 81,519 sites identified across six different published miCLIP experiments in human cell lines revealed a median of only 1,500 sites shared between any two datasets^7^. This appears to be explained by sites modified at low stoichiometries that are stochastically identified in one miCLIP experiment, but not another, due to the low efficiency of UV cross-linking^7^.

We used miCLIP as an orthogonal technique here, using only wild-type material. Although we detect enrichment of miCLIP sites in 3’UTRs (Figure 5D and Figure 5—figure supplement 1D in the revised version of manuscript), the consensus sequence associated with miCLIP is oligo(A) – indicating either the identification of sites distinct from those written by the mRNA m^6^A writer complex, or cross- hybridisation of the antibody to epitopes other than m^6^A. We used the Synaptic Systems anti-m^6^A antibody for miCLIP studies because it is widely used in the field and because the errors induced in miCLIP sequencing libraries had already been established^6^. However, it has also previously been shown that this antibody cross-reacts with non-methylated purine-rich RNA sequences^5^, highlighting limitations to this approach.

We use nanopore DRS datasets to link m^6^A to changes in RNA abundance and processing (specifically in relation to 3’ end processing), in the same experiment. For an miCLIP-only approach to provide the same insight, further analysis would be required, such as (i) control procedures for mapping m^6^A sites in a methylation defective mutant that are not yet routinely established or accepted, (ii) different datasets of RNAseq to investigate changes in RNA levels, (iii) further datasets of specialised RNAseq to map 3’ ends (eg 3P-Seq^12^), (iv) since we demonstrate that some of the changes in pre-mRNA processing in the absence of VIRILIZER-dependent m^6^A (chimeric RNAs, for example) are distinct from events annotated in TAIR10, Araport11 and AtRTD2, long read sequencing of the *vir-1* transcriptome with PacBio, for example, would be necessary to establish an appropriate Reference Transcriptome to be used to quantify the different de novoRNA processing events identified in this study. Overall, nanopore DRS can give m^6^A positions in the context of full- length RNA reads which can be used for other analyses, but miCLIP cannot.

Reviewer #3:[…] The authors use cap-dependent ligation to enrich for capped mRNAs, reducing the 3' end bias observed in DRS. It would be of interest to discuss if the transcripts with early 5' ends result from technical artifacts (for example, degradation during RNA preparation) or represent transcripts present in the cell that lack a cap structure that is compatible with the ligation protocol. For example, the authors demonstrate that nanopore DRS can identify rare anti-sense transcripts. Could transcripts with early 5' ends, which are selected against in the cap dependent libraries, represent rare transcripts or degradation intermediary products?

We thank the reviewer for their insightful comment. We had considered it difficult to clearly distinguish what might be biology from what might be experimental induced RNA decay aside from our RNA quality control measures. Consequently, we focused instead on capturing RNAs that did have a 5’ cap to establish which RNAs were full-length, by at least this criterion. However, in response to the reviewer's question, we looked at this in more detail.

To address what might be biology-dependent decay in the datasets that we had, we asked if there was an enrichment of 5’ nanopore DRS reads terminating in proximity to established miRNA cleavage sites in our datasets prepared without cap capture. We approached this question in a similar way to that used by Michael Nodine’s recent study using Illumina nanoPARE data^3^. Target sites for known Arabidopsis small (s)RNAs (miRNAs, tasiRNAs) in protein coding transcripts were predicted using GSTAr. To prevent false positives caused by poor alignments and full-length mRNAs, target sites spanning introns and in 5'UTRs were filtered out. For each putative target site, the number of nanopore DRS reads terminating in a 20 nt region upstream and downstream of the cleavage site was calculated and used to produce an enrichment score for reads terminating immediately downstream of the site. This method was used instead of measuring 5' ends precisely at the cleavage site, because as we demonstrated in our study, approximately 11 nt are lost from the 5' end of RNA reads in standard nanopore DRS. Enrichment scores and Allen scores for sRNA-target alignments were tested for significance using the same permutation testing method as in Schon et al.^3^ sRNA sequences were shuffled 1,000 times and target sites for shuffled sequences were predicted. Null distributions for Allen scores and enrichments scores were then produced and used to generate *p* values for significance. Enrichment and Allen score *p* values were combined using Fisher's method and multiple testing correction was applied.

Using this approach, we were able to identify peaks of downstream cleavage products in a number of well-established target mRNA-miRNA interactions. For example, at mRNA encoding HAM1 (Figure 3—figure supplement 1H in the revised version of manuscript) and HAM2 (miR170/miR171); NAC1 (miR164); ARF proteins (miR160); and SPL domain containing proteins (miR156/7). Nanopore DRS reads aligning to the non-protein coding *TAS* genes also reveal clear evidence of established cleavage products (Author response image 8). The full list of identified miRNA targets identified in nanopore is now included in the manuscript as Supplementary file 4.

**Author response image 8. respfig8:** Nanopore DRS captures downstream products of miRNA cleavage at transcripts encoded by TAS genes. Orange, 5’ coverage from capped nanoPARE reads; purple, 5’ coverage from uncapped nanoPARE reads; blue, nanopore DRS 5’ coverage; black miRNA target site alignment; black, Araport11 annotation.

We conclude that, in principle, nanopore DRS can be used to reveal sites of internal mRNA (and *TAS* ncRNA) cleavage and we thank the reviewer for making this suggestion. Since we have shown that nanopore DRS does not sequence the extreme 5’ end of an RNA accurately, the ideal way to address this question in the future would be to compare adapter ligated libraries that were cap-dependent with those that were cap-independent in a conceptually similar approach to that used in Michael Nodine’s nanoPARE study^3^. In addition, the use of mutants defective in the Arabidopsis cytoplasmic 5’-3’ exonuclease XRN4 would be useful in revealing degradation intermediary products that are otherwise too unstable to be readily detected.

Additionally, the authors leverage DRS to expand the set of annotated transcripts and splicing isoforms. Can the authors use this data to describe new endogenous targets of NMD, poison exons, or find examples of proteins with new functional domains?

Indeed, our study suggests that nanopore DRS can provide a useful complementary approach to revealing new details of the Arabidopsis transcriptome. Consequently, this approach can be used in the analysis of the sequence features listed by the reviewer and possibly others too (e.g. identifying uORFs).

We can, for example, identify well-established examples of cassette exons, and alternative donor and acceptor sites which are known to be targets of the NMD pathway as indicated by upregulation in *upf1* mutants^13^ (Author response table 3). This includes, for example, mRNA encoding the splicing factor SR34a, in which inclusion of a cassette exon in the first intron has been suggested to introduce a destabilising uORF^13^ (Author response table 3). We also identified further examples of unannotated alternative splice isoforms which may introduce frameshifts, such as an exon skipping event at mRNA encoding the KH domain- containing protein AT5G56140 (Figure 4—figure supplement 1C in the revised version of manuscript), and an alternative acceptor site at *XRCC* mRNA (Figure 4—figure supplement 1D in the revised version of manuscript), establishing the principle that nanopore DRS can be used to identify novel alternative splicing products which may be targets of NMD. We suggest that the best way to study these phenomena in the future would be to use biological material with genotypes that disrupt the processes of interest in order to provide a means to validate the events under study. For example, sequencing an NMD mutant such as *upf1,* as was previously done with Illumina RNAseq, would validate the discovery of potential new NMD substrates.

**Author response table 3. resptable3:** Known *upf1* sensitive alternative splicing events which are supported by nanopore DRS read alignments. Table showing examples of NMD sensitive alternative splicing events taken from Table 1 of Kalyna et al.^13^ which are supported by nanopore DRS reads. Duplicate genes and unsupported examples have been removed from the table.

Gene ID	Gene Name	Alternative splicing class
*AT1G55310*	*At-SCL33*	Exon 4 alternative acceptor
*AT2G37340*	*At-RS2Z33*	Exon 3 alternative acceptor
*AT4G16845*	*VRN2*	Intron 2 retention
*AT4G39260*	*GRP8/CCR1*	Exon 1 alternative donor
*AT1G77080*	*MAF1*	Exon 4 alternative acceptor
*AT2G02960*	*Zinc finger (C3HC4) protein*	Exon 2 alternative acceptor
*AT2G46790*	*APRR9/TL1*	Exon 2 alternative donor
*AT2G38880*	*NF-YB1/HAP3a*	Exon 6 alternative donor
*AT3G49430*	*At-SR34a*	Intron 1 cassette exon
*AT3G01150*	*At-PTB2a*	Intron 2 Cassette exon, exon 8 alternative donor
*AT3G13570*	*At-SCL30a*	Intron 3 cassette exon
*AT3G53500*	*At-RS2Z32*	Exon 3 alternative acceptor
*AT3G61860*	*At-RS31*	Intron 2 cassette exon
*AT2G21660*	*GRP7/CCR2*	Exon 1 alternative donor
*AT5G53180*	*At-PTB2b*	Intron 3 cassette exon
*AT4G25500*	*At-RS40*	Intron 2 cassette exon
*AT3G55460*	*At-SCL30*	Intron 3 cassette exon
*AT2G29210*	*Splicing factor PWI protein*	Exon 6 alternative acceptor
*AT1G07830*	*RPL29 family*	Exon 1 alternative donor
*AT1G02090*	*FUS5/CSN7/COP15*	Exon 8 alternative acceptor
*AT1G72560*	*PSD/Exportin-t*	Exon 13 alternative donor
*AT4G33060*	*CYP57*	Intron 5 cassette exon
*AT5G65060*	*MAF3*	Exon 4 alternative acceptor
*AT5G35410*	*SOS2/CIPK24*	Exon 9 alternative acceptor
*AT4G36960*	*RRM-containing protein*	Intron 1 retention

These findings are now added to the Results section relating to splicing analysis.

In addition to the detection of novel alternative splicing events, nanopore DRS is also able to identify novel alternative cleavage and polyadenylation (APA) sites. For example, PTM is a plant homeodomain transcription factor with C-terminal transmembrane domains that sequester it to the chloroplast. PTM is cleaved from the membrane in high light conditions and the N-terminal fragment translocates to the nucleus, where it acts as a retrograde transcription factor positively regulating flowering through the control of *FLC*^14^. We find that alternative polyadenylation in intron 10 of *PTM* mRNA is common and produces transcripts with a truncated open reading frame which removes the C-terminal transmembrane domains (Figure 2—figure supplement 1D in the revised version of manuscript). Translation of these APA transcripts would be predicted to produce transcription factors which have similar activity to the cleaved N-terminal fragment generated from full length PTM. Consequently, regulation of this APA site may provide an alternative mechanism for the production of active PTM that bypasses the retrograde signalling pathway.

This finding is now included in the Results section related to RNA 3’ end detection.

One feature of the transcriptome the authors focus on is the localization and effect of m^6^A modification on RNA metabolism. It would be informative if in addition to the ERCC controls, a set of RNAs with known m^6^A sites were also included.

This point is addressed in response to the Essential Revisions Point 4.

One aspect of m^6^A biology that can't easily be studied with current methodologies is the stoichiometry of the modification at each position in each transcript. Can the authors comment at all on stoichiometry of m^6^A from DRS? Furthermore, can the authors comment on how many reads per transcript are necessary to detect a modification? Is there a bias towards more abundant transcripts?

These points are addressed in response to Essential Revisions Point 5.

In mammalian cells it has been shown that modified RNAs tend to have shorter 3' UTRs (Molinie et al., 2016)(PMID: 27376769). Can this observation be tested in DRS, and if so, does the same phenomena occur in Arabidopsis?

When considering the impact of loss of m^6^A in the *vir-1* m^6^A defective mutant on 3’ end formation, we found a clear directional shift to proximal poly(A) site selection. These findings were reported in Figure 6E of our originally submitted manuscript. In addition, our analysis of phasing, described in Essential Revisions section Point 6, revealed a clear directional shift to more proximal poly(A) site selection in transcript isoforms detected from the same gene in m^6^A defective mutant backgrounds compared to complemented line VIRc (see Figure 7C in the revised version of manuscript).

We have now also addressed this question in an orthogonal manner. We used the DaPars^15^ algorithm which is designed to detect shifts in RNA 3’ end formation in Illumina RNAseq data. We used our Illumina RNAseq datasets that we carried out in parallel to nanopore DRS, comparing the *vir-1* mutant with the VIR complemented line. We find that DaPars analysis also identifies a global shortening of 3'UTRs in the *vir-1* mutant (Figure 7—figure supplement 1B in the revised version of manuscript).

We have incorporated the DaPars analysis (and the phasing analysis) into our revised manuscript.

Our interpretation is that there is not yet a clear consensus on all the possible interrelationships between m^6^A and 3’ end formation in published mammalian cell studies. For example, Ke et al.^16^, reported that dense methylation was associated with an inhibition of proximal poly(A) site selection and the use of more distal poly(A) sites. Therefore, our findings are consistent with the suggestion made by Ke et al.^16^ that m^6^A could inhibit the use of proximal sites and hence favour longer 3’UTR transcript isoforms of the same gene. Moline et al. state the following in the Discussion of their publication:

“A recent study reported that genes with longer last exons have a higher density of m^6^A peaks. Consistent with this observation, we found that m^6^A levels of genes are positively correlated with maximum 3′-UTR lengths as measured in input steady-state RNAseq data (Supplementary Figure 5E). Detailed analyses showed that genes with higher proximal APA usage in the m^6^A-positive fraction (P/D; m^6^A+ > m^6^A−) had significantly longer 3′UTRs (Supplementary Figure 5F) and lower ratios of proximal versus distal signals (Supplementary Figure 5G,H) in the input RNAseq data as compared to genes with higher distal APA usage in the m^6^A-positive fraction (P/D; m^6^A+ < m^6^A−) or no change in APA usage between fractions (P/D; m^6^A+ = m^6^A−).”

The further analysis of 3’ end formation prompted by this reviewer’s question and the phasing analysis requested in Essential revisions point 6 led us to look more closely at the shift in 3’ formation. When we plotted the position of the most-frequently occurring motifs at differential error sites and the poly(A) signal AAUAAA (and single nucleotide variants) we detected a close overlap in position (Figure 7E in the revised version of manuscript). We have included this new finding into the Results section of the revised manuscript.

Lastly, can the authors comment on the ability of detecting modifications other than m^6^A through DRS?

We have not yet examined other RNA modifications. A crucial tool in such analyses is the availability of a genotype that disrupts the modification. However, the ability to use nanopore to map other modifications in DNA and indeed nucleotide analogues^17^ has already been demonstrated.

Consequently, we would anticipate that other RNA modifications will be detectable by nanopore DRS.

**References**

1 Pontefract, A., Hachey, J., Zuber, M. T., Ruvkun, G. and Carr, C. E. Sequencing nothing: Exploring failure modes of nanopore sensing and implications for life detection. *Life Sci Space Res (Amst)*
**18**, 80-86, doi:10.1016/j.lssr.2018.05.004 (2018).

2 Mojarro, A., Hachey, J., Ruvkun, G., Zuber, M. T. and Carr, C. E. CarrierSeq: a sequence analysis workflow for low-input nanopore sequencing. *BMC Bioinformatics*
**19**, 108, doi:10.1186/s12859-018-2124-3 (2018).

3 Schon, M. A., Kellner, M. J., Plotnikova, A., Hofmann, F. and Nodine, M. D. NanoPARE: parallel analysis of RNA 5' ends from low-input RNA. *Genome Res*
**28**, 1931-1942, doi:10.1101/gr.239202.118 (2018).

4 Liu, N. et al. Probing N6-methyladenosine RNA modification status at single nucleotide resolution in mRNA and long noncoding RNA. *RNA*
**19**, 1848-1856, doi:10.1261/rna.041178.113 (2013).

5 Schwartz, S. et al. High-resolution mapping reveals a conserved, widespread, dynamic mRNA methylation program in yeast meiosis. *Cell*
**155**, 1409-1421, doi:10.1016/j.cell.2013.10.047 (2013).

6 Linder, B. et al. Single-nucleotide-resolution mapping of m6A and m6Am throughout the transcriptome. *Nat Methods*
**12**, 767-772, doi:10.1038/nmeth.3453 (2015).

7 Garcia-Campos, M. A. et al. Deciphering the "m6A Code" via antibody-independent quantitative profiling. *Cell*
**178**, 731-747 e716, doi:10.1016/j.cell.2019.06.013 (2019).

8 Liu, H. et al. Accurate detection of m(6)A RNA modifications in native RNA sequences. *Nat Commun*
**10**, 4079, doi:10.1038/s41467-019-11713-9 (2019).

9 Garcia-Campos, M. A. et al. Deciphering the "m(6)A Code" via Antibody-Independent Quantitative Profiling. *Cell*
**178**, 731-747 e716, doi:10.1016/j.cell.2019.06.013 (2019).

10 Luo, G. Z. et al. Unique features of the m6A methylome in *Arabidopsis thaliana. Nat Commun*
**5**, 5630, doi:10.1038/ncomms6630 (2014).

11 Wan, Y. et al. Transcriptome-wide high-throughput deep m(6)A-seq reveals unique differential m(6)A methylation patterns between three organs in *Arabidopsis thaliana. Genome Biol*
**16**, 272, doi:10.1186/s13059-015-0839-2 (2015).

12 Jan, C. H., Friedman, R. C., Ruby, J. G. and Bartel, D. P. Formation, regulation and evolution of *Caenorhabditis elegans* 3 ' UTRs. *Nature*
**469**, 97-101, doi:10.1038/nature09616 (2011).

13 Kalyna, M. et al. Alternative splicing and nonsense-mediated decay modulate expression of important regulatory genes in Arabidopsis. *Nucleic Acids Res*
**40**, 2454- 2469, doi:10.1093/nar/gkr932 (2012).

14 Feng, P. Q. et al. Chloroplast retrograde signal regulates flowering. *P Natl Acad Sci USA*
**113**, 10708-10713, doi:10.1073/pnas.1521599113 (2016).

15 Xia, Z. et al. Dynamic analyses of alternative polyadenylation from RNA-seq reveal a 3'-UTR landscape across seven tumour types. *Nat Commun*
**5**, 5274, doi:10.1038/ncomms6274 (2014).

16 Ke, S. et al. A majority of m6A residues are in the last exons, allowing the potential for 3' UTR regulation. *Genes Dev*
**29**, 2037-2053, doi:10.1101/gad.269415.115 (2015).

17 Muller, C. A. et al. Capturing the dynamics of genome replication on individual ultra- long nanopore sequence reads. *Nat Methods*
**16**, 429-436, doi:10.1038/s41592-019- 0394-y (2019).

18 Gierlinski, M. et al. Statistical models for RNA-seq data derived from a two-condition 48-replicate experiment. *Bioinformatics*
**31**, 3625-3630, doi:10.1093/bioinformatics/btv425 (2015).

19 Schurch, N. J. et al. How many biological replicates are needed in an RNA-seq experiment and which differential expression tool should you use? *RNA*
**22**, 839-851, doi:10.1261/rna.053959.115 (2016).